# Anthropogenic pressures driving the salinity intrusion in the Guadalquivir Estuary: Insights from 1D Numerical Simulations

Sara Sirviente[1], Juan J. Gomiz-Pascual[1], Marina Bolado-Penagos[1], Sabine Sauvage[2], José M. Sánchez-Pérez[2], Miguel Bruno[1.]

[1]Department of Applied Physics, Faculty of Marine and Environmental Sciences, Marine Research Institute (INMAR), International Campus of Excellence of the Sea (CEI·MAR), University of Cadiz, Puerto Real, 11510 Cadiz, Spain

[2]Centre de Recherche sur la Biodiversité et l'Environnement (CRBE), Université de Toulouse, CNRS, IRD, Toulouse INP, Université Toulouse 3 – Paul Sabatier (UT3), Toulouse, France

*Correspondence to*: Sara Sirviente (sara.sirviente@uca.es)

**Abstract.** The study presents a dynamic analysis of the present day behavior of salinity concentration in the Guadalquivir estuary and evaluates the impact of anthropogenic pressures on the dynamics of the horizontal salinity gradient. A one-dimensional (1D) hydrodynamic model with an advection and dispersion module is used to study the effects of human pressure involved in the salinity concentration along the estuary. The observations, which correspond to continuous measurements taken during different oceanographic campaigns from 2021 to 2023, show an excessive salt intrusion in the estuary (with salinities of 5 psu at 60 km from the mouth) as compared to the idealized situation when anthropogenic water withdrawals are absent. This highlights the need to include a water withdrawal term in the simulations to accurately reproduce the system's real behavior, thereby reflecting the magnitude of anthropogenic pressures. The model successfully reproduces the observations when this forcing factor is included. Under constant low flow conditions, experiments show that increasing water withdrawals leads to an increase in the horizontal salinity gradient upstream. Similarly, under constant water withdrawal conditions, a decrease in the horizontal salinity gradient is observed when freshwater flows exceed 40 m³ s⁻¹. Variations in anthropogenic pressures, such as water withdrawals for agriculture or saline industry and reductions in freshwater flow, play a fundamental role in the evolution of salinity along the estuary. Under the current circumstances, the Guadalquivir estuary requires an urgent regulation of these uses in order to avoid further damage on the aquatic ecosystems.

## 1. Introduction

Estuaries are considered one of the most valuable and threatened ecosystems in the world (Lopes et al., 2024). Under optimal conditions, estuary mouths are biologically more productive than rivers and adjacent ocean zones because of the high concentration of nutrients that stimulates higher primary productivity (Miranda et al., 2017). The characteristics of estuarine communities are strongly and directly linked to physical parameters, such as temperature, turbidity, and salinity (Dauvin, 2008). These coastal regions play a key role in supporting significant socioeconomic and environmental activities, serving as

essential nursery habitats on continental shelves and providing invaluable ecosystem services to other ecological systems (Donázar-Aramendía et al., 2019). However, estuaries have been severely degraded by anthropogenic activities. In recent decades, land use for agriculture and changes in channel geomorphology for navigation and other purposes have altered the

natural hydrodynamics, morphology, and water quality conditions of these systems (Sirviente et al., 2023; Gomiz-Pascual et al., 2021). Water abstraction for activities such as irrigation, domestic use, and industrial activities is the most significant factor contributing to the degradation of these ecosystems (Algaba et al., 2024)

The Guadalquivir River Estuary (GRE) (Fig. 1) is a positive estuary, in which the freshwater discharges from the basin are sufficient to compensate for evaporation losses (Elliot and McLusky, 2002; Diez-minguito et al., 2013; Losada et al., 2017).

It is generally considered a well-mixed estuary, though this characteristic can change during periods of high discharge, when mixing conditions deviate from the typical pattern (Álvarez et al., 2001). It is a meso-tidal estuary with semi-diurnal tidal components being the most energetic (Álvarez et al., 2001). Flowing into the Atlantic Ocean, this estuary is one of the most important socioeconomic areas in southern Spain. It extends 110 km from its mouth in Bonanza to its head at the Alcalá del Río dam (Fig. 1a), with the first 85 km being navigable, making it the only navigable river in Spain (Navarro et al., 2011).

The dam (black triangle in Fig. 1) strongly regulates freshwater input to the estuary by blocking upstream tidal waves, creating a closed boundary in this semi-enclosed channel, causing the tidal wave to arrive with sufficient energy to reflect and interact with the incident wave (Diez-Minguito et al., 2012). The largest freshwater input (80%) to the estuary is through the dam, the last point in the extensive network of reservoirs that regulate the GRE basin. This estuary is characterized by low-flow conditions ($< 40$ m$^3$s$^{-1}$) most of the time (75% of the year), where tidal dominance and well-mixed conditions prevail (Siles-

Ajamil et al., 2019). In contrast, during high-flow conditions ($> 400$ m$^3$s$^{-1}$), tidal propagation is disrupted, increasing the potential energy of the water column and decreasing saline intrusion (Diez-Minguito et al., 2013; Ruiz et al., 2015).

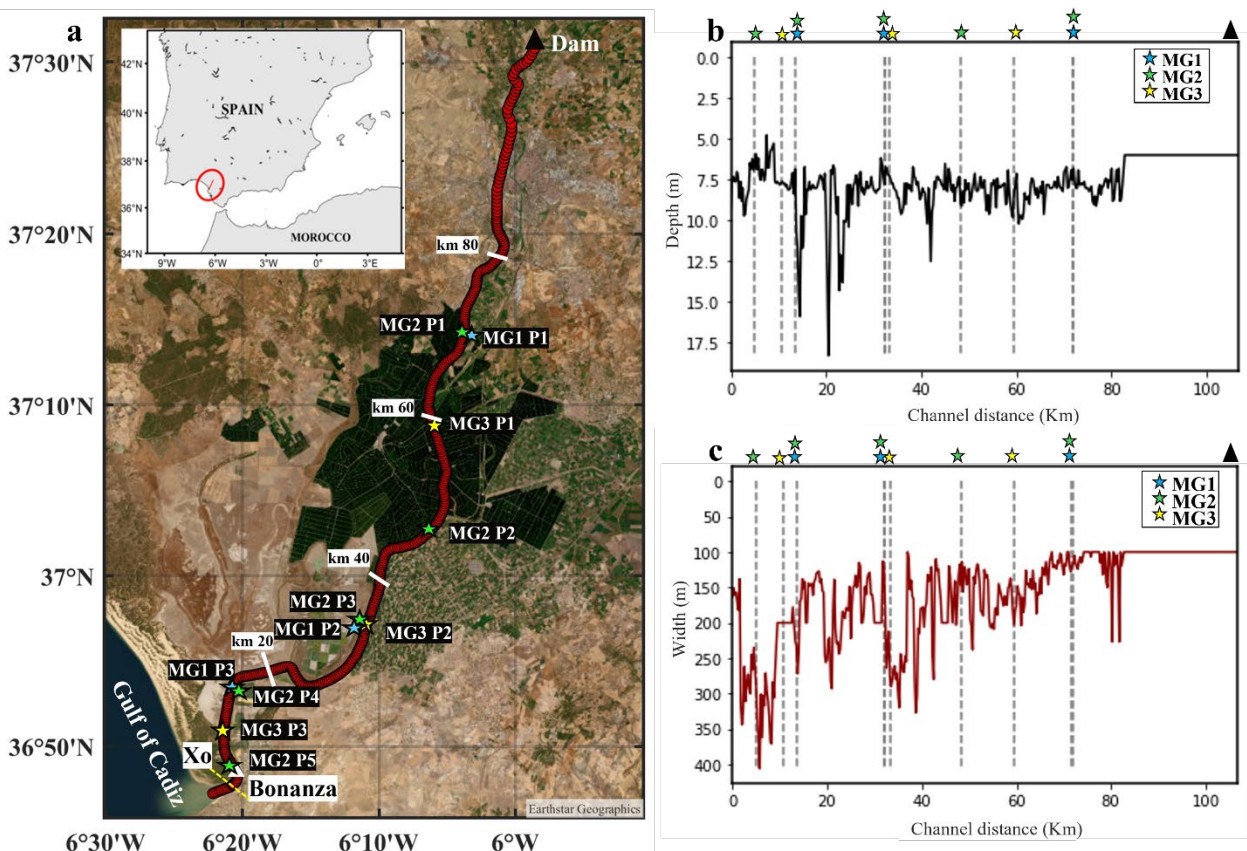

**Figure 1: (a) Geographic region of the study area. Xo represents the starting section of the simulations; the white triangle indicates Bonanza point. The blue, green and yellow stars represent the sample points for each campaign MG1, MG2, and MG3, respectively. The dam is defined by a black triangle. (b) Depth profile and (c) width profile (m) along the entire channel (km) obtained by the nautical chart of 2019 developed by the Hydrographic Institute of the Spanish Navy. The dashed gray lines represent the position of each sampling point.**

The GRE is highly valued for its rich biodiversity and serves as a critical habitat for several economically important marine species. It plays a crucial role in feeding coastal areas along the Gulf of Cadiz, promoting high levels of primary productivity in the region (Ruiz et al., 2015; Cañavate et al., 2021). However, it is also a source of economic and environmental conflict due to the coexistence of multiple activities (salt production, agriculture -especially rice, which requires large amounts of freshwater- fishing and navigation). Extensive human modification of the estuary has led to a significant reduction in the original marshes and a reduction in their original length to accommodate agricultural and navigation needs (Ruiz et al., 2017). This high anthropogenic pressure has caused changes in both the hydrodynamics and morphology of the system, favoring the constant degradation of the ecosystem (Mendiguchía et al., 2007; Ruiz et al., 2015: Siles-Ajamil et al. (2019); Sirviente et al., 2023).

The detrimental effects of anthropogenic activities have been demonstrated in other estuaries around the world. Alcérreca-Huerta et al. (2019) show that the construction of a dam system in the estuary of the Grijalva River (Mexico) in 1959 altered

the hydrological regime, reducing the seasonality of water discharge and decreasing the amount of available freshwater. This, together with changes in land use (more agricultural land, less mangrove cover and less vegetation), leads to variations in salinity concentration, with saline intrusion observed up to 46 km upstream, with salinity levels reaching 32.8 psu. Studies such as Huang et al. (2024), based on numerical simulations using a 3D model, show that anthropogenic activities, in particular the regulation of freshwater flows by infrastructure projects, are drastically changing the dynamics of saline intrusion in the

Changjiang River estuary (China). This study shows how an increase in freshwater flows (due to releases from the Three Gorges Reservoir) mitigate the advance of saline intrusion. However, water withdrawals in the city of Yangzhou as part of the implementation of the East Route of the South-to-North Water Transfer Project will inevitably lead to a reduction in inflow during the dry season, resulting in an increase in saline intrusion in this system by approximately 6-7 km. This relationship between salinity and freshwater flow was previously observed by Webber et al. (2015), who assessed the effects of the Three

Gorges Dam, the South-to-North Water Transfer Project, and local water withdrawals on the probability of intrusion in the Changjiang River estuary. They conclude that these projects will increase the probability of saline intrusion and suggest that water management should be adapted to mitigate the risk.

This estuary has been extensively studied in terms of chemical contamination (e.g. Grimalt, 1999; Gomez-Parras et al., 2000; Riba et al., 2002) and fisheries (e.g. Baldó and Drake, 2002; Fernández-Delgado et al., 2007; González-Ortegón et al., 2012).

Other studies have focused on biogeochemical aspects (e.g. Navarro et al., 2011; Ruiz et al., 2013, 2015, 2017; Huertas et al., 2018) and microplastic transport (Bermudez et al., 2021). However, only a few studies have focused on the hydrodynamic aspects (Álvarez et al., 2001; Diez-Minguito et al., 2012, 2014; or Losada et al., 2017). According to Siles-Ajamil et al. (2019), using a linearized multichannel analysis model, an increase in the mean channel depth leads to both increased tidal height and salt intrusion. In a more recent study, Sirviente et al. (2023) used numerical simulations to show a significant current

amplification of the M2 tidal wave at the head of the estuary. This amplification exceeded the amplitude of the wave at the mouth of the estuary due to channel deepening. The authors also showed altered tidal current dynamics characterized by increased intensity at the mouth of the estuary, potentially leading to greater saline intrusion.

The vertical salinity structure in the Guadalquivir estuary is characterized by intense mixing that prevents the formation of significant gradients in salinity and temperature, resulting in a homogeneous distribution of water properties (García-Luque et

al., 2003; Diez-Minguito et al., 2012). The low average flow of the river, combined with the high tidal prism resulting from the wide tidal range and shallow channel depth, contributes to the Guadalquivir estuary being a well-mixed estuary with very low vertical gradients in salinity and temperature (García-Lafuente et al., 2012). This type of well-mixed estuary is characterized by a uniform distribution of salinity, facilitated by strong tidal currents that prevent stratification. Similarly, the vertical circulation shows a relatively uniform current pattern during low-flow periods, with no significant changes in velocity

or flow direction at different depths (Losada et al., 2017). The hydrodynamic model presented by Sirviente et al. (2023), validated with different observations recorded both at the surface and at depth, shows that there are no significant variations in velocities at different depths; therefore, the presence of significant stratification is unlikely, which favors the homogeneous distribution of salinity in the water column.

Salinity is the main environmental factor determining the spatial distribution of species in estuaries (Marshall and Elliott, 1998). Therefore, it is essential to have a clear understanding of the behaviour of this variable. In the GRE, salinity presents a clear horizontal gradient that decreases upstream (Gonzalez-Ortegon et al., 2014). Regarding salinity structure, the decrease in freshwater flows in the GRE has triggered gradual salinization of the system. Based on data from a real-time monitoring network (Navarro et al., 2011), Díez-Minguito et al. (2013) found that salinity distribution is controlled by nontidal transport, Stokes transport, and tidal pumping. These authors suggested that the effective longitudinal dispersion coefficient is dependent on the length of the system, with a higher mean value upstream. They also analyzed the estuarine recovery response to strong discharge conditions (usually short but intense regimes), proposing that the displacement of saline intrusion towards the mouth causes an increase in stratification. The authors proposed a net propagation speed of 4 cm s$^{-1}$ for the saline intrusion within the estuary.

Reyes-Merlo et al. (2013) assessed the relative influence of climatic forcings (freshwater discharge, tidal currents, and wind) on saltwater intrusions in the GRE using Markov chain Monte Carlo simulations. They argued that under low-flow regimes, saline intrusion depends more on mean tidal fluctuations than on river discharge. They also proposed tidal pumping and baroclinic circulation as the drivers of salt transport. Their results indicate that the mean length of salt intrusion would increase by approximately 8% under the expected scenario of a 15% decrease in freshwater discharge over the next 15 years.

Using a linear analytical model, Siles-Ajamil et al. (2019) explored the effects of management alternatives on salinity distribution, with channel deepening and marsh recovery potentially affecting tidal wave propagation and salinity. The authors argue that the mean salinity distribution shifts upstream, increasing the maximum salinity gradient, saline intrusion, and M2 tidal salinity generated by the advection of the mean salinity gradient. These studies collectively highlight the dynamic nature of salinity in the Guadalquivir Estuary, influenced by both natural and human factors. However, there is a lack of information on the temporal and spatial variability of the saline intrusion in the GRE, as well as the impact of anthropogenic pressures.

Biemond et al. (2022) investigate the impact of freshwater pulses (brief periods of high river flow) on estuarine saline intrusion in GRE. They constructed an idealized three-dimensional nonlinear model based on the model of MacCready (2007), but not based on Pritchard equilibrium. The model features constant width and depth, as well as constant viscosity and diffusivity coefficients in space and time. The new model demonstrates that the intensity and duration of the pulse are the primary factors controlling the reduction of salt intrusion, with tidal force having a relatively minor influence. Additionally, the time required for saline intrusion to return to its initial position is found to be dependent on the river discharge following the pulse, rather than the distance the intrusion has traveled upstream.

Constant dredging (Gallego and García-Novo, 2006; Donázar-Aramendía et al., 2018) and cultivation fields over the last decade have caused changes in the behavior of the saline intrusion. This, coupled with the limited contribution of freshwater from the dam, leads to high salinity levels in the upper reaches of the river.

According to Algaba et al. (2024), the primary water abstractions in the Guadalquivir River basin are irrigation (88%), domestic uses (10%), industrial activities (1.1%), and energy production (0.9%). This indicates that the Guadalquivir basin is subject to

constant human pressure, which modifies the ecological flow rate of the system. Therefore, it is necessary to characterize the actual state of the saline intrusion and the effects of anthropogenic pressure on the GRE.

Obtaining in situ measurements over time and space is a challenging task that is often not feasible. Therefore, numerical simulations capable of replicating the current state of estuarine salinity intrusion are essential for understanding the system and developing management strategies for its preservation.

This manuscript provides an analysis of the current state of saline intrusion in the GRE and evaluates the impact of anthropogenic pressures on the behavior of the saline intrusion. This study represents the first attempt to analyze the impact of anthropogenic water withdrawals on estuary hydrodynamics and the associated saline intrusion. To achieve this, we used a realistic 1D model which includes a hydrodynamic module (Sirviente et al., 2023) and an advection-dispersion module. This model was calibrated and validated using observations gathered during multiple oceanographic campaigns conducted from 2021 to 2023. Measurements were taken continuously during both upstream and downstream voyages of the ship, with sampling points along the river monitored over different time coverage (hours to days). We conducted various numerical experiments, designing scenarios with different freshwater inputs and degrees of water withdrawals, to perform a comprehensive simulation-based analysis.

## 2.Methodology

The analysis is based on a 1D hydrodynamic model capable of realistically simulating tidal-fluvial dynamics (sea surface elevation and tidal current velocity), coupled with an advection and dispersion transport model to simulate salinity concentration along the GRE. The hydrodynamic module has been previously calibrated and validated with a high reliability (Sirviente et al., 2023). This manuscript presents an analysis of salinity behavior along the GRE. In this context, in situ observations collected from several short-duration oceanographic campaigns conducted during the dry seasons from 2021 to 2023 will be employed for the calibration and validation of the advection and dispersion module. Once calibrated and validated, the model is used to analyze saline intrusion behavior under different freshwater flow and anthropogenic pressure scenarios (section 3). Anthropogenic pressures are defined as water volume reductions (water withdrawals) caused by diverse activities, such as water use in adjacent crop fields, water losses due to flow modifications in estuary channels or avenues or illegal wells among others.

### 2.1. Data source

During the years 2021, 2022 and 2023, several oceanographic campaigns were conducted along the GRE during dry months (low or negligible rainfall and reduced freshwater flow). These short-term campaigns provided observations at different times along the estuary (Table 1). Measurements were obtained from a conductivity sensor installed on the oceanographic vessel UCADIZ, which recorded data every minute as the ship traveled upstream the GRE (MG1, from 5 km to 73 km from the mouth; MG2, from 5 km to 73 km from the mouth; MG3, from 5 km to 60 m from the mouth; MG4, from 5 km to 80 km

from the mouth) and downstream the GRE (MG3, from km 10 to km 58; MG4, from km 7 to km 65) . This validation strategy, which will be further justified in section 3.1, allowed for simultaneous validation of the simulation outputs in both space and time. The observations corresponding to each vessel trip were taken at different points and time instances as the vessel ascended or descended along the GRE. Once the simulations corresponding to each campaign for validation are obtained, the same time instances as each observed data point are used. Additionally, model points are used as close as possible to the observed points, with small differences between them (Fig SM1). This ensures that the same points and time instances as the observed data are compared (tidal phase lag are contemplated in our simulations). Additionally, short-term observations collected from fixed stations (Fig. 1a) were analyzed to validate the time variability of the simulations throughout the tidal cycles at various points along the river. Specific sample points, dates, and time lengths of the observations are detailed in Table 1.

**Table 1: Information about the salinity data (psu) records collected along the Guadalquivir Estuary during different oceanographic campaigns. The sample point locations are shown in Fig. 1a. The temporal resolution of each measurement is indicated as Δt. "Up" and "down" of vessel trips refer to upstream (from Xo to the dam) and downstream (from the dam to Xo) of the GRE.**

| Campaign | Station | Date | Time-interval | Δt |
|---|---|---|---|---|
| **MG1** | Vessel trip (up) | 20/09/2021 | 5h | Minute |
| | MG1 P1 | 21/09/2021 | 9h | Hour |
| | MG1 P2 | 23/09/2021 | 10h | Hour |
| | MG1 P3 | 24/09/2021 | 10h | Hour |
| **MG2** | Vessel trip (up) | 31/01/2022 | 6h | Minute |
| | MG2 P1 | 01/02/2022 | 10h | Hour |
| | MG2 P2 | 02/02/2022 | 10h | Hour |
| | MG2 P3 | 03/02/2022 | 10h | Hour |
| | MG2 P4 | 04/02/2022 | 10h | Hour |
| | MG2 P5 | 05/02/2022 | 9h | Hour |
| **MG3** | Vessel trip (up) | 19/07/2022 | 6h | Minute |
| | Vessel trip (down) | 20/07/2022-21/07/2022 | 4h | Minute |
| | MG3 P1 | 19/07/202-20/07/2022 | 25h | Hour |
| | MG3 P2 | 20-07/2022-21/07/2022 | 25h | Hour |
| | MG3 P3 | 21/07/2022-22/07/2022 | 11h | Hour |
| **MG4** | Vessel trip (up) | 17/10/2023 | 8h | Minute |
| | Vessel trip (down) | 18/10/2023 | 4h | Minute |

CTD profiles collected from the same sampling stations as the thermosalinograph data (Table 1) for campaigns MG1, MG2, and MG3, as depicted in Figure 1a, are also utilized. These profiles facilitate the analysis of the vertical characteristics of the water column.

## 2.2. Numerical hydrodynamic module description

The hydrodynamic module used in this study was described, calibrated, and validated by Sirviente et al. (2023). This 1D module integrates a system of long-wave equations along an elongated channel (Godin and Martinez, 1994)

$$\frac{\partial u}{\partial t} + u\frac{\partial u}{\partial x} = -g\frac{\partial \eta}{\partial x} - \frac{k|u|u}{(h+\eta)} \quad , \tag{1}$$

$$\frac{A}{h}\frac{\partial \eta}{\partial t} = -\frac{\partial}{\partial x}[Au] \quad . \tag{2}$$

The first equation represents the balance of along-channel forces per unit mass, while the second ensures volume conservation. Here, $u$ represents along-channel current velocity (m s$^{-1}$), $t$ denotes time, $x$ denotes the along-channel position (m), $k$ is the bottom friction coefficient ($k = 0.003$), $\eta$ is sea level elevation above mean sea level (m), $h$ (m) denotes average bottom depth across the channel below mean sea level, $b$ represents the across-channel width ($m$), and $A = (h + \eta)b$ is the instantaneous cross-sectional area across the channel. The spatio-temporal distributions of $\eta$ and u variables were derived by the numerical integration of these equations using an explicit finite differences scheme (leapfrog type) (Dronkers, 1969).

The GRE is assimilated to a 110 km channel extending from its mouth ($x = 0$) to the head ($x = L$). For the numerical integration of the independent variables ($x, t$), the values $\Delta x = 25$ m and $\Delta t = 1$ s are set, yielding a Courant-Friedrichs-Lewy coefficient of 0.39, which ensures numerical stability.

The channel cross-sections are approximated by rectangles, where the sides are defined by the across-channel averaged bottom depth $h(x)$ and channel width $b(x)$. Sea level predictions, based on harmonic constants derived from the harmonic analysis of sea level during 2019 at Bonanza station (data provided by *Puertos del Estado*, PdE), were used to force the simulation at the open boundary near Bonanza (Fig. 1a). A no-flow boundary condition normal to the section of the *Alcalá del Río* dam was also prescribed. The module considers the contributions from three tributary flows, A33 (*Rivera de Huelva*), H09 (*Alcalá del Río*), and A50 (*Zufre*), whose flow rates are not negligible. Hourly data of the tributary flows, covering the 7 days of simulation for each campaign (sections 2.1 and 2.2) and 15 days (for section 2.3) were provided by the Guadalquivir Hydrological Confederation (Guadalquivir SAIH, https://www.ch guadalquivir.es/saih/, last access: 25 March 2024) and used as freshwater flow input.

To account for water volume withdrawals from the estuary, a parameter called "sink" (denoted by $\delta$) was introduced. It represents the thickness (m) of a water slide of horizontal area equal to $b \cdot \Delta x$, the horizontal area contained between each pair of transversal sections. This parameter is subtracted at each integration time step $\Delta t$ from the previously computed $\eta$ value. This is equivalent to withdrawing a water volume $b \cdot \Delta x \cdot \delta$ at each integration time step $\Delta t$. The suitable value of $\delta$ for each pair of transversal sections is determined together with the validation of the advection and dispersion model, as explained in section 3.1.

The bathymetry used is the most recent available, obtained from the 2019 nautical chart provided by the Hydrographic Institute of the Spanish Navy (*Instituto Hidrográfico de la Marina*, IHM). This information was used to determine the average depth

and width of the transverse channel at each of the equally spaced transverse sections of the GRE, with a distance of $\Delta x = 25$m between sections.

## 2.3. Numerical Advection and Dispersion Module Description

A one-dimensional advection and dispersion module was developed to simulate the evolution of salt concentration within the estuary. To discretize the advection and diffusion terms with respect to space and time, a numerical integration of the across-channel-averaged 1D transport equation:

$$\frac{\partial}{\partial t}(AS) + \frac{\partial}{\partial x}(AuS) = D\frac{\partial}{\partial x}\left[\frac{\partial}{\partial x}(AS)\right] \tag{3}$$

where $S$ is the salinity concentration (psu); $t$ is the time coordinate; $x$ is the along-channel coordinate (m); $u$ is the along-channel current velocity (m s$^{-1}$), $D$ is the turbulent diffusion coefficient (m$^2$s$^{-1}$); $A = (h + \eta)b$ is the instantaneous across-channel section, where $\eta$ is the sea level elevation above the mean sea level (m), $h$ (m) is the across-channel averaged bottom depth below the mean sea level, and $b$ is the cross channel width.

The numerical scheme incorporates the Multidimensional Positive Definite Advection Transport Algorithm (MPDATA) (Smolarkiewicz and Margolin, 1998), which is a type of finite-difference discretization of the advective term to prevent excessive numerical diffusion in the model. The technical characteristics of the algorithm make it suitable for problems involving complex geometries and inhomogeneous flows (Smolarkiewicz and Szmelter, 2005).

The sea level elevation ($\eta$) and current velocity ($u$) were simulated by the hydrodynamic module and implemented as inputs in the transport module, thus developing a 1D model for GRE.A free transmission condition ($\frac{\partial^2 S}{\partial x^2} = 0$) at the open boundary and a reflection condition ($\frac{\partial S}{\partial x} = 0$) at the channel edge were imposed as boundary conditions.

As an initial condition for the numerical integration, a salinity profile following a logistic function was established. This was unique for each simulation according to the characteristics of the observations, starting all of them from a value of 36 psu at the mouth and becoming 0, from 85 km until 110 km from the mouth. Likewise, the module was forced in a section near Bonanza (Fig. 1a), specifying an appropriate temporal variation of the salinity calculated as a function of the current (u) in this section and derived from the available observations.

The horizontal dispersion coefficient was calculated using the equation proposed by Bowden (1983). For sufficiently long diffusion times, and effective coefficient of longitudinal dispersion K$_{xe}$ may be defined for a simple case as: $\boldsymbol{K_{xe} = (U^2\ H^2)}$ $/\boldsymbol{(30k_z)}$, where $U$ is the current intensity, $H$ corresponds to mean depth, and K$_z$ is the coefficient of vertical eddy diffusion (assumed to be uniform throughout the depth).

This calculation was carried out for all the campaigns considered in the analysis to ensure the use of a constant dispersion appropriate to the system. The results indicate that the maximum constant dispersion, based on the intensity of the current speed and depth, is 150 m$^2$ s$^{-1}$. Given the lack of comprehensive observational data on the coefficient's variability across the estuary, it was determined that a constant value would be an adequate representation of the general conditions. This selection

of a constant dispersion coefficient is based on the assumption that lateral dispersion is homogeneous and that strong tidal currents will induce vertical mixing, thereby rendering advection the dominant process in the behavior of the saline intrusion. The Peclet number (Pe), defined as $uL/D$, measures the relative contribution between the nonlinear advection and horizontal dispersion, where u is an averaged (in time and along the whole estuary) absolute value of the along-channel gradient of velocity, $L$ is the estuary length, and D corresponds to the horizontal dispersion coefficient (Deng et al., 2024). Taking a value

u= 0.5 ms$^{-1}$, extracted from realistic simulations performed with the hydrodynamic module and the values L= 107 km and D=150 m$^2$ s$^{-1}$ yields a value Pe=356, clearly indicating a dominance of the advective transport rate over the diffusive one. Numerous studies in the literature have demonstrated that models with a constant dispersion coefficient are capable of accurately reproducing salinity distributions (e.g., Lewis and Uncles, 2003; Brockway et al., 2006; Gay and O'Donnell, 2007, 2009; Xu et al., 2019; Siles-Ajamil et al., 2019; Biemond et al., 2024). This choice not only maintains the stability of the

model, avoiding numerical instabilities, but also ensures that the results are consistent with theoretical expectations and experimental observations. Finally, model validation with observational data has demonstrated that employing this constant coefficient is an effective method for accurately reproducing the essential characteristics of the system, thereby supporting this approach within the context of the present study (Table 2).

In addition, a sensitivity analysis was carried out with different dispersion coefficients to optimize this parameter as much as

possible. In our 1D model, which simplifies the equations governing the balance forces, volume conservation, and advection-dispersion processes, a dispersion coefficient exceeding 200 m² s$^{-1}$ leads to numerical instabilities.

As previously mentioned, logistic approximation is used as the initial condition. This approximation varies for each simulation due to the differences observed in the records (different months and years).

During the calibration process it became necessary to include in the simulations the processes responsible for reducing the

volume of water (sinks) in the channel. Although these sinks have similar values across campaigns, they are unique to each campaign and serve as the basis for the initial condition, which represents the effect of the sinks over a given period of time. The rationale for including this condition is explained in detail in Section 3.1.

## 3.Results and discussion

In this section, the main results obtained with the numerical model are presented, focusing on evaluating the dynamics of the

horizontal salinity gradient in the GRE. As previously mentioned, the hydrodynamic module validation can be found in Sirviente et al. (2023). Initially, the outcomes from the experimental validation are presented. Subsequently, numerical experiments were conducted to assess the impact of saline intrusion resulting from changes in freshwater inflow and in water volume reductions along the GRE.

The salinity profiles obtained from the CTDs show a strong vertical mixing of the water column throughout the whole period

and at all points (see Figures SM2, SM3 and SM4). Very reduced vertical salinity gradients can be observed, which allows us

to conclude that the vertical behavior of the water column in the GRE is homogeneous under the conditions of these campaigns, allowing the use of a 1D model to simulate the salinity concentration along the river.

### 3.1. Effect of Horizontal Dispersion and Water Withdrawal on the Horizontal Salinity Gradient in the Estuary

First, the model is used to analyze the effect of horizontal dispersion on the development of the horizontal salinity gradient. To this end, experiments were conducted considering only the effect of horizontal dispersion, representing the natural behavior of the system without any additional intervention. Subsequently, experiments incorporating the presence of sinks were carried out, simulating a reduction in the water volume of the channel.

Figure 2 shows the observed horizontal salinity gradient during the MG3 campaign (upstream and downstream trips) along with simulations corresponding for a 13-day period. This figure compares the simulated horizontal gradients under different conditions: Figures 2a and 2b consider only the effect of horizontal dispersion; Figures 2c and 2d include both horizontal dispersion and a uniform reduction in water volume ($\delta$), constant in time and space; and finally, Figures 2e and 2f incorporate horizontal dispersion along with time-varying sinks ($\delta$). It is important to emphasize that all times shown in Figure 2 correspond to moments when tidal behavior matches the observations at each location during the reference period. The simulations shown in the figure represent salinity concentrations generated by the model at each observation time, corresponding to the nearest possible point to the vessel's sampling location.

When analyzing the behavior of the horizontal salinity gradient along the estuary, considering only horizontal dispersion, it becomes clear that the system cannot reproduce the observed salinity concentrations from the different campaigns. Even when the dispersion coefficient is increased to 190 m² s$^{-1}$ (Figures 2a and 2b) – the highest dispersion coefficient that can be used with this model without causing numerical instability – the observed concentrations do not match.

Considering only the effect of horizontal dispersion, it can be observed that the system tends to slightly increase the salinity concentration over time in almost all sections of the river up to 45 km from the estuary. Beyond this point, the behavior becomes almost linear, with very low salinities close to 0 psu. These results show that if only the effect of horizontal dispersion is considered, the system is unable to reproduce the observed salinity range.

This highlights the need to include processes in the simulations that significantly enhance the horizontal salinity gradient. These processes may include those capable of reducing water volume. Considering only natural effects, such as evaporation or the small natural channels present in the estuary, would not generate a sufficient volume reduction to account for the high salinity range observed along the river during the different campaigns. Therefore, it is necessary to consider anthropogenic processes such as water withdrawal for agricultural, industrial and urban uses, illegal wells, the creation of secondary channels and marshland reductions, among others. Together, these natural and anthropogenic processes lead to a reduction in water volume, which could be responsible for the high salinity concentrations observed in the inner part of the estuary.

The parameter $\delta$ is a key factor in quantifying the effect of extractions and minor contributions to water volume along the river. It encompasses all the natural and anthropogenic processes that influence water volume, such as agricultural abstraction, industrial use, small side channels and evaporation. $\delta$ represents an average value between these two actions, and for our study

it was positive, indicating that on average extractions exceed the contributions from smaller channels that drain into the main
channel.

However, there is an inherent uncertainty in estimating $\delta$ due to the complexity of accurately quantifying the amount of water withdrawn from the channel. The Guadalquivir system is heavily influenced by human activities (high levels of agriculture, industry, dense population in nearby areas, port operations, etc.). In addition, numerous illegal water withdrawals have been documented.  This makes it difficult to obtain accurate data on water withdrawals within the estuary, as both the specific
locations and volumes of water withdrawn are unknown.

When a constant water volume reduction term ($\delta$ = 0.005 mm) is applied uniformly in time and space (Figures 2c and 2d), representing a consistent volume removal across the estuary over 13 days of simulation (dt = 1s and dx = 25m), the system produces higher salinity concentrations over time, closely matching the observed salinity range. This demonstrates the importance of including $\delta$ in the numerical model to simulate the high salinity concentrations observed along the GRE
accurately. The amount of water removed in this experiment during the 13 days of simulation was 47.36 $10^6$ m$^3$, which is not excessive when compared to the amount of water needed to irrigate, for example, a rice field of 32000 hectares (the actual extension of the involved rice field), which requires 384.0 $10^6$ m$^3$ of water consumption.

Figures 2c and 2d show that a certain period of time is required for the sinks to effectively influence the system and produce salinity values close to those observed. Figures 2e and 2f show experiments where a stronger sink ($\delta$ = 0.01 mm) is applied to
all sections during the first 3 days of the simulation, after which it is relaxed, and a $\delta$ = 0.001 mm is imposed for the remaining 10 days. These simulations closely replicate the observed horizontal salinity gradient, highlighting the importance of incorporating the temporal variation of sinks to achieve realistic salinity simulations.

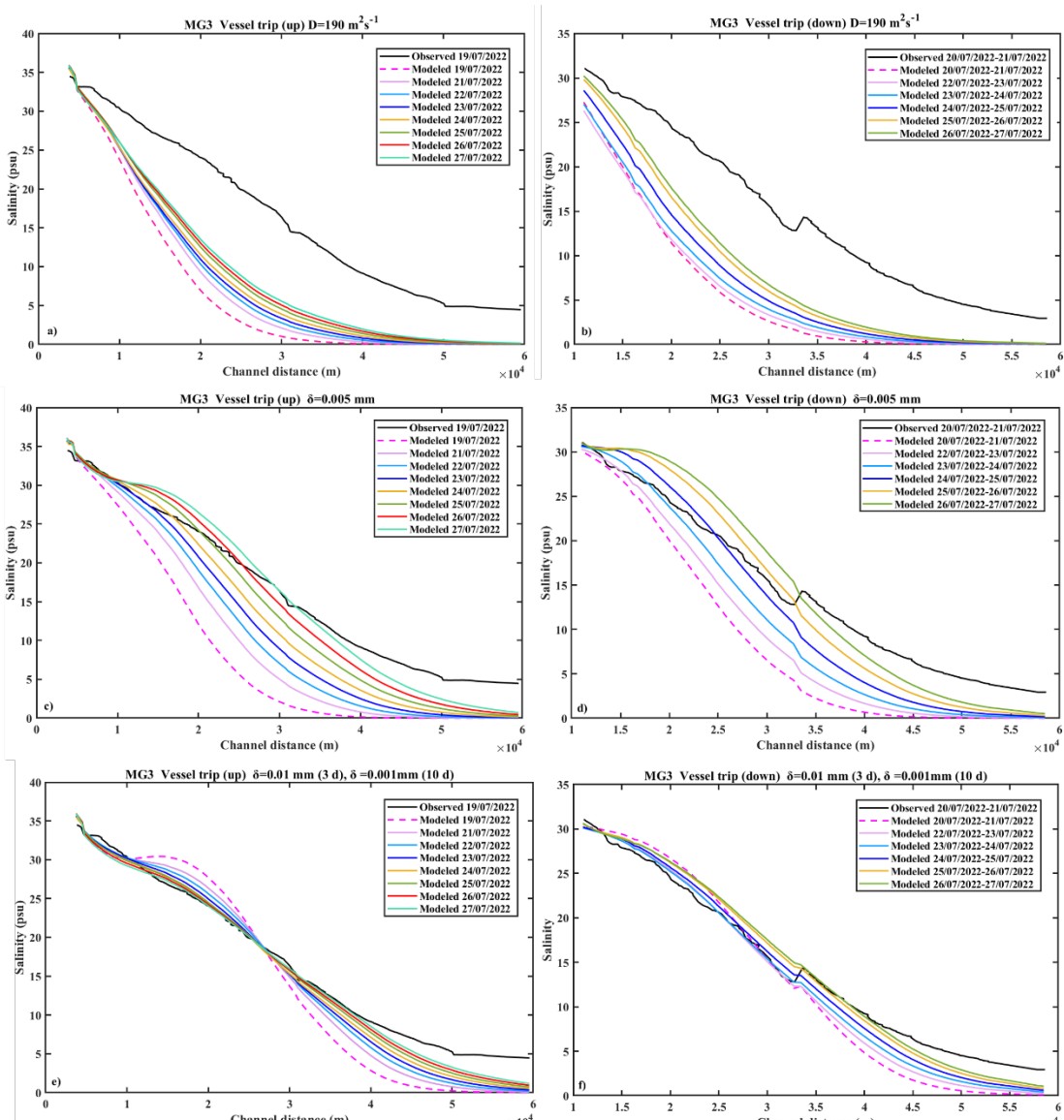

**Figure 2: Comparison of observed and simulated salinity during the MG3 vessel trips. (a) and (b) show simulations including only horizontal dispersion for upstream (a) and downstream (b) vessel trips, with observations in black. (c) and (d) show simulations incorporating δ term (δ) for the entire simulation period as constant value, and (e) and (f) are the simulation including a time-varying δ Observations are shown in black for upstream (c and e) and downstream (d and f) vessel trips.**

These experiments underscore the necessity of including key parameters in simulations, as the horizontal salinity gradient would otherwise not reach observed values. As shown in Figure 2, a certain duration of sink activity is required for the simulated salinity concentrations to approach the observed range. Therefore, it is essential to define an initial condition that considers this progression, allowing the simulation to adequately capture the evolution of water withdrawals and their effects

on salinity over time. This approach is particularly supported by Figures 2e and 2f, where applying a stronger δ initially, followed by a weaker δ, successfully reproduces the observed behavior.

To define these initial conditions effectively, it is essential to consider that observations from different oceanographic campaigns were collected at varying times (months and years). This implies that each simulation, corresponding to a specific time period, will have a slightly different initial condition due to the variations in the characteristics of the system over time. This approach emphasizes the importance of adjusting the initial conditions according to the temporal differences observed in the data, allowing for a more accurate representation of the system's behavior during each period considered.

It is important to emphasize that this numerical model, although simple, has been designed as a very useful tool for studying the hydrodynamic and physicochemical properties of the estuary. As a high-resolution 1D model, it is optimized to simulate relatively short time periods, ensuring high computational efficiency. The model is particularly effective at representing specific moments in time and extrapolating to a given time interval. Although it is designed to simulate shorter time periods, it can be used for longer simulations provided that similar conditions are maintained, such as low discharge regimes where the water column is well mixed and vertical gradients are homogeneous. However, in situations with significant stratification, alternative approaches would need to be considered as 1D simulations would not be suitable to accurately simulate the estuary.

### 3.2. Experimental validation of the salt transport and dispersion model

During the model validation phase, it was found that the extension of the saline intrusion into the interior of the estuary coming from the data measurements was much greater than those simulated using only the flow data provided by official sources. Hence, we became aware that significant undocumented water withdrawals were occurring during the different campaigns (as it has been described in section 3.1).

Considering that the intensity, spatial location, and temporal variability of these withdrawals are unknown, the numerical models had to undergo an ad hoc experimental validation for each campaign (MG1, MG2, MG3, and MG4). As mentioned before, for each numerical integration, an initial salt concentration field was defined using a logistic function that was determined by the behavior of the observations. The procedure begins by establishing a $\delta$ value in the hydrodynamic model, running the model, and later using the resulting $u$ and $\eta$ outputs in the advection and dispersion model to fit the salinity observations. The value of $\delta$ was determined empirically through sensitivity analysis until the best-fitting simulation was obtained.

The experimental validation determined that the best fitting of the simulated salinity values to the observations are those presented in the following lines. In MG1, a constant sink δ = 0.0015 mm was implemented in all sections and time steps. For MG2, δ = 0.0005 mm was applied uniformly from 0 km to 22 km and from 42 km to 85 km. From 22 km to 42 km it was increased to δ =0.0045 mm. In MG3 and MG4, a uniform sink of δ =0.00225 mm and δ =0.0012 mm was employed throughout all the sections, respectively. The slightly higher sinks between 22 km to 42 km for MG2 can be justified by the location of crop fields (Fig. 1a) and secondary channels.

Fig. 3 shows the behavior of the longitudinal observations collected demonstrating the high accuracy of the simulations
(including $\delta$) in replicating the observed data and demonstrating a robust fit across all campaigns. It is important to note that
the simulations presented in this figure represent the simulated salinity concentration generated by the model at each time
instance of the observations, corresponding to the nearest possible point to the sampling location of the vessel (Fig. SM1). We
analyze the discrepancies between observations and simulations (including sinks). For MG1 (Fig. 3a), there are some
discrepancies in the first 6 km, where the model overestimated the salinity concentration, and between 10–18 km and 30–50
km, where it slightly underestimates observed salinity. In MG2 (Fig. 3b), the model shows a slight overestimation within the
first 22 km, followed by a slight underestimation between 22 and 42 km. Regarding MG3, the observations collected on July
19, 2022, indicate a minor underestimation across all kilometers of the GRE (Fig. 3c). In contrast, the model indicates a slight
overestimation of salinity between 45 and 60 km for the simulation corresponding to the vessel's downward voyage (Fig. 3d).
For the MG4 campaign, the simulation for 17/10/2023 (Fig. 3e) shows a very slight underestimation of salinity concentrations,
while for 18/10/2023 (Fig. 3f), the model slightly overestimates the observations of 19/10/2023. The differences between
simulations (including $\delta$) and observations near the mouth of the estuary can be explained by their proximity to the model
boundary conditions, which generates a disturbance that can add noise in the first kilometers of the simulation. The
discrepancies found between 30 km and 60 km can be attributed to the crop fields adjacent to the channel (it is the area where
more crop fields are located on both sides of the GRE). The October 2023 observations show particularly notable results.
While all analyzed observations correspond to low-flow regimes ($\overline{Q}_{MG1}$ = 8 m³s⁻¹, $\overline{Q}_{MG2}$ = 12 m³ s⁻¹, $\overline{Q}_{MG3}$ = 8 m³ s⁻¹), the MG4
campaign recorded the lowest flow, with an average of just $\overline{Q}_{MG4}$ =3 m³ s⁻¹. Despite these conditions, salinity levels in the upper
estuary were remarkably high, reaching values of 5 psu at 70 km from the mouth. These high concentrations are anomalous
and could be caused by eventual stronger water withdrawals further along the river.

The black lines in Figure 3 represent the simulation without considering the inclusion of the sinks. As can be observed, they
show a much lower horizontal salinity gradient compared to the one observed in all the campaigns.

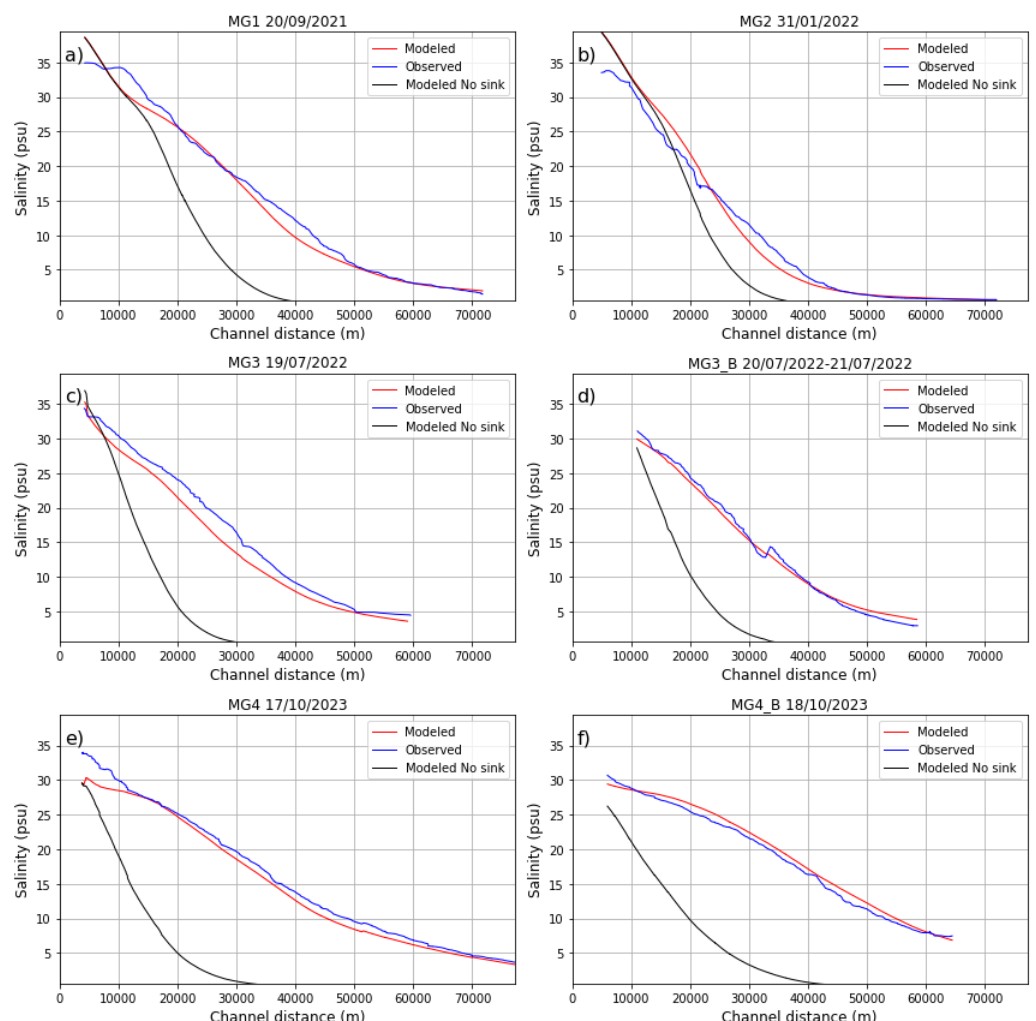

**Figure 3: Comparison between the observations (blue line) and the simulation including sinks (red line) and simulations no including sinks (black lines) of the salinity (psu) for the hole channel for different campaigns (Table 1).**

The good agreement between observations and simulations is shown in Table 2. Correlation coefficients higher than 0.95 are obtained for all simulations, along with small root mean square errors (RMSE). The Nash-Sutcliffe Efficiency Coefficient (NSE) is used to evaluate the variability of the observations explained by the simulation. Values close to 1 indicate that the model is able to reproduce the observations. Similarly, the inequality between the series has been assessed as not significant, indicating that there are no significant differences between them (t-student, Table 2). Therefore, it can be said that the transport

model can reproduce the salt concentration along the Guadalquivir with high reliability.

**Table 2: Determination coefficient at 95% of confidence level, root mean square error (RMSE) for salinity (psu), Nash-Sutcliffe efficiency coefficients, and t-student values for each oceanographic campaign.**

| Campaign | $R^2$ ($P_{value}$ 0.0) | RMSE | NSE | P value (t-student) |
|---|---|---|---|---|

| | | | | |
|---|---|---|---|---|
| **MG1** | 0.99 | 1.36 | 0.99 | 0.53 |
| **MG2** | 0.99 | 1.46 | 0.98 | 0.75 |
| **MG3** | 0.99 | 1.86 | 0.96 | 0.08 |
| **MG3_B** | 0.99 | 0.70 | 0.99 | 0.80 |
| **MG4** | 0.98 | 1.34 | 0.98 | 0.19 |
| **MG4_B** | 0.97 | 0.81 | 0.99 | 0.45 |

Based on the analysis of the data from the different stationary sampling stations, it can be inferred that there is a strong
relationship between the two sets of data. This is evident from the high coefficients of determination obtained for the
measurements taken at MG3 P1, MG3 P2 and MG3 P3 (see Fig. 1), which were all greater than 0.80. The RMSE for each
sample was also relatively small (see in Supplementary Material SM5). Furthermore, a similar agreement was obtained for
stations MG2 P1 to MG2 P5 (Fig. 1), with $R^2$ values greater than 0.70 and RMSE errors less than 1 psu. However, for the
sample points of MG1 (Fig. 1), the coefficients of determination were lower ($R^2$ values greater than 0.7 for MG1-P1 and MG1-
P3, but moderate correlation for MG1-P2, $R^2$=0.5), and the RMSE was higher (values less than 2 psu) for all stations (see in
Supplementary Material SM6 and SM7, respectively). At stations MG1 and MG2, the model overestimated the signal, with
smaller differences observed at stations in the upper and middle parts of the estuary and larger differences at stations in the
lower part of the GRE.

It is worth noting that the introduction of sinks into the model provides only a rough approximation of the actual way in which
withdrawals occur on both spatial and temporal scales. These sinks, which are assumed to occur at a constant rate over time,
only partially resolve this uncertainty. The actual volume of water removed is unknown. This uncertainty may be behind this
slight overestimation of the model. Other sources of uncertainty could be the salinity increases caused by drainage from the
marshes to the channel mouth. These waters, enclosed in extremely shallow marshes, may experience salinity increases due to
evaporation before being discharged into the estuary. In addition, water drained from crop fields, such as during rainy periods,
can bring additional concentrations of salt due to the presence of fertilizers carried by the water from the soil of the crop fields.
Despite this, considering the overall consistency of the validations and statistical analyses, we can assert that the model
demonstrates a good level of reliability in reproducing the salinity concentrations along the GRE.

### 3.3. Saline intrusion behaviour in the observation campaigns.

Figs. 4a-d display the salinity at 2 m depth recorded by the thermosalinograph of the vessel while sailing along the estuary in
the center of the channel in September 2021 and January 2022 (Table 1). As illustrated, the saline intrusion extends beyond 30
km in both cases, exceeding 20 psu at 20 km. The highest concentrations are found at the mouth of the estuary, with values
close to 36 psu. This observation confirms the basic premise of our modelling approach, in which we incorporate an initial
condition by imposing a temporal salinity fluctuation calculated as a function of current velocity. This is based on the salinity

benchmark of 36 psu at the open sea mouth. In September 2021 (Fig. 4a), concentrations below 5 psu are obtained from

approximately 50 km onwards, while for January 2022 (Fig. 4d), these concentrations are observed from 40 km onwards.

Model simulations that included water withdrawals (Figs. 4b-e) closely resembled the observed behavior of the saline intrusion. In general, simulations including this factor can effectively replicate the observed salinity. However, when sinks were not considered (Figs. 4c and f), the simulated salinity was significantly lower than the observed values. Note that concentrations of practically 0 psu were obtained from 30 km onwards. This configuration could be regarded as the natural state of the saline

intrusion, unaltered by anthropogenic intervention.

These results remained consistent throughout all the comparisons with experimental data examined (shown in Fig.SM8 and SM9). At this point of the analysis, on the one hand it is clear that including these water withdrawals is necessary to accurately simulate salt transport throughout the GRE. On the other hand, the existence of these sinks reveals the significant impact that the usage of water, such as those demanded by the adjacent crop fields or other domestic needs, generates on the horizontal

salinity gradient. It is important to note that even though the actual consumption and consumers of the estuarine water remain unknown, the model has detected that a clear deficit of water had been produced in the estuary during the analyzed periods. As previously pointed out, a horizontal salinity gradient beyond 30 km from the mouth would not occur if the consumed water (not only by anthropic activities but also by natural processes) had been compensated with an increased flow from the dam. In this sense, the applied analysis methodology could be used to check, using the vessel-based observations and the described

numerical simulations, the degree of alteration of the horizontal salinity gradient at a given time. This can be done by comparing the observed and simulated salinity gradient in the absence of sinks as a reference.

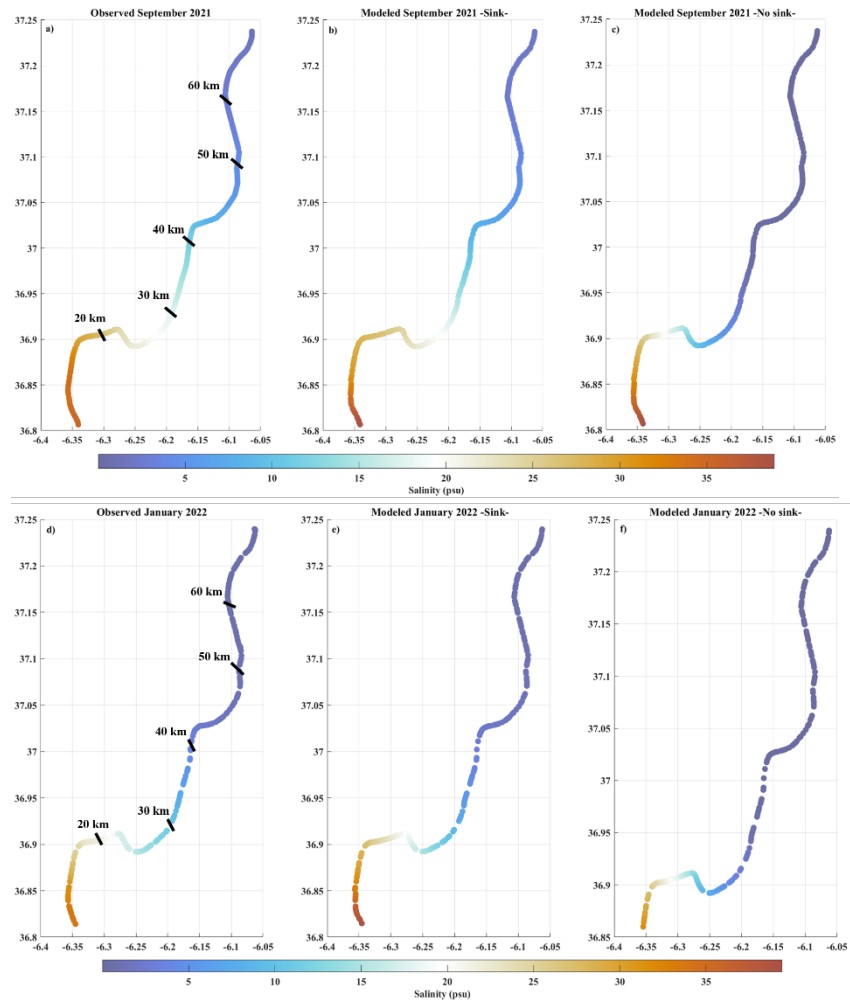

**Figure 4: Salinity concentration maps (psu) of the Guadalquivir River Estuary from the oceanographic campaigns MG1 and MG2 on 20/09/2021 (a, b, c) and on 30/01/2022 (d, e, f), respectively. Observational data are shown in (a) and (d). Simulation including water volume reductions are presented in (b) and (e). (c) and (f) correspond to model simulation without sinks.**

### 3.4. Tidal cycle dynamics.

Once the reliability of the model had been confirmed by the results of the experimental validation presented in the previous sections, it was used to simulate the dynamic of the saline intrusion during a spring-neap tidal cycle. To do this, we conducted a simulation extended over 15 days (15/07/2022-30/07/2022) using the same model configuration presented in section 3.1 for the MG3 campaign. This period was selected because it comprised records of observations distributed throughout the spring-neap tidal cycle, allowing for the validation of the simulations. A spin-up of 3 days is necessary to stabilize the initial conditions and achieve realistic outputs.

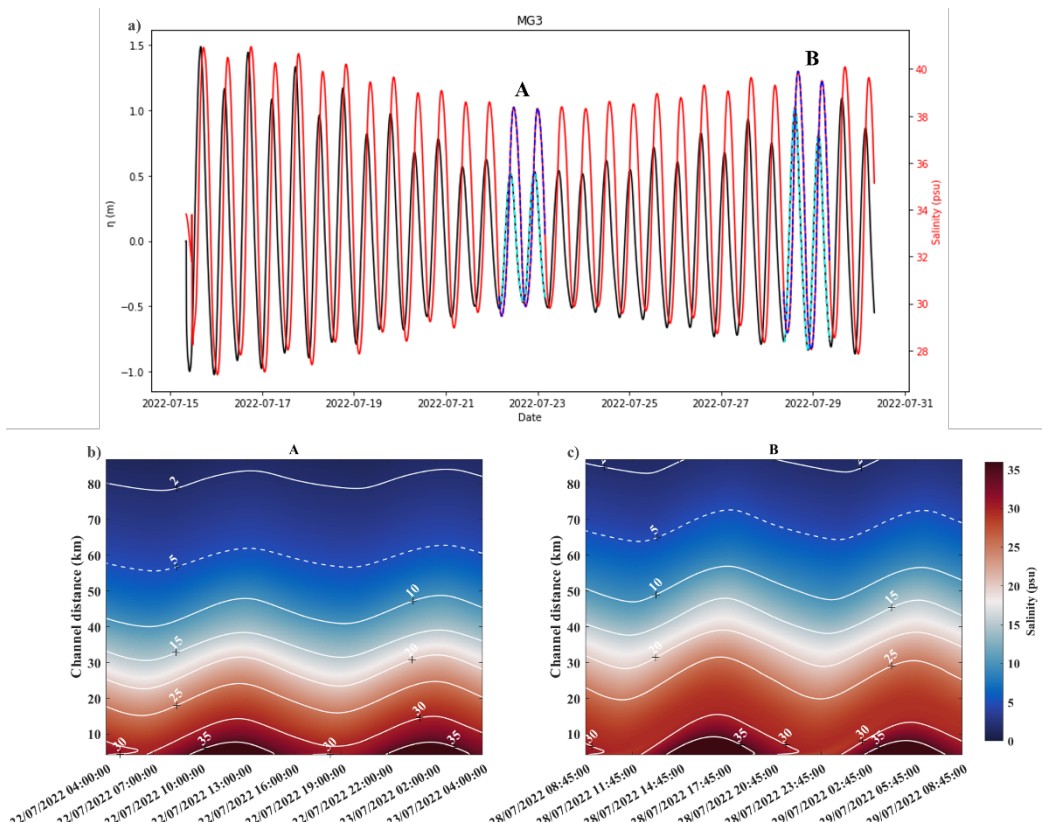

**Figure 5: (a) Superposition of tidal height (m) and salinity (psu) simulated time series at Bonanza section throughout 15 days of July 2022. Dashed lines indicate the selected 24 h periods referred to in (b) and (c); Hovmöller diagrams of simulated salinity variation over these two daily cycles (24 h) during neap tides(b) and spring tides (c) along the Guadalquivir estuary. Isohalines are represented as white lines.**

We focused on two 24-hour periods to describe the dynamics of the horizontal salinity gradient during different phases of the semi-diurnal cycle (Fig. 5a). A lag of about 1.5 h was observed between tidal height and salinity profiles (Fig. 5a), which means that the maximum and minimum salinity concentration values coincide with moments just before high and low tide, respectively. Fig. 5b and d, shows a gradual decrease in salinity values upstream.

In Figure 5a, it can be observed that the maximum salinity levels occur near the high-water slack, while the minimum salinity levels are recorded around the low water slack. Figure 5b shows the progression of the saline intrusion during neap tides (A). Using the 5 psu isohaline as the boundary for the horizontal salinity gradient, it can be seen that the maximum salinity extends up to 63 km from the mouth, while the minimum values of this isohaline do not exceed 56 km. In contrast, during spring tides (B) (Figure 5c), as expected, higher salinity values are observed throughout all sections of the estuary compared to neap tides. The 5 psu isohaline extends up to 72 km from the mouth, while the minimum values do not exceed 65 km. This shows a difference of approximately 5-8 km between the moments of maximum and minimum intrusion, being this displacement higher for spring tides than neap tides.

In the same way, when comparing the behavior during spring tides to neap tides, we can observe a difference of 8 km between the minimum values and up to 10 km between the maximum values. Therefore, there is an oscillation of approximately 10 km between spring and neap tides. During spring tides, the horizontal salinity gradient reaches higher concentrations further upstream compared to neap tides, where both the maximum and minimum salinity values are lower. This finding is consistent with the results suggested by Díez-Minguito et al. (2013), who documented a net displacement of approximately 10 km between spring and neap tides.

These results suggest that the constant anthropogenic pressure on the estuary has caused a change in saline intrusion, resulting in higher salinity levels upstream of the river compared to the records of previous studies, such as that of Fernández-Delgado et al. (2007). In this study, it was found that over a six-year period (1997–2003), the 5 psu isohaline boundary was located near 25 km at low tide and at 35 km at high tide. The 18 psu isohaline limit was also found to be 5 km and 15 km upstream of the river mouth at low and high tides, respectively.

### 3.5. Discussion of the main anthropogenic pressures driving the horizontal salinity gradient.

The natural flow regime of the Guadalquivir estuary has undergone significant changes due to different human activities in the basin (Bramato et al., 2010 Lee et al., 2024). Future projections indicate a reduction in freshwater flow for this estuary by the end of the 21st century (Lee et al., 2024). Agricultural activities, land demand, and the channelization of the estuary for navigation purposes have collectively contributed to significant changes in its geomorphology (Ruiz et al., 2015). The construction of the *Alcalá del Río* dam has affected tidal propagation, and multiple water uses - including agriculture, human needs, and industry- have resulted in a 60% reduction of freshwater inflows (Contreras and Polo, 2010).

The results from previous numerical simulations clearly suggest that human activities (and also natural processes) developed in the estuary, along with the decrease in the freshwater inputs, are behind the excessive saline intrusion. It is not possible to simulate the behavior of the horizontal salinity gradient along the river without adding these effects. As previously mentioned, river discharges into estuaries are regulated mainly for economic reasons: power generation, irrigation, and freshwater supply to populations located near the basin (Vieira and Bordalo, 2000; Jassby et al., 2002). This freshwater use leads to a decrease in estuarine water volume and thus to increased saline intrusion. In recent years, freshwater discharges to the estuary have been reduced by more than 50% on average, and during dry periods, freshwater input is almost completely controlled by the *Alcalá* dam (Baldo et al., 2005; Fernandez-Delgado et al., 2007).

The model used in the present study, which accurately reproduces the current salinity along the river, has shown that without the introduction of water withdrawals, the system would not achieve the salinity presented in the observations. This fact highlights the need to understand the effects that changes in the amount of these water withdrawals, along with changes in freshwater flow, could exert on the horizontal salinity gradient behavior. For this purpose, different numerical experiments were designed for various freshwater flow and water sinks. The resulting simulations will be compared with those obtained in

the two 2022 campaigns, MG2 and MG3 under the real freshwater flow presented for these time periods and sinks values previously determined in section 3.2.

### 3.5.1. Changes in freshwater flows

The freshwater discharge observed in MG2 and MG3 are respectively $Q = 12$ m³ s⁻¹ and $Q = 8$ m³ s⁻¹; these values were used as reference values. Five experiments were conducted under the following different freshwater flows:

(i) Original freshwater flow ($Q_{MG2}$=12 m³ s⁻¹; $Q_{MG3}$=8 m³ s⁻¹)

(ii) Observed freshwater flow reduced by 50%. ($Q_{MG2}$=6 m³ s⁻¹; $Q_{MG3}$=4 m³ s⁻¹)

(iii) Observed freshwater flow set to zero.

(iv) Observed freshwater flow increased by 50%. ($Q_{MG2}$=18 m³ s⁻¹; $Q_{MG3}$=12 m³ s⁻¹)

(v) Observed freshwater flow increased up to $Q$=40 m³ s⁻¹, established as the low-flow condition, following Díez-Minguito et al. (2012).

(vi) Yearly average freshwater flow of 185 m³ s⁻¹, following Costa et al. (2009) and Morales et al. (2020).

The resulting simulations were analyzed at two specific moments of the tidal cycle at Bonanza station: The resulting simulations were analyzed at two specific moments of the tidal cycle at Bonanza station: at high water slack (continuous lines) and at low water slack (discontinuous lines), which closely correspond to maximum and minimum salinity values, respectively (Fig. 6a and 6b) The Original freshwater flow presented in both campaigns (MG2 and MG3) is used as reference case (Experiment -i-). A 50% reduction in freshwater flow (ii) presented for the MG2 simulation barely differs from the current state (Fig. 6c, blue lines), with a maximum difference of 0.5 psu for both tidal instances. The highest increase is found in the experiment (iii) (Fig 6c, cyan lines), where the freshwater flow is cancelled. As seen in Fig. 6c, maximum changes do not exceed 0.9 psu. During low water slacks, when minimum saline intrusion occurs, the zone with the highest differences for both experiments is within the first 20 km from the mouth. Conversely, during high water slack moments, the maximum saline intrusion is present, this zone moves by approximately 10 km with respect to the position in maximum low water slacks moments, the highest salinity differences oscillate from km 15 to km 30.

These results confirm those of section 3.4, showing a shift of the horizontal salinity gradient of about 10 km between the maximum and minimum saline intrusion. For MG3 (Fig. 6d), the same behavior can be observed. The differences between reducing the freshwater inputs by 50% (Experiment ii) or 100% (Experiment iii) show approximately the same salinity concentration as the simulation including the actual freshwater inputs (Experiment i) during this campaign. In this case, it can be seen that the areas with the greatest differences are between 10 km and 40 km for both tidal moments and for both experiments, with the maximum differences for experiments (ii) and (iii) being 0.3 and 0.6 psu, respectively. The small increases observed in the horizontal salinity gradient when reducing freshwater input at the head were expected, as the original freshwater flow during these campaigns was very low.

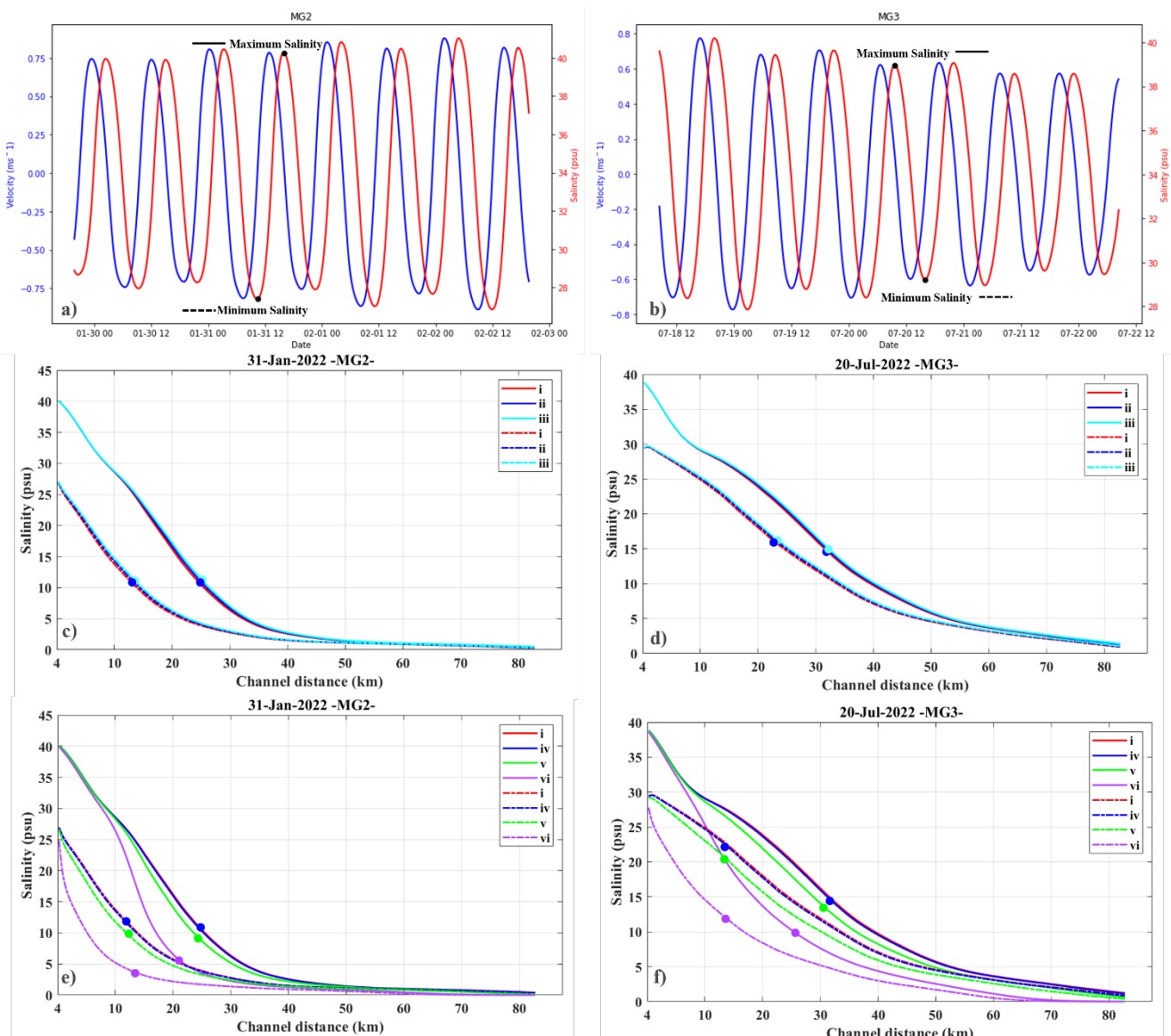

**Figure 6: Superposition of current velocity (m s⁻¹) time series and Salinity (psu) time series at Bonanza station (4 km). black dots mean maximum and minimum salinity moments selected for MG2 (a) and MG3 (b) oceanographic campaigns. Series of salinity (psu) along the Guadalquivir estuary (km) between real flow and various reductions in freshwater flow for MG2 (c) and MG3 (d). In c and d, the red lines represent experiment (i), the blue lines correspond to Experiment (ii) and the cyan lines are experiment (iii). (b) and (d) are the series of salinities using the real freshwater flow and greater freshwater flows for MG2 and MG3, respectively (Experiment (iv) is represented by the blue line, Experiment (v) is green line and experiment vi is presented by pink lines). The solid lines represent the time of maximum salinity at Bonanza, and the dashed lines represent the time of minimum salinity at Bonanza. Color dots represent the km of maximum differences between each experiment with Experiment (i).**

Greater differences were found in experiments (iv), (v), and (vi), where the responses, as expected, were reversed (Figs. 6e and 6f), indicating a reduction of the horizontal salinity gradient. Increasing the flow by 50% (Experiment iii) had no effect in

any of the cases (MG2, MG3), with practically negligible differences (not exceeding 0.3 psu). However, a freshwater flow of 40 $m^3 s^{-1}$ (Experiment v) produced a maximum difference of 2 psu for MG2 and 2.2 for MG3, although these differences practically vanished 40 km upstream from the mouth (Fig 6e and 6f, green lines).

In addition, experiment (vi) (Figs. 6e and 6f, pink lines) show the behavior of the horizontal salinity gradient corresponding to yearly average freshwater flows. These experiments present the largest differences with respect to the reference periods. Maximum differences up to 10 psu are obtained for both tidal periods and both campaigns. For MG2, differences are observed within the first 40 km, and for MG3, the differences between both experiments are observable beyond 80 km from the river mouth. In the upper part of the river, the salt concentration was lower, with differences of less than 3 psu. This indicates that under this flow condition, the estuary has a lower salt concentration along the estuary, showing that the horizontal salinity gradient is less pronounced in this freshwater regime than in the reference case.

These results suggest that, under conditions of high freshwater flows, the saline intrusion from the Gulf is blocked by freshwater flow, resulting in a reduction in the salinization of the estuary. Low flow conditions prevail over 75% of the year (Díez-Minguito et al., 2012).

### 3.5.2 Changes in the water volume sinks.

To evaluate the effect of decreasing or increasing the water withdrawals from GRE, four experiments were conducted, taking the sinks established in the validation of the numerical model for MG2 and MG3 campaigns as a reference:

(i)      Reference $\delta$ ($MG2_{0-22km}$=0.0005 mm, $MG2_{22-42km}$=0.0045 mm, $MG2_{42-85km}$=0.0005 mm, $MG3_{0-85km}$=0.00225 mm)

(ii)      $\delta$ Decrease by 15%. ($MG2_{0-22km}$=0.000425 mm, $MG2_{22-42km}$=0.0038 mm, $MG2_{42-85km}$=0.000425 mm, $MG3_{0-85km}$=0.0019 mm)

(iii)      $\delta$ Decrease by 50%. ($MG2_{0-22km}$=0.00025 mm, $MG2_{22-42km}$=0.00225 mm, $MG2_{42-85km}$=0.0005 mm, $MG3_{0-85km}$=0.0011 mm)

(iv)      $\delta$ Increase by 15%. ($MG2_{0-22km}$=0.000525 mm, $MG2_{22-42km}$=0.0052 mm, $MG2_{42-85km}$=0.000575 mm, $MG3_{0-85km}$=0.0026 mm)

(v)      $\delta$ Increase by 50%. ($MG2_{0-22km}$=0.00075 mm, $MG2_{22-42km}$=0.0067 mm, $MG2_{42-85km}$=0.00075 mm, $MG3_{0-85km}$=0.0034 mm)

(vi)      $\delta$ Increase by 100%. ($MG2_{0-22km}$=0.001 mm, $MG2_{22-42km}$=0.009 mm, $MG2_{42-85km}$=0.001 mm, $MG3_{0-85km}$=0.0045 mm)

Reducing the sinks by 15% (Experiment ii) produces a decrease in salinity up to 0.4 and 0.5 psu for MG2 and MG3 (Fig. 7c and 7d, black lines), respectively, over both analyzed tidal phases (Fig. 7a and 7b). Similarly, a greater reduction (50%, Experiment iii) of these sinks would result in a slight decrease of the horizontal salinity gradient along the GRE, with differences reaching 1.3 psu for MG2 and 1.6 psu for MG3 (Fig. 7c and 7d, red lines). It should be noted that the largest differences occur in the area with the highest variability in the salinity gradient, defined as the zones where the gradient changes most significantly between tidal phases (10km - 40 km) while small differences are obtained further upstream, more close to the head of the estuary (70-85 km).

Analyzing the salinity profiles during the high water slack in Bonanza (maximum salinity), the regions with the largest differences for the MG2 time period (Fig. 7c, solid lines) are located between 12 km and 35 km from the estuary mouth for both Experiments (ii) and (iii). For the minimum salinity instant, the zone of maximum differences extends from the mouth to 25 km upstream (Fig. 7c, dashed lines).

For July 2022 (Fig. 7d), during Experiment (ii), the zones of maximum differences extend 10-30 km from the mouth for both tidal phases. For Experiment (iii), these zones extend from 10-50 km during high water slack and from the mouth to 50 km during low water slack (minimum salinity concentration). Therefore, it can be observed that reducing these sinks slightly decreases the salinity gradient, resulting in less saline intrusion into the system.

Conversely, an increase in the sinks leads to higher salinization of the estuary, especially in the inner part of the river. It must be noted that these are idealized experiments designed to understand how the horizontal salinity gradient would respond to increased water withdrawal, potentially linked to greater anthropogenic pressures (e.g., increased water extraction for crop fields, proliferation of illegal wells, or the creation of new secondary river channels to divert water for anthropogenic purposes).

Referring to Fig. 7e, it can be observed that a 15% or 50% increase in water withdrawal (Experiments (iv) and (v), respectively) leads to a gradual increase in salinity, with maximum differences of 0.7 psu (Experiment iv) and 1.4 psu (Experiment v) for both tidal phases analyzed. Doubling the water withdrawal along the entire channel (Experiment vi) results in a more pronounced increase, with salinity concentrations differences by almost 3 psu throughout the estuary (Fig. 7e, green lines). The areas of greatest change during periods of maximum saline intrusion are between 10 and 40 km from the mouth, while during periods of minimum saline intrusion, the most affected areas are within the first 30 km.

Similarly, for the July 2022 analysis period, Experiment (iv) shows the smallest differences and behaves almost identically to the reference case (Experiment i), with a maximum increase of only 0.5 psu. However, in this case, Experiment (v) shows an increase of nearly 2 psu, while Experiment (vi) records maximum increases of up to 3.5 psu. The areas of greatest discrepancy during periods of minimum saline intrusion are located within the first 40 km from the mouth, while during periods of maximum intrusion these areas extend from 10 to 60 km upstream. Notably, in Experiment vi, the changes also affect the mouth of the estuary during maximum salinity moment.

It is important to highlight that these differences remain relatively small due to the low intensity of the implemented sinks. This is particularly relevant for Experiment (vi), where the largest volume of water is removed, and the threshold for more significant changes is revealed. In other words, more intensive sinks will lead to an increase in the salinity gradient, resulting in much greater saline intrusion into the estuary.

The most affected areas are those near agricultural fields. An increased salinity gradient will result in greater salinization of the water, which will directly affect these fields that rely on water from the main channel for irrigation. This phenomenon will affect not only agricultural activities, but also all other socio-economic and environmental activities that depend on the estuary. The selected times correspond to the moments of maximum and minimum salinity in Bonanza; however, the tidal phase lag along the estuary must be taken into account. This will result in slightly higher or lower salinity concentrations at the exact moments of maximum and minimum salinity at each point along the river.

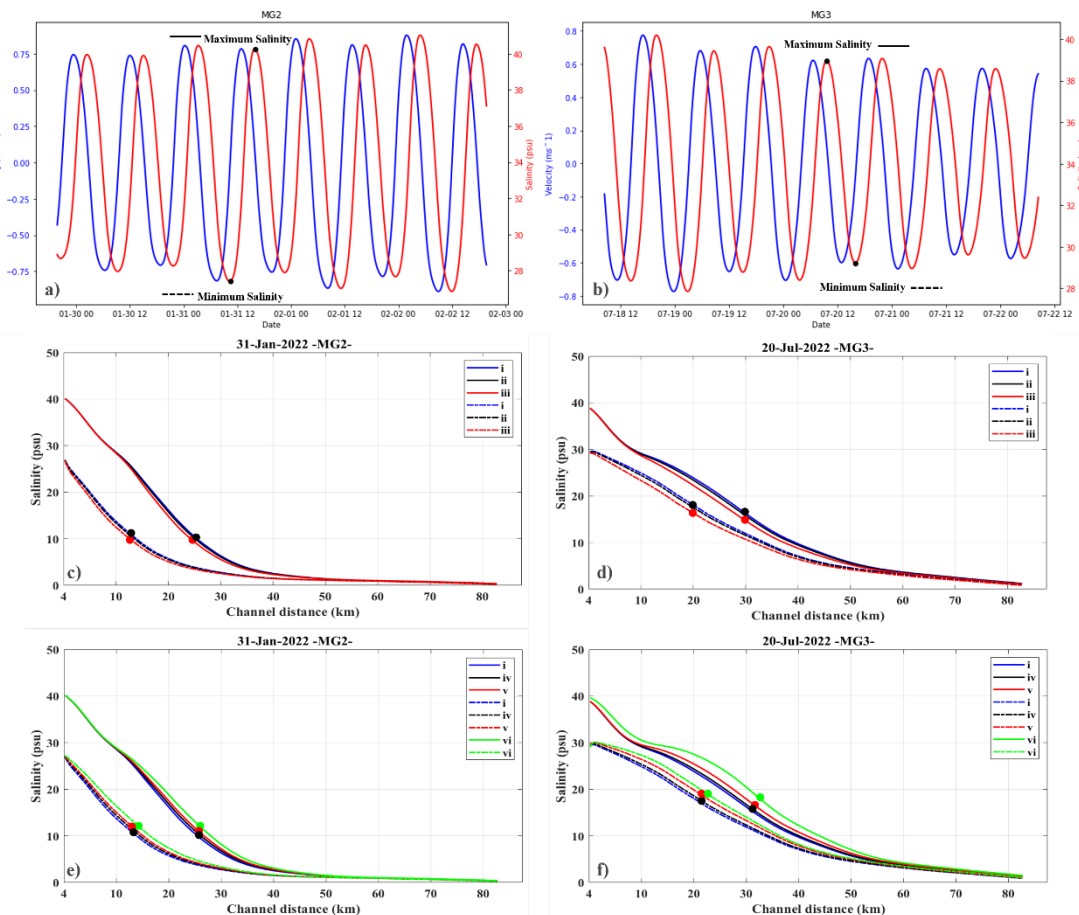

**Figure 7: Superposition of current velocity (ms⁻¹) time series and Salinity (psu) time serie at Bonanza station (4 km). black dots mean maximum and minimum salinity moments selected for MG2 (a) and MG3 (b) oceanographic campaigns. Series of salinity (psu) along the Guadalquivir estuary (km) under a reduction of water withdrawals values are presented in (c) and (d) where original value are represented by blue lines (Experiment i), experiment (ii) corresponding to a smaller reduction is presented in black and a higher reduction of water withdrawal value is presented in red (Experiment (iii). (e) and (f) correspond to experiments increasing water withdrawal values. Original value is presented in blue (Experiment i), and different progressive increases are presented by black lines (Experiment iv), red lines (Experiment v) and green lines (experiment vi). The solid lines represent the time of maximum salinity at Bonanza, and the dashed lines represent the time of minimum salinity at Bonanza. Color dots represent the km of maximum differences between each experiment with experiment (i).**

These findings underscore the significant impact that changes in water volume sinks can have on the salinity dynamics of the Guadalquivir estuary. Reducing the sinks leads to a decrease in salinity, mitigating the extent of the horizontal salinity gradient, while increasing the sinks amplifies it, pushing higher salinity concentrations upstream. This sensitivity to water sinks emphasizes the importance of managing water usage to control salinity levels in the estuary, which is crucial for maintaining ecological balance and water quality.

The relationship between saline intrusion, freshwater flows and the effect of water withdrawals is consistent with findings from other estuaries where changes in freshwater flow regimes have been shown to directly influence horizontal salinity

gradient. For instance, Alcérreca-Huerta et al. (2019) demonstrated an increase in the horizontal salinity gradient, with high salinities reaching up to 46 km from the mouth in the Grijalva River estuary, as a result of reduced freshwater discharge due to dam construction. Similarly, using a model to analyze the relationship between salinity and freshwater flow in the Yangtze River estuary, Webber et al. (2015) showed that reduced freshwater flow leads to greater salt intrusion. In essence, the lower and more prolonged the freshwater discharge, the greater the probability of more intense and prolonged salt intrusion. Huang et al. (2024) in the Changjiang estuary, showed that horizontal salinity gradient in the Changjiang estuary could be limited by controlled and sufficiently high freshwater flows from the Three Gorges reservoir. Extrapolating these findings to the GRE, it is clear that under high flow regimes, or if enough freshwater is released, saline intrusion could be halted by the substantial volume of freshwater flowing down the estuary, counteracting tidal forces.

The effect of water withdrawals, although not in the exact form presented in this study, has been proposed by Huang et al. (2024). These authors analyzed the effect of water withdrawals through three experiments where the volume of water withdrawn was increased from 0 to 500 $m^3$ $s^{-1}$ and finally to 1000 $m^3$ $s^{-1}$, resulting in an increase in the horizontal salinity gradient of approximately 6-7 km (at high water slack and low water slack, respectively) further into the estuary. These withdrawals directly affect the freshwater flow, reducing its volume. These results are consistent with the findings of this study, where water withdrawals are made directly from the channel under low flow conditions, leading to excessive salinization of the Guadalquivir. It is shown that the greater the volume of water withdrawn, the greater horizontal salinity gradient is shown.

The high salinity concentrations associated with the saline intrusion have direct and detrimental impacts on the ecosystem. These impacts include altering water column properties such as turbidity, affecting primary production, and affecting crop fields and domestic water use. Moreover, these effects extend beyond the ecosystem to impact the ecosystem services and associated socio-economic aspects in this coastal area. Therefore, this study highlights the importance of establishing a much higher ecological freshwater flow to mitigate salt intrusion, alongside strict control of water withdrawals in the estuary.

## 5. Conclusions

The nonlinear 1D numerical model used in this study, which incorporates realistic variations in channel width and bottom depth (average values across the channel), as well as withdrawals originating from the system, has demonstrated a highly satisfactory performance achieved through an extensive process of calibration and validation. This process includes comparing simulations with numerous observations of salinity at various points along the estuary.

The changes in freshwater flow ($Q$) lead to slight variations in the current situation within the GRE. A decrease or increase in $Q$ result in slight increases or decreases in salinity concentration. Significant reductions in the horizontal salinity gradient were observed only at moderate to high discharges (above 40 $m^3 s^{-1}$). However, the main driver influencing the behavior of the horizontal salinity gradient under low freshwater regime is the water withdrawals from the estuary which main cause could be related to anthropogenic activities.

The experiments conducted, based on idealized conditions, provide insight into the magnitude of anthropogenic pressures on the salinization of the GRE. The current state reveals an excessive saline intrusion into the GRE, as indicated by the advection and dispersion numerical model incorporating a sink parameter ($\delta$) to simulate the effect of water volume withdrawals from the inner estuary. Thanks to this parameterization we can define a reference situation which would correspond to a situation where no sinks ($\delta=0$) are considered. This reference situation would represent the ideal functioning of the estuary with a minimum horizontal salinity gradient that would never reach more than 25 km from its mouth.

The changes in freshwater flow ($Q$) result in slight variations to the current situation. A decrease/increase in $Q$ slightly increases/decreases salinity concentration. Only at moderate to high discharges (above 40 m$^3$ s$^{-1}$), was a significant reduction in the horizontal salinity gradient observed. However, under low flow regime the main mechanism driving the observed behavior of the horizontal salinity gradient can be attributed to water withdrawals from the estuary, where the main source of this water withdrawal can be attributed to anthropogenic activities taking place in this coastal system.

Furthermore, the impact of anthropogenic activities extends beyond salinity. Upstream, various physicochemical and biological variables, such as nutrients, organic matter, and contaminants, may also accumulate. The removal of the salt intrusion from the estuary depends on the magnitude of freshwater discharge. Moderate discharges (100-200 m³/s) typically shift the salt intrusion to the estuary mouth and reduce tidal currents in the upper estuary. In contrast, significantly higher discharges (approximately one order of magnitude greater) are required to dampen tidal currents throughout the entire estuary. Under such conditions, accumulated substances at the estuary mouth can be exported into the Gulf of Cadiz.

*Data availability.* SAIH data are freely accessible via https://www.ch guadalquivir.es/saih/. Observational dataset and numerical simulations will be available in an online repository if the manuscript is accepted for publication.

*Authors contribution.* SS, MB planned and designed the study. SS has processed an analyzed the data. SS, JJG-P, MB-P, SS, JMS-P, and MB contributed with the analysis performance and interpretation of the results. MB revised the final version. SS has prepared everything.

*Competing interests.* The contact author has declared that none of the authors has any competing interests.

*Acknowledgements.* This work has been developed within the framework of the Spanish National Research Plan through the project TRUCO (RTI1018-100865-B-C22). Sara Sirviente was supported by a grant from the University of Ferrara and the University of Cadiz. Bolado-Penagos was supported by Plan Propio Investigadores Noveles UCA (BOLA642 PR2023-030) and the CEIMAR Jóvenes Investigadores (ASTRAL—CEI-JD-23-08).

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
