# Peer review of "Anthropogenic pressures driving the salinity intrusion in the Guadalquivir Estuary: Insights from 1D Numerical Simulations"

_EGUsphere, 2024_

## Referee Comment (RC1)

1. **The author needs to compare previous related studies. For instance, I have listed some studies, including those on the impact of water extraction on salinity intrusion. Could the author elaborate on the differences and innovations compared to these earlier studies?**

Webber, M., Li, M. T., Chen, J., Finlayson, B., Chen, D., Chen, Z. Y., ... & Barnett, J. (2015). Impact of the Three Gorges Dam, the South–North Water Transfer Project and water abstractions on the duration and intensity of salt intrusions in the Yangtze River estuary. Hydrology and Earth System Sciences, 19(11), 4411-4425.

Huang, H.; Wang, Y.; Wang, S.; Lan, Y.; Huang, X. Saltwater Intrusion in the Changjiang River Estuary in Response to the East Route of the South-to-North Water Transfer Project in the New Period after 2003. Sustainability 2024, 16, 683. https://doi.org/10.3390/su16020683

Jung, C.; Lee, G.; Park, J. Cause Analysis of Salinity Intrusion by Environmental Changes Considering Water Intake and Sand Mining on Seomjin River Estuary Using Model for Maintaining Corbicula Habitats. Water 2024, 16,1035. https://doi.org/10.3390/w16071035

Alcérreca-Huerta, J. C., Callejas-Jiménez, M. E., Carrillo, L., & Castillo, M. M. (2019). Dam implications on salt-water intrusion and land use within a tropical estuarine environment of the Gulf of Mexico. Science of The Total Environment, 652, 1102-1112.

2. **Why can the Guadalquivir River Estuary (GRE) be simplified into a one-dimensional model for study? What is the structure of the vertical circulation, and how does it affect salinity intrusion?**

3. **The tuning of the δ parameter was adjusted to match the observational data. Could the model be influenced by other factors, such as the bottom friction coefficient or the horizontal diffusion coefficient D? How should the δ value be determined when studying other estuaries? In other words, what insights does the δ value used in this study offer for applications to other estuaries?**

4. **What is the basis for determining D = 0.5 m²/s? Would using other parameterization schemes for D across the entire area significantly affect the salinity intrusion?**

5. **How can the impact of human pressure on salinity intrusion be quantitatively**

assessed based on the 1D diffusion equation in this study? Is it through its effect on advective transport or horizontal diffusive transport, thereby influencing salinity transport? Which of these two processes contributes more?

6. There are three tributary estuaries in this study, but they don't seem to be marked on the figures. Additionally, how was the runoff distributed among these three estuaries? In the experiments with increased or decreased runoff, was the flow rate adjusted simultaneously for all three tributary estuaries?

7. How is the fact that water withdrawal does not occur throughout the entire estuary, but at specific locations, taken into account? This localized withdrawal will also lead to a reduction in the overall runoff of the estuary. Would this have any impact on the study's results?

8. How is the water withdrawal process represented in the governing equations? In other words, how is the dynamic process of water withdrawal parameterized in the governing equations?

9. In the introduction, could you add some related studies on the impact of human activities on salinity transport in other estuaries?

---

## Referee Comment (RC2)

In this paper, the authors implement 1D modelling to investigate salt intrusion in the Guadalquivir River Estuary. They validate their model using observations from four different campaigns and attribute any discrepancies to water withdrawals related to human activities around the estuary. They also do further experiments by changing the freshwater input. The topic is interesting, and the paper is well written. However, there are a few concerns regarding their approach and methodology.

1. The authors will need to justify better the use of an 1D model for salt intrusion as this neglects the effect of vertical salinity gradients which contribute to salt intrusion. In this case, a constant diffusion coefficient is not enough to account for any unresolved mixing. In addition, Figure 1b and d show that both the width and the depth of the channel can be significant and so it is dubious if averaging can be justified. Have the authors considered the use of a 2DV model instead?

2. There is an inconsistency in the terminology. In some instances, the authors refer to salt intrusion and in others to salt wedge or even salt front and it seems they don't distinguish between these terms. I would advise to remain consistent throughout the manuscript and give an explicit definition. Salt intrusion is usually measured as the landward penetration of a bottom isohaline while the salt wedge is defined as a bottom layer of denser than the surface water. Consequently, I reckon that what is seen in the figures is rather the salinity horizontal gradient (or salinity front) instead of salt intrusion or wedge. Furthermore, the model results are compared with observations taken at 2m below the surface, but the depth can be much deeper in certain sections as it can be seen in Figure 1b. Therefore, I think it is possible that the discrepancy observed between model results and observations without the sinks may be due to the depth averaging which may moderate higher bottom salinity.

3. In continuation to the previous comment. The authors assume that the salinity deficit in their uncalibrated model is exclusively due to water withdrawals. I appreciate that this is an important parameter and even more true for this specific study case, but I believe that the assumption neglects all the other complex physical processes and mechanisms taking place in an estuary. The authors already mention in their manuscript tidal amplification and channel deepening. Don't these two also account for an upstream increase in salinity?

4. The salt transport module was run for the periods when observations from the measurement campaigns that took place between 2021-2023 where available but the hydrodynamic model is forced with data from 2019! How is this justified? This could be already a source of errors.

**Minor comments**

1. I understand the notation used throughout the manuscript as km 60, km 40 etc. but it doesn't read very well. It is better if it is written as 60 km from the mouth, 40 km from the mouth etc.

2. Please use superscript numbers when giving units (e.g., lines 48, 50 ,197 etc.)

3. Where are the river flows implemented?

4. It is implied that there is no freshwater input from the upstream boundary which is set at the dam. Is this realistic? Is it true for every season?

5. In Line 90, I think the authors of this paper refer to salt intrusion length and not duration.

6. There is a confusion in the manuscript. In some instances, the authors write that the maximum salt intrusion corresponds to the flood and in others to the ebb tide. For example:
Lines 324-325 the authors write ' *The maximum and minimum extent of the saline wedge within the channel coincided with moments just before high and low tides respectively'.* In the next paragraph they write ' *during the flood tide the wedge demonstrates minimal intrusion in the estuary ...... during the ebb tide, the maximum saline intrusion occurred'.*
Line 375-376 ' *the maximum ebb current and the maximum flood current which closely correspond to the maximum and minimum salt wedge intrusion, respectively'.*
But then a few lines further down:
Line 380 ' During maximum ebb current (just after low tides), when minimum salt wedge intrusion occurs......during flood tides (just after high tides), the maximum salt intrusion is present'.
In the legend of Figure 5 '*The solid lines represent the time of maximum salinity (F,Flood) and the dashed lines represent the time of minimum salinity (E,Ebb).'*

At least, Figure 4a shows that the maximum salinity corresponds to the flood tide which is reasonable for a well-mixed estuary.

7. The term 'salt wedge intrusion' is not right. It is either salt intrusion or salt wedge, not all together.
8. Figure 3, indicate where km 30 , 40, 50 etc. is
9. Line 323-324 what do you mean '*a gradual decrease in salinity values upstream can be seen'* . Do you mean gradual decrease during neap tide?

---

## Referee Comment (RC3)

Review of Anthropogenic pressures driving the salinity intrusion in the Guadalquivir Estuary: Insights from 1D Numerical Simulations by Sirviente et al.

The authors use ship-based observations and 1-D hydrodynamic model to analyze the impact of anthropogenic-driven freshwater withdrawal on the salt intrusion in the Guadalquivir Estuary. Model sensitivity experiments indicate that enhancing the freshwater flow and volume (mostly regulated by the dam) leads to a reduction in salt intrusion into the estuary. The study highlights the need to include the sink term associated with freshwater withdrawal from human activities in the model to accurately depict the observed salinity wedge in the estuary.

This is an interesting study and has implications for regulating domestic water use and understanding the impacts of salinity intrusion on the primary production and marine ecosystems. The manuscript is generally well written with good quality figures. However, the authors need to address the following concerns related to the methodology and analysis.

Major comments:

1. The study is based on assumptions which need to be clearly stated in section 2. Please mention how processes such as vertical mixing at the edge of the salinity front, which can significantly influence salinity distribution across the estuary, are accounted for in the model. Include a discussion of the vertical structure of the salt wedge and related citations in the Introduction. The model is validated using salinity data collected at 2 m depth. Salt intrusions could be happening at deeper depths, which seem to be unaccounted for in this study. Please justify.

2. Apart from the anthropogenic freshwater withdrawal, the sink term may also include uncertainties related to unaccounted processes such as drainage from marshes and crop lands, evaporation, vertical mixing etc. A strong justification on the attribution of sink term to anthropogenic effects has to be provided.

3. Fig. 1b shows that the channel is deep in the 15-25 km distance range, where the salt intrusions appear to be more pronounced (Figs. 5,6). It could be that the mixing induced by strong tidal currents at these depths result in increase in salinity, which is not related to freshwater withdrawal.

4. As noted by the other reviewers, there is confusion regarding the different terminology used for terms such as 'salt wedge' and 'salinity front'. Be consistent with the terminology and define a salt front/wedge.  I guess it indicates the region where the lateral gradient in salinity is maximum. In Figs. 5,6 – Mark the location of maximum lateral change in salinity on each curve with a dot in respective color. It will be helpful for the readers to see the spatial variation in the salinity front in each model run.

5. Observation data from the cruises are gathered in different months, ranging between July-February each year. I'm assuming the anthropogenic water withdrawals do not vary much across these months. Please mention that in the data section.

Minor comments:

Authors mention mooring observations are used. Are MG1, MG2 and MG3 mooring locations or sampling points for ship? Are the mooring observations integrated with the ship-based data? It may be good to mark the moorings in Fig. 1 and mention the locations in the caption. The validation of model results using mooring observations is not shown. It may also be good to add a scatter plot between near-surface salinity from moorings and 2 m salinity from ship-based thermosalinograph data to see how they compare.

Fig.1c , y-axis label needs to be corrected to "width"

Line 37: Not sure what the word "positive" means in this context.

Lines 48 and 50: m3/s should be $m^3$/s. Superscript missing in the units in several other places. Please correct.

Fig. 4 – It is not clear if this model simulation includes sink term or not. Also, please mention in the caption what the contours represent. How does the salt intrusion differ during the spring and neap tidal cycles before and after including the sink term? It may be worth checking that.

Fig. 5 – Is this the model surface salinity plotted? Please mention the depth of salinity in the caption. Also, change the legend label in panels (b) and (d) to F +50% Q=18 $m^3$/s

Fig. 6 – Use the same y axis limits for panels (a) and (b).

Line 249-250: The November 2023 results are not shown in Fig. 2

Line 300: may have "an impact" on the salinity wedge penetration

Line 396: What is 2.5 psu difference? Is it the difference between the slopes of the two lines? Also, in what distance regime?

Line 446: through idealized model setup?

---

## Author Comment (AC1)

**CC1**: 'Comment on egusphere-2024-2451', Wenping Gong, 19 Aug 2024  reply

*1.The dispersion coefficient D was defined as a constant in your simulations, it seems not be justified as it should vary with tidal strength and bathymetry and geometry of the estuary;*

The dispersion coefficient D was modeled as a constant in this study for several reasons, as outlined below. Primarily, the simplicity of a constant value ensures numerical stability and facilitates the interpretation of results. This selection of a constant diffusion coefficient is based on the assumption that lateral dispersion is homogeneous and that strong currents will induce vertical mixing, thereby rendering advection the dominant process in the behavior of the salinity intrusion. Considering an along-channel current velocity gradient of $\Delta U$=0.5 m in a distance of L=25000 m and an unrealistic high value for D=10.0 m$^2$/s yields a Péclet number, Pe=$(\Delta U.L)/D$=1250 (largely exceeding 1) which is indicating that advective transport clearly dominates over diffusive transport.

Furthermore, given the lack of comprehensive data on the coefficient's variability across the estuary, it was determined that a constant value would be an adequate representation of the general conditions. Finally, model validation with observational data has demonstrated that employing a constant coefficient is an effective method for accurately reproducing the essential characteristics of the system, thereby supporting this approach within the context of the present study. Furthermore, numerous studies in the literature have demonstrated that models with a constant dispersion coefficient are capable of accurately reproducing salinity distributions (e.g., Lewis and Uncles, 2003; Brockway et al., 2006; Gay and O'Donnell, 2007, 2009; Xu et al., 2019).

Brockway, R., Bowers, D., Hoguane, A., Dove, V., and Vassele, V.: A note on salt intrusion in funnel–shaped estuaries: Application to the Incomati estuary, Mozambique, Estuar. Coastal Shelf S., 66, 1–5. https://doi.org/10.1016/j.ecss.2005.07.014, 2006.

Gay, P. and O'Donnell, J.: A simple advection–dispersion model for the salt distribution in linearly tapered estuaries, J. Geophys. Res., 112, C07021. https://doi.org/10.1029/2006JC003840, 2007.

Gay, P. and O'Donnell, J.: Comparison of the salinity structure of the Chesapeake Bay, the Delaware Bay and Long Island Sound using a linearly tapered advection–dispersion model, Estuaries Coasts, 32, 68–87. https://doi.org/10.1007/s12237-008-9101-4, 2009.

Lewis, R. E. and Uncles, R. J.: Factors affecting longitudinal dispersion in estuaries of different scale, Ocean Dynam., 53, 197–207. https://doi.org/10.1007/s10236-003-0030-2, 2003.

Xu, Y., Hoitink, A. J. F., Zheng, J., Kästner, K., and Zhang, W.: Analytical model captures intratidal variation in salinity in a convergent, well-mixed estuary, Hydrology and Earth System Sciences, 23, 4309–4322. https://doi.org/10.5194/hess-23-4309-2019, 2019.

*2. The specification of withdrawl amount of freshwater along the estuary is not justified. Not sure if it can be determined from observation data or from other statistical data.*

Thank you for your comment. With regard to the withdrawal values, these were established through empirical means during the calibration process. In this coastal area, the available information is scarce and imprecise due to the prevalence of unregulated withdrawals. This makes it challenging to obtain accurate data on the withdrawals occurring in the estuary, as both the locations and the volume of water withdrawn are

unknown. The data provided by the authorities is insufficient for the purposes of this study, as it offers values at the basin level rather than for the specific estuary area targeted by this study. Accordingly, we have calculated this factor through a comprehensive calibration process, using a distinct factor for each time period and selecting the factor that yielded the most accurate simulation in line with the observations. Therefore, we present these experimentally derived approximate coefficients, which are accepted when validating simulations against observations. In fact, the use of this approach provides the more remarkable result, in the sense that it has allowed to demonstrate the necessity of incorporate this anthropogenic contribution in the hydrodynamical simulations as a necessary process to accurately reproduce the actual behavior of the saline intrusion in the estuary.

3. The terminology of "salt wedge" is confusing, as you mentioned that the estuary is well-mixed and your salt transport equation is based on the assumption of well-mixed estuary.

Thank you for your comment. Indeed, in well-mixed estuaries like this one, the term "salt wedge" may not be the most accurate. We will use the term "salinity intrusion" or "saline front" accordingly to the vertically mixed character of the estuary. We have changed it in the new version of the manuscript.

Lines
(12,15,76,89,97,99,105,110,215,287,300,305,308,311,313,320,330,332,339,343,352,359,360,369,382,387,396,400,404,410,424,438,440,452,454,458,460,464)

---

## Author Comment (AC2)

**CC1**:

Thanks to the reviewers' comments, we have adjusted the parameterization of the dispersion coefficient. Therefore, we would like to respond to your questions again, including an updated answer that takes these changes into account. We appreciate your feedback.

1.The dispersion coefficient D was defined as a constant in your simulations, it seems not be justified as it should vary with tidal strength and bathymetry and geometry of the estuary;

Thanks to the reviewers' comments, we reviewed and adjusted the transport model to use a higher, more realistic constant diffusion coefficient without introducing instabilities into the model. In the revised version, based on Bowden (1983), we defined the most appropriate horizontal dispersion coefficient, taking into account the mean depth and tidal amplitude. This calculation was performed for all campaigns included in the analysis to ensure the use of a constant dispersion value appropriate for the system. The results indicate that the maximum constant dispersion based on velocity and depth is 150 m²/s (this has also been added and explained in detail in the revised version of the article). In addition, a sensitivity analysis was performed with different dispersion coefficients to optimize this parameter as much as possible. It was found that a dispersion value higher than 200 m²/s leads to numerical instabilities in the system.

Based on Bowden (1983), the effective horizontal dispersion coefficient can be calculated as $K_X = U^2 H^2 / 30 * K_z$, where $U$ is the maximum tidal velocity, $H$ is the mean channel depth, and $Kz$ is the vertical eddy dispersion coefficient, assumed to be constant. In our case, we used $Kz=0.01$, as proposed by Bowden (1983).

| Campaign | U (ms$^{-1}$) | K$_x$ m$^2$s$^{-1}$ |
|----------|---------------|---------------------|
| MG1 | 0,85 | 143 |
| MG2 | 0,88 | 154 |
| MG3 | 0,88 | 153 |
| MG4 | 0,80 | 127 |

Furthermore, given the lack of comprehensive data on the coefficient's variability across the estuary, it was determined that a constant value would be an adequate representation of the general conditions. Finally, model validation with observational data has demonstrated that employing a constant coefficient is an effective method for accurately reproducing the essential characteristics of the system, thereby supporting this approach within the context of the present study. Furthermore, numerous studies in the literature have demonstrated that models with a constant dispersion coefficient are capable of accurately reproducing salinity distributions (e.g., Lewis and Uncles, 2003; Brockway et al., 2006; Gay and O'Donnell, 2007, 2009; Xu et al., 2019; Siles-Ajamil et al., 2019; Biemond et al., 2024). This choice not only maintains the stability of the model, avoiding numerical instabilities, also ensures that the results are consistent with theoretical expectations and experimental observations.

Regarding the use of a 2D model, although we have not explicitly applied a 2D model to the Guadalquivir estuary, we have analyzed its analytical solution. This approach allows us to observe

that the differences in velocity in the longitudinal direction of the estuary are not significantly different from those obtained with the 1D version.

In Figure R1, we present the analytical solution for a channel-cross section using a 2D model that includes the channel width and a parabolic depth variation that approximates the change in depth from the lateral boundaries to the center of the channel. For comparison, we also include the 1D solution for the same channel (length = 5 km, width = 525 m, 2-day simulation) with an average depth of 6.7 m.

As shown, there are differences at the lateral boundaries, within the first 100 m on either side of the channel. However, the oscillations are not significant, and the velocities are very similar across most of the channel width. This shows that the behavior is generally homogeneous, validating the 1D solution. This conclusion is confirmed by the average velocities obtained for each solution at different times (Table R1), where the differences between the average velocities of the two models are minimal.

It is true that using the 1D solution slightly underestimates the velocity in the center of the channel and slightly overestimates it at the boundaries. However, these discrepancies do not affect the results, as the model has been shown in Sirviente et al. (2023) to reproduce the observations with high reliability.

This reinforces the idea that the use of a 2D model does not provide a substantial improvement over the 1D model.

[Figure]

**Figure R1.** The top panels display the time series of the longitudinal velocity (uu) for section 20 of the channel. Colored markers highlight the three consecutive hours analyzed in the bottom panels. The bottom panels compare the velocity profiles obtained from the 2D model (solid lines) with those from the 1D model (dashed lines) at the selected times, illustrating the differences across the channel width.

**Table R1.** Average Velocity in Section 20 of the Idealized Channel for 2D and 1D Simulations Over a 6-Hour Period (3 Hours of Flood Tide and 3 Hours of Ebb Tide)

|  | u average 2D (ms$^{-1}$) | u average 1D (ms$^{-1}$) |
|---|---|---|
| *Flood Hour 1* | -0.75 | -0.86 |
| *Flood Hour 2* | -0.52 | -0.61 |
| *Flood Hour 3* | -0.16 | -0.21 |
| *Ebb Hour 1* | 0.82 | 0.89 |
| *Ebb Hour 2* | 0.62 | 0.69 |
| *Ebb Hour 3* | 0.26 | 0.32 |

We appreciate the reviewer's suggestion and will take it into account for future work.

Biemond, B., de Swart, H. E., & Dijkstra,H. A. (2024). Quantification of salttransports due to exchange flow and tidalflow in estuaries. Journal of GeophysicalResearch: Oceans, 129, e2024JC021294. https://doi.org/10.1029/2024JC021294

Bowden, K. F. (1983). *Physical Oceanography of Coastal Waters*. Ellis Horwood Series in Marine Science. E. Horwood. [University of Michigan, Digitized Nov 3, 2007]. ISBN 0853126860, 9780853126867.

Brockway, R., Bowers, D., Hoguane, A., Dove, V., and Vassele, V.: A note on salt intrusion in funnel–shaped estuaries: Application to the Incomati estuary, Mozambique, Estuar. Coastal Shelf S., 66, 1–5. https://doi.org/10.1016/j.ecss.2005.07.014, 2006.

Gay, P. and O'Donnell, J.: A simple advection–dispersion model for the salt distribution in linearly tapered estuaries, J. Geophys. Res., 112, C07021. https://doi.org/10.1029/2006JC003840, 2007.

Gay, P. and O'Donnell, J.: Comparison of the salinity structure of the Chesapeake Bay, the Delaware Bay and Long Island Sound using a linearly tapered advection–dispersion model, Estuaries Coasts, 32, 68–87. https://doi.org/10.1007/s12237-008-9101-4, 2009.

Lewis, R. E. and Uncles, R. J.: Factors affecting longitudinal dispersion in estuaries of different scale, Ocean Dynam., 53, 197–207. https://doi.org/10.1007/s10236-003-0030-2, 2003.

Xu, Y., Hoitink, A. J. F., Zheng, J., Kästner, K., and Zhang, W.: Analytical model captures intratidal variation in salinity in a convergent, well-mixed estuary, Hydrology and Earth System Sciences, 23, 4309–4322. https://doi.org/10.5194/hess-23-4309-2019, 2019.

2. The specification of withdrawl amount of freshwater along the estuary is not justified. Not sure if it can be determined from observation data or from other statistical data.

Thank you for your comment. With regard to the withdrawal values, these were established through empirical means during the calibration process. In this coastal area, the available information is scarce and imprecise due to the prevalence of unregulated withdrawals. This makes it challenging to obtain accurate data on the withdrawals occurring in the estuary, as both the locations and the volume of water withdrawn are unknown. The data provided by the authorities is insufficient for the purposes of this study, as it offers values at the basin level rather than for the specific estuary area targeted by this study. Accordingly, we have calculated this factor through a comprehensive calibration process, using a distinct factor for each time period and selecting the factor that yielded the most accurate simulation in line with the observations. Therefore, we present these experimentally derived approximate coefficients, which are accepted when

validating simulations against observations. This approach also demonstrates the necessity of including this anthropogenic factor in simulations to accurately reproduce the real behavior of the system.

3. The terminology of "salt wedge" is confusing, as you mentioned that the estuary is well-mixed and your salt transport equation is based on the assumption of well-mixed estuary.

Thank you for your comment. Indeed, in well-mixed estuaries like this one, the term "salt wedge" may not be the most accurate. We will use the term "saline intrusion" instead, as it better describes the area of the estuary where high salinity is present due to tidal influence.

We have change it in the new version of the manuscript.

---

## Author Comment (AC3)

**Reviewer 1**

1. **The author needs to compare previous related studies. For instance, I have listed some studies, including those on the impact of water extraction on salinity intrusion. Could the author elaborate on the differences and innovations compared to these earlier studies?**

Thank you for the suggestion. We have included some paragraphs comparing our study with these early studies. It has been included in line 460- 478 as follows:

"The relationship between saline intrusion, freshwater flows and the effect of water withdrawals is consistent with findings from other estuaries where changes in freshwater flow regimes have been shown to directly influence saline intrusion. For instance, Alcérreca-Huerta et al. (2019) demonstrated an increase in saline intrusion, with high salinity reaching up to 46 km in the Grijalva River estuary, as a result of reduced freshwater discharge due to dam construction. Similarly, using a model to analyze the relationship between salinity and freshwater flow in the Yangtze River estuary, Webber et al. (2015) showed that reduced freshwater flow leads to greater saline intrusion. In essence, the lower and more prolonged the freshwater discharge, the greater the probability of more intense and prolonged saline intrusion. Huang et al. (2024) in the Changjiang estuary, showed that salinity intrusion into the Changjiang estuary could be limited by controlled and sufficiently high freshwater flows from the Three Gorges reservoir. Extrapolating these findings to the GRE, it is clear that under high flow regimes, or if enough freshwater is released, salinity intrusion could be halted by the substantial volume of freshwater flowing down the estuary, counteracting tidal forces.

The effect of water withdrawals, although not in the exact form presented in this study, has been proposed by Huang et al. (2024). These authors analyzed the effect of water withdrawals through three experiments where the volume of water withdrawn was increased from 0 to 500 m³/s and finally to 1000 m³/s, resulting in an increase in saline intrusion of approximately 6-7 km (at flood and ebb tide, respectively) further into the estuary. These withdrawals directly affect the freshwater flow, reducing its volume. These results are consistent with the findings of this study, where water withdrawals are made directly from the channel under low flow conditions, leading to excessive salinization of the Guadalquivir. It is shown that the greater the volume of water withdrawn, the greater the salinity intrusion into the system.

Therefore, this study highlights the importance of establishing a much higher ecological freshwater flow to mitigate saltwater intrusion, alongside strict control of water withdrawals in the estuary."

2. **Why can the Guadalquivir River Estuary (GRE) be simplified into a onedimensional model for study? What is the structure of the vertical circulation, and how does it affect salinity intrusion?**

Thank you for the question.

The possibility of simplifying the Guadalquivir model to a one-dimensional (1D) channel is mainly based on its geometric and hydrodynamic characteristics. This estuary is considered a semi-closed system due to the presence of the Alcalá del Río dam and is characterized by homogeneous mixing (Álvarez et al., 2001; Diez-Minguito et al., 2013). With a length of 110 km, its depth and width are reduced compared to its length. According to our bathymetry, based on the 2019 nautical chart provided by the Hydrographic Service of the Spanish Navy, the depth

ranges from 5 to 18 m and the width varies between 100 and 400 m. This configuration favors the channeling of the flow along the estuary and makes that longitudinal transport processes dominate over the transversal ones.

Under low freshwater flow conditions, the estuary is dominated by tidal influence (Diez-Minguito et al., 2012), resulting in minimal variations in the transverse direction (perpendicular to the flow). Thus, the incoming flow is characterized by a predominant longitudinal direction, where transverse variations are small and insignificant for representing the hydrodynamic behavior of the estuary.

The Guadalquivir is a meso-tidal estuary, with a tidal range of about 3.5 m during spring tides, with the M2 tide as the main component (TM2 = 12.42 hours). The propagation of the tide under normal conditions (low freshwater flow) is due to reflection, friction and convergence of the main channel (Díez-Minguito et al., 2012). In terms of the vertical structure of salinity, the estuary is predominantly characterized by intense mixing (Díez-Minguito et al., 2013), which results in a homogeneous distribution of water properties, such as salinity, with minimal vertical differences in this aspect. Similarly, vertical circulation is characterized by a relatively uniform current pattern during low flow periods, with no significant variations in flow velocity or direction at different depths (Sirviente et al., 2023).

These conditions allow the equations describing the hydrodynamics to be reduced to those of a one-dimensional channel. 1D hydrodynamic models have been shown to be effective in representing the hydrodynamic behavior of natural systems, such as rivers, and significantly reduce computational time compared to 2D and 3D simulations. Previous studies have validated the effectiveness of 1D hydrodynamic models in the Guadalquivir estuary (e.g., Álvarez et al., 2001; Siles-Ajamil et al., 2019).

Furthermore, a thorough validation of the applied one-dimensional model is detailed in Sirviente et al. (2023). In this study, the good performance of the hydrodynamic model is demonstrated by validating the simulations with numerous observations collected over six years of oceanographic campaigns. The 1D simulations are validated against observations from current meters moored at different points in the estuary. In addition, data from surface tide gauges provided by *Puertos del Estado* are used. The results show that the model is in good agreement with all observations, supporting the conclusion that the 1D model can be effectively used to study tidal dynamics in this estuary, where the simulations show high reliability.

The vertical mixing and the reduced salinity gradient are evidenced by the data recorded by the CTD during each campaign. CTD profiles were performed at the different sampling points shown in Figure 1 of the manuscript, allowing us to analyze the vertical behavior of the salinity. The figures below correspond to the vertical salinity profiles measured during the MG1, MG2 and MG3 campaigns.

For MG3 (Fig. R1), CTD profiles were taken at 1-hour intervals over two tidal cycles, for a total of 25 hours at each site. For the MG1(Fig. R2) and MG2 (Fig. R3) campaigns, measurements were taken at 1-hour intervals for a maximum of 10 hours. The sampling points for MG3 are the same as those shown in Figure 1 of the manuscript, as is the case for MG2. However, for MG1 there are more CTD sampling points than shown in the figure. The points for MG1 shown in the CTD plots correspond exactly to the positions of MG2 (see Figure 1a of the manuscript). It should be noted that there are no vertical profiles available for MG1-1 and not all sampling stations have 10 profiles.

When analyzing the behavior of the vertical profiles, it can be observed that in MG3 the water column always remains mixed, showing only very slight vertical salinity gradients during certain hours. Fig. R1 shows that vertical mixing prevails during the tidal cycles. Similarly, during the MG2 and MG1 campaigns, a strong vertical mixing is observed throughout the water column at all CTD profiles in all points of the river.

This observation was essential in simplifying our approach, allowing us to adopt a one-dimensional (1D) model, assuming that the salt concentration is homogeneous throughout the water column.

Motivated by the reviewer comment, we have included these figures in the supplementary material and added a few lines in the methodology and results section to reflect that the estuary has a practically homogeneous vertical behavior for the periods analyzed in this study.

[Figure]

**Fig. R1**. Top panel corresponds to the tidal current velocity at each sampling station during the MG3 campaign, with different colors indicating the tidal phases during which each CTD profile was taken. The bottom panel displays the CTD profiles at each sampling point along the Guadalquivir River during the MG3 campaign.

[Figure]

**Fig. R2.** CTD profiles at each sampling point along the Guadalquivir River during the MG2 campaign.

[Figure]

**Fig. R3.** CTD profiles at each sampling point along the Guadalquivir River during the MG1 campaign.

3. **The tuning of the δ parameter was adjusted to match the observational data. Could the model be influenced by other factors, such as the bottom friction coefficient or the horizontal diffusion coefficient D? How should the δ value be determined when studying other estuaries? In other words, what insights does the δ value used in this study offer for applications to other estuaries?**

Thank you for the question.

The parameter δ is a key factor in quantifying the effect of extractions and minor contributions to water volume along the river. In other words, it represents all the natural and anthropogenic processes that can affect the volume of water, such as agricultural abstraction, industrial use, small side channels and evaporation. δ should be interpreted as a bulk value that characterizes the balance between inputs and outputs of water in the estuary. In our study was positive, indicating that on average extractions exceed the contributions from smaller channels that drain into the main channel.

However, there is an inherent uncertainty in this parameter due to the complexity of accurately quantifying the amount of water extracted from the channel. The Guadalquivir system is heavily influenced by human activities (high levels of agriculture, industry, dense population in nearby areas, port activities, etc.), and it is also documented that numerous illegal extractions take place. This makes it difficult to obtain accurate data on abstraction within the estuary, as both the specific locations and volumes of water taken are unknown.

The main idea of this study is to show that these actions have a significant effect under low flow conditions, because without them the observed salinity levels would not be reached.

Thanks to the comments of the reviewers, we have reviewed the parameterizations and adapted the code to use higher dispersion values while maintaining the numerical stability of the model. This allowed us to perform a sensitivity analysis using a higher dispersion coefficient (150 m²/s) to evaluate whether the system could reproduce these observed salinity conditions with horizontal dispersion alone. All this information has been included in the new manuscript version as a new section: "3.1. Effect of Horizontal Dispersion and Water Withdrawal on the Horizontal Salinity Gradient in the Estuary".

In this article, the horizontal dispersion coefficient was calculated using the equation proposed by Bowden (1983) for estimating the horizontal dispersion coefficient. This calculation was carried out for all the campaigns considered in the analysis to ensure the use of a constant dispersion appropriate to the system. The results indicate that the maximum constant dispersion, based on speed and depth, is 150 m²/s (this has also been added and explained in detail in the new version of the article). In addition, a sensitivity analysis was carried out with different dispersion coefficients to optimize this parameter as much as possible. In our 1D model, which simplifies the equations governing the balance forces, volume conservation, and advection-dispersion processes, a dispersion coefficient exceeding 200 m²/s leads to numerical instabilities.

If we analyze the behavior of the horizontal salinity gradient along the estuary, taking into account only the horizontal dispersion, we can see that the system would never reproduce the observed salinity concentrations in the different campaigns. Even when the dispersion coefficient is increased to 190 m²/s, the behavior remains the same (Fig R4). It can be observed that, over time, the salinity concentration increases slightly from 10 km to 40 km. However, it is evident that the observed values are not fully reached. Therefore, it can be said that horizontal dispersion alone does not achieve the high salinity values observed along the channel, which opens the hypothesis that some additional effect is likely to cause a greater penetration of saline intrusion into the estuary.

If the same experiment is carried out but the parameter δ (representing all the processes that reduce the volume of water in the estuary) is included (Fig. R4), the results show that the system reaches the observed salinity values over time. This shows that this term must be included in order to reproduce the salinity concentrations observed in the different campaigns.

Figures 4c and 4d present the experiments that include water withdrawals as a constant value in time and space ($\delta = 0.005$ mm). As shown, as the simulation time progresses, the system achieves the salt concentrations range presented in the observations. This contrasts with the previous cases, where only the dispersion term was included in the experiments (Figures 4a and 4b), and the observed range could not be achieved.

On the other hand, Figures 4e and 4f show the experiments employing time-varying $\delta$. In this simulation, a stronger sink is applied during the first three days ($\delta = 0.01$ mm), which is then reduced and held constant for the remainder of the simulation ($\delta = 0.001$ mm). In this case, the obtained values closely match those recorded in the observations.

These experiments highlight the necessity of including this term ($\delta$) in the simulations, as otherwise, the horizontal salinity gradient would never reach the observed values. As illustrated in the figure, a certain duration of sink activity is required for the simulated salinity concentrations to approach the range observed. Therefore, it is essential to define an initial condition that considers this progression, allowing the simulation to adequately capture the evolution of the water withdrawals and their impact on salinity over time. This is particularly justified by Figures 4e and 4f, where using a stronger $\delta$ during the first three days followed by a weaker $\delta$ reproduces the observed behavior.

[Figure]

**Fig. R4.** Comparison of simulated (temporal behavior of the simulation at the corresponding observation points) and observed salinity over the MG3 vessel trips: (a) and (b) show simulations including only the horizontal dispersion for the MG3 vessel trip upstream (a) and downstream (b), with the observations in black. (c) and (d) show simulations incorporating δ term (δ) for the entire simulation period as constant value, and (e) and (f) are the simulation including a time-varying δ, compared to the observations (in black) for the MG3 vessel trip upstream (c and e) and downstream (d and f).

The δ value was determined empirically during the calibration process, through a sensitivity analysis in which different δ values were tested in simulations to identify the one that produced concentrations within the observed range while maintaining the temporal and spatial stability of the model.

Once the δ value was identified, experiments were carried out to analyze its behavior. These included constant use of the parameter over time, and experiments including them at specific time intervals, as well as spatial distribution experiments where δ was applied to specific points (e.g. high areas of the river) and regions. However, due to the limited understanding of the true behavior of these processes, and to avoid introducing assumptions or speculation that could affect

the validity of the results, it was considered more appropriate to use a constant value  instead of other assumptions.

As observed, when sinks are included in the model, a certain amount of time is required for the system to reach the salinity concentrations observed. This behavior is represented in the model by an initial condition, designed as a logistic curve, which describes how the effect of the sinks manifests and evolves over a given period of time. This curve makes it possible to simulate the gradual adaptation of the system until it reaches the observed concentrations, providing a useful tool for evaluating and validating the model.

The choice of a logistic curve is justified by its ability to model gradual processes, which makes it suitable to reflect the temporal behavior of the sinks and their impact on the estuarine system. Therefore, the use of this parameter ($\delta$) is an efficient way to quantify the inflows and outflows of water from the main channel, largely due to anthropogenic activities. This approach can be extrapolated to other estuaries with excessively high salinity concentrations in the estuary interior that cannot be explained by dispersion alone. Similarly, this method can be applied to systems under high anthropogenic pressure and similar environmental conditions.

In estuaries with behavior similar to that of the Guadalquivir River, especially in low flow conditions where the tidal action dominates the hydrodynamic behavior, the omission of the anthropogenic effect may lead to an underestimation of salinity concentrations. Therefore, including these effects through the $\delta$ parameter allows for more realistic simulations and helps to understand the impact of these activities. This understanding is essential for effective estuary management, both from a socio-economic and environmental perspective.

4. **What is the basis for determining D = 0.5 m²/s? Would using other parameterization schemes for D across the entire area significantly affect the salinity intrusion?**

Thank you for the question.

We have reviewed and adjusted the parameterization used in the advection and transport module implemented in our model. In the previous version, we used a very low parameterization coefficient obtained from a sensitivity analysis to ensure that no numerical instabilities were introduced into the model. However, we agree with the reviewers that this coefficient is particularly low. We have therefore reviewed the implemented parameterization and adjusted it to allow the use of more realistic dispersion coefficients in line with the literature.

We particularly appreciate this comment as it has allowed us to improve the parameterization while maintaining the numerical stability of the model. In the new version, we have been able to increase this coefficient, which does not change the results or objectives of the study but allows a more effective analysis of the behavior of the salinity gradient in terms of horizontal dispersion. It also gives us the opportunity to check the model's configuration and improve the model parameterizations.

As discussed in the previous comment, based on the definition proposed by Bowden (1983), we have consistently calculated the coefficient from the tidal current amplitude range and the mean channel depth, thereby yielding a more physically based dispersion coefficient average to be implemented in the model.

The dispersion coefficient D was assumed to be constant in this study for several reasons, as outlined below. Primarily, the simplicity of a constant value ensures numerical stability and

facilitates the interpretation of results. This selection of a constant dispersion coefficient is based on the assumption that lateral dispersion is homogeneous and that strong currents will induce vertical mixing, thereby rendering advection the dominant process in the behavior of the salinity intrusion. The Peclet number (Pe), defined as uL/D, measures the relative contribution between the nonlinear advection and horizontal dispersion, where u is an averaged (in time and along the whole estuary) absolute value of the along-channel gradient of velocity, L is the estuary length, and D corresponds to the horizontal dispersion coefficient (Deng et al., 2024). Taking a value u= 0.5 ms$^{-1}$, extracted from realistic simulations performed with the hydrodynamic module and the values L= 107 km and D=150 m$^2$ s$^{-1}$ yields a value Pe=356, clearly indicating a dominance of the advective transport rate over the diffusive one. To understand this better, we analytically evaluate the result for the Pe corresponding to the current velocity of each campaign (MG3 is shown as an example, but the result is the same for all campaigns), for the entire modelled estuary and with a dispersion coefficient D=150m. This results in an advection dominance exceeding 90% (7-day simulation).

[Figure]

**Fig. R5.** Péclet number calculated for the MG3 campaign. The results represent a 7-day simulation of the MG3 campaign, with the Péclet number computed at each time interval (in minutes) for the hole channel as an average.

Once it has been determined that advection is the dominant process in the estuary for this specific period (low discharge regime), it can be concluded that horizontal dispersion will play a secondary role in the estuary.

Furthermore, given the lack of comprehensive data on the coefficient's variability across the estuary, it was determined that a constant value would be an adequate representation of the general conditions. Finally, model validation with observational data has demonstrated that employing a constant coefficient is an effective method for accurately reproducing the essential characteristics of the system, thereby supporting this approach within the context of the present study. Furthermore, numerous studies in the literature have demonstrated that models with a constant dispersion coefficient are capable of accurately reproducing salinity distributions (e.g., Lewis and Uncles, 2003; Brockway et al., 2006; Gay and O'Donnell, 2007, 2009; Xu et al., 2019; Siles-Ajamil et al., 2019; Biemond et al., 2024). This choice not only maintains the stability of the model, avoiding numerical instabilities, but also ensures that the results are consistent with theoretical expectations and experimental observations.

Regarding parameterizations, in this case, as mentioned, it is constant, but we agree with the reviewer that an alternative parameterization could be evaluated and perfectly feasible, such as one that is variable in time and space, as has been done in previous studies (e.g. Diez-Minguito et al., 2013). However, due to the dominance of advection, we believe that the use of an alternative parameterization would not lead to a significant change in the salinity intrusion results.

The available observations are not long enough or rich enough to allow a detailed analysis of the behavior of horizontal dispersion. However, this will be addressed in future work, where we have planned numerous sampling campaigns that will allow a more precise evaluation of the spatial and temporal behavior of horizontal dispersion in the Guadalquivir estuary.

Biemond, B., de Swart, H. E., & Dijkstra,H. A. (2024). Quantification of sal ttransports due to exchange flow and tidal flow in estuaries. Journal of Geophysical Research: Oceans, 129, e2024JC021294. https://doi.org/10.1029/2024JC021294

Bowden, K. F. (1983). *Physical Oceanography of Coastal Waters*. Ellis Horwood Series in Marine Science. E. Horwood. [University of Michigan, Digitized Nov 3, 2007]. ISBN 0853126860, 9780853126867.

Brockway, R., Bowers, D., Hoguane, A., Dove, V., and Vassele, V.: A note on salt intrusion in funnel–shaped estuaries: Application to the Incomati estuary, Mozambique, Estuar. Coastal Shelf S., 66, 1–5. https://doi.org/10.1016/j.ecss.2005.07.014, 2006.

Gay, P. and O'Donnell, J.: A simple advection–dispersion model for the salt distribution in linearly tapered estuaries, J. Geophys. Res., 112, C07021. https://doi.org/10.1029/2006JC003840, 2007.

Gay, P. and O'Donnell, J.: Comparison of the salinity structure of the Chesapeake Bay, the Delaware Bay and Long Island Sound using a linearly tapered advection–dispersion model, Estuaries Coasts, 32, 68–87. https://doi.org/10.1007/s12237-008-9101-4, 2009.

Lewis, R. E. and Uncles, R. J.: Factors affecting longitudinal dispersion in estuaries of different scale, Ocean Dynam., 53, 197–207. https://doi.org/10.1007/s10236-003-0030-2, 2003.

Xu, Y., Hoitink, A. J. F., Zheng, J., Kästner, K., and Zhang, W.: Analytical model captures intratidal variation in salinity in a convergent, well-mixed estuary, Hydrology and Earth System Sciences, 23, 4309–4322. https://doi.org/10.5194/hess-23-4309-2019, 2019.

5. **How can the impact of human pressure on salinity intrusion be quantitatively assessed based on the 1D diffusion equation in this study? Is it through its effect on advective transport or horizontal diffusive transport, thereby influencing salinity transport? Which of these two processes contributes more?**

Thank you for the question.

The quantitative assessment of the impact of human pressure on salinity intrusion in this study is carried out by introducing a sink term (parameter δ) in the hydrodynamic model, which reduces the volume of water. This has a direct effect on advection transport, as water withdrawal reduces the total volume of water, which has an effect on flow velocities (causing a slight intensification of incoming currents) and thus increases salinity intrusion. Although dispersion transport may also be affected, advective transport is the process that contributes most, as it is the primary mechanism controlling salinity movement in the estuary.

A reduction in water volume can be caused by natural processes such as evaporation. However, in light of our results and as discussed in the response to question 3, achieving the observed salt concentration along the river requires considering this reduction in volume, which cannot be attributed solely to natural processes like evaporation or small secondary channels. Modifications to the estuary due to port activities, the reduction of marshlands, water extractions for legal crops, illegal water withdrawals, the creation of new channels, etc., combined with natural processes,

are what are causing this reduction in water volume and, consequently, a greater penetration of saline intrusion.

Comparing the temporal average magnitude of advection and dispersion transport in salinity (for MG3 campaign), it can be understood that advective transport is significantly more substantial and is the dominant transport mechanism (Fig. R6).

[Figure]

***Fig. R6.*** (a) Simulated salinity concentration along the Guadalquivir Estuary for the MG3 campaign without incorporating the water volume reduction parameter. (b) and (d) show the time-averaged salinity concentration variations in each section due to advection and dispersion transport, respectively. (c) and (e) present the time-averaged percentage of salinity concentration variations in each section due to advection and dispersion transport, respectively.

Fig.R6 shows a contour plot of salinity concentration over time in all sections of the channel, with the highest concentration located at the mouth of the estuary and decreasing as the fluid moves through the channel, reaching values of 0 at the head. This corresponds to MG3 salinity simulation just including dispersion (water withdrawals ($\delta$) are not included). A temporal average has been calculated for both advection and dispersion transport to provide a representative measure over time. Advective transport dominates over dispersion transport, reinforcing the idea that the movement of salinity is primarily controlled by current flow rather than dispersion.

6. **There are three tributary estuaries in this study, but they don't seem to be marked on the figures. Additionally, how was the runoff distributed among these three estuaries? In the experiments with increased or decreased runoff, was the flow rate adjusted simultaneously for all three tributary estuaries?**

Thank you for the question.

In this study, the Alcalá del Río dam is considered the main source of freshwater discharge, contributing approximately 80% of the flow received by the estuary (Diez-Minguito et al., 2012). The remaining 20% comes from small tributaries flows. All tributary flows that discharge into the estuary were examined, selecting those that had a significant flow for the study periods. This resulted in the inclusion of two additional tributary flows, in addition to the flow from the dam.

These tree tributaries flows are included in the upper part of the estuary as they discharge close to this area. Therefore, the sum of these three flows is considered to be the freshwater input to the estuary, specifically at its head, which is indicated in Figure 1 by a black triangle marked "dam".

Thanks to your comment, we have realized that this information was not clearly defined in the manuscript. For greater clarity, the exact location of these tributaries is indicated in lines 161-165.

In the experiments where the freshwater flow is modified, all three tributaries are considered. The model incorporates a single freshwater flow from the dam, which is the sum of the three tributaries (Alcalá del Río, Rivera de Huelva and Zufre). Therefore, when a change in freshwater flow is mentioned in the experiments, it refers to this combined flow.

Hourly flow data for each of the tributaries (Alcalá del Río, Rivera de Huelva and Zufre) were obtained from the *Confederación Hidrológica del Guadalquivir* database (Guadalquivir SAIH, https://www.chguadalquivir.es/saih/, last accessed: March 25, 2024). The average flow for each tributary was calculated and the sum of the three flows was determined. The resulting total flow was then used as a constant overtime in the final part of the hydrodynamic model. The decision to use the average value for each tributary was made because the flow rates during these campaigns are very low and there is minimal difference between using the average or the exact value at each time step (with hourly data extrapolated to seconds). In order to avoid introducing unnecessary uncertainty, it was decided to treat the flows as constant over time.

This methodology was the same used in Sirviente et al., 2023, where the high reliability of the hydrodynamical model is presented.

7. **How is the fact that water withdrawal does not occur throughout the entire estuary, but at specific locations, taken into account? This localized withdrawal will also lead to a reduction in the overall runoff of the estuary. Would this have any impact on the study's results?**

Thank you for the question.

In our study, water withdrawals are consistently integrated in both time and space. As mentioned in question 3, these water volume reductions are accounted for by the parameter $\delta$, which is present at all dt and in dx of the estuary. However, in certain locations, depending on the season analyzed (MG1, MG2, MG3 or MG4), they have a higher $\delta$ than the rest of the sections. The fact that some campaigns show a higher $\delta$ in the first 20 km can be attributed to the presence of marshes in the Doñana Natural Park. Similarly, the use of a higher $\delta$ between km 30-70 can be explained by the presence of agricultural fields.

These withdrawals will affect the total volume of the estuary, which, given the very low freshwater flows, will be compensated by saline water due to volume conservation, resulting in an increase in saline intrusion in the estuary.

8. **How is the water withdrawal process represented in the governing equations? In other words, how is the dynamic process of water withdrawal parameterized in the governing equations?**

Thank you for the question.

The process of water withdrawal is represented by the parameter $\delta$, which represents the thickness of a water slice that could be removed from the estuary at each integration time step $\Delta t$. It is implemented into the continuity equation in the following way:

$$b \frac{\partial \eta}{\partial t} = \frac{\partial}{\partial x}[Au] - \frac{b \, \delta}{\Delta t}$$

Where b is the width of the channel. Note that the second term on the r.h.s. of that equation represents a loss of volume per unit of channel length and unit time. The action of this term is translated, through the numerical integration of this equation along with the momentum balance equation, into the corresponding sea level and current velocities variations in order to compensate these volume losses. All this give arise to the creation of a mean net transport directed towards the head of the estuary that promotes the penetration of the saline front more and more inwards while the volume losses are maintained.

9. **In the introduction, could you add some related studies on the impact of human activities on salinity transport in other estuaries?**

Yes, thanks for the suggestion. We have added some related studies of other estuaries in the Introduction section. We have modified lines 65-80 adding this paragraph:

"The detrimental effects of anthropogenic activities have been demonstrated in other estuaries around the world. Alcérreca-Huerta et al. (2019) show that the construction of a dam system in the estuary of the Grijalva River (Mexico) in 1959 altered the hydrological regime, reducing the seasonality of water discharge and decreasing the amount of available freshwater. This, together with changes in land use (more agricultural land, less mangrove cover and less vegetation), leads to variations in salinity concentration, with saline intrusion observed up to 46 km upstream, with salinity levels reaching 32.8 PSU. Studies such as Huang et al. (2024), based on numerical simulations using a 3D model, show that anthropogenic activities, in particular the regulation of freshwater flows by infrastructure projects, are drastically changing the dynamics of saline intrusion in the Changjiang River estuary (China). This study shows how an increase in freshwater flows (due to releases from the Three Gorges Reservoir) counteracts the advance of saline intrusion. However, water withdrawals in the city of Yangzhou as part of the implementation of the East Route of the South-to-North Water Transfer Project will inevitably lead to a reduction in inflow during the dry season, resulting in an increase in salinity intrusion in this system by approximately 6-7 km. This relationship between salinity and freshwater flow was also observed by Webber et al. (2015) in Yangtze River Estuary (China), who assessed the effects of the Three Gorges Dam, the South-to-North Water Transfer Project, and local water withdrawals on the probability of intrusion in the Changjiang River estuary. They conclude that these projects will increase the probability of saline intrusion and suggest that water management should be adapted to mitigate the risk."

---

## Author Comment (AC4)

**#Reviewer 2**

We thank the reviewers for their valuable comments, which have been essential in improving our study. In response to feedback from several reviewers, we have optimized the parameterization of the advection-dispersion module, allowing us to use a more realistic dispersion coefficient. In the new version of the article, as detailed in the responses to the questions, we have provided a more precise explanation of the dispersion coefficient used. Additionally, a new section (Section 3.1) has been added, where we justify in detail the need to include sinks, as well as provide a thorough description of the sink parameter. All the results presented in the article have been updated to reflect the new simulations performed with the improved model version. Although these new simulations do not alter the results or discussion presented, they contribute to greater clarity and precision in the presentation of the findings.

Thanks to reviewer 2 for their comments and suggestions, and for taking the time to review the manuscript. Please find below the answers:

1. **The authors will need to justify better the use of an 1D model for salt intrusion as this neglects the effect of vertical salinity gradients which contribute to salt intrusion. In this case, a constant diffusion coefficient is not enough to account for any unresolved mixing. In addition, Figure 1b and d show that both the width and the depth of the channel can be significant and so it is dubious if averaging can be justified. Have the authors considered the use of a 2DV model instead?**

Thank you for your comment.

The bathymetric data we used, based on the 2019 nautical chart provided by the Spanish Navy Hydrographic Service (the most recent available), show that the depth varies from 5 to 18 m and the width from 100 to 400 m. This configuration indicates that the cross-sectional areas of the estuary do not show significant variations, considering that the total length of the estuary is 107 km. This allows a simplification of the equations to a 1D model.

Similarly, under conditions of low freshwater flow, the hydrodynamic behavior of the estuary is dominated by the tidal influence (Diez-Minguito et al., 2012). This results in minimal variation in the transverse direction (perpendicular to the flow). The inflow predominantly follows a longitudinal direction, and the transverse variations are small enough to adequately represent the hydrodynamic behavior of the estuary.

This aspect is supported and demonstrated by the hydrodynamic validation presented by Sirviente et al. (2023), which shows an extensive and thorough validation of the 1D hydrodynamic module, illustrating its ability to simulate tidal height and current velocity with high reliability. This study shows how the 1D model is able to reproduce both surface and depth observations from moored current meters. The high reliability of the model indicates that there are no significant velocity variations at different depths, suggesting that vertical mixing is sufficiently strong. This supports the idea that a 1D model is appropriate.

The variability in depth and width is accounted for as the bathymetry is inherently incorporated into the model and averages are used for each section. The effectiveness of this approach is demonstrated in Sirviente et al. as well as in the strong correlations obtained in the present study.

In terms of vertical salinity structure, the estuary is primarily characterized by intense mixing that prevents the formation of significant vertical salinity or temperature gradients (Diez-Minguito et

al., 2012), resulting in a homogeneous distribution of water properties. In other words, a well-mixed estuary is defined by uniform salinity mixing due to strong tidal currents, which prevents stratification. Similarly, the vertical circulation shows a relatively uniform flow pattern during low-flow periods, with no significant variations in flow velocity or direction at different depths (Losada et al., 2017). These conditions allow the hydrodynamic equations to be simplified to those of a one-dimensional channel, as validated in previous studies of the Guadalquivir estuary (Álvarez et al., 2001; Siles-Ajamil et al., 2019; Sirviente et al., 2023). The vertically homogeneous salinity behavior is further demonstrated in the answer to the following question and the accompanying figures (Reviewer comment 2).

Regarding the ability of the 1D model to capture the effects of vertical salinity gradients and unresolved mixing processes, it is important to note that in systems characterized by intense mixing and uniform vertical circulation, such as the GRE, a one-dimensional model can adequately represent the hydrodynamic behavior and salinity distribution without necessarily requiring a variable diffusion coefficient.

Thanks to the reviewers' comments, we reviewed and adjusted the transport model to use a higher, more realistic constant diffusion coefficient without introducing instabilities into the model. In the revised version, based on Bowden (1983), we defined the most appropriate horizontal dispersion coefficient, taking into account the mean depth and tidal amplitude. This calculation was performed for all campaigns included in the analysis to ensure the use of a constant dispersion value appropriate for the system. The results indicate that the maximum constant dispersion based on velocity and depth is 150 m²/s (this has also been added and explained in detail in the revised version of the article). In addition, a sensitivity analysis was performed with different dispersion coefficients to optimize this parameter as much as possible. It was found that a dispersion value higher than 200 m²/s leads to numerical instabilities in the system.

Based on Bowden (1983), the effective horizontal dispersion coefficient can be calculated as $K_X = U^2 H^2 / 30 * K_z$, where $U$ is the maximum tidal velocity, $H$ is the mean channel depth, and $Kz$ is the vertical eddy dispersion coefficient, assumed to be constant. In our case, we used $Kz = 0.01$, as proposed by Bowden (1983).

| Campaign | U (ms$^{-1}$) | K$_X$ m$^2$s$^{-1}$ |
|----------|---------------|---------------------|
| MG1 | 0,85 | 143 |
| MG2 | 0,88 | 154 |
| MG3 | 0,88 | 153 |
| MG4 | 0,80 | 127 |

Furthermore, given the lack of comprehensive data on the coefficient's variability across the estuary, it was determined that a constant value would be an adequate representation of the general conditions. Finally, model validation with observational data has demonstrated that employing a constant coefficient is an effective method for accurately reproducing the essential characteristics of the system, thereby supporting this approach within the context of the present study. Furthermore, numerous studies in the literature have demonstrated that models with a constant dispersion coefficient are capable of accurately reproducing salinity distributions (e.g., Lewis and Uncles, 2003; Brockway et al., 2006; Gay and O'Donnell, 2007, 2009; Xu et al., 2019; Siles-Ajamil et al., 2019; Biemond et al., 2024). This choice not only maintains the stability of the

model, avoiding numerical instabilities, also ensures that the results are consistent with theoretical expectations and experimental observations.

Regarding the use of a 2D model, although we have not explicitly applied a 2D model to the Guadalquivir estuary, we have analyzed its analytical solution. This approach allows us to observe that the differences in velocity in the longitudinal direction of the estuary are not significantly different from those obtained with the 1D version.

In Figure R1, we present the analytical solution for a channel-cross section using a 2D model that includes the channel width and a parabolic depth variation that approximates the change in depth from the lateral boundaries to the center of the channel. For comparison, we also include the 1D solution for the same channel (length = 5 km, width = 525 m, 2-day simulation) with an average depth of 6.7 m.

As shown, there are differences at the lateral boundaries, within the first 100 m on either side of the channel. However, the oscillations are not significant, and the velocities are very similar across most of the channel width. This shows that the behavior is generally homogeneous, validating the 1D solution. This conclusion is confirmed by the average velocities obtained for each solution at different times (Table R1), where the differences between the average velocities of the two models are minimal.

It is true that using the 1D solution slightly underestimates the velocity in the center of the channel and slightly overestimates it at the boundaries. However, these discrepancies do not affect the results, as the model has been shown in Sirviente et al. (2023) to reproduce the observations with high reliability.

This reinforces the idea that the use of a 2D model does not provide a substantial improvement over the 1D model.

[Figure]

**Figure R1.** The top panels display the time series of the longitudinal velocity (uu) for section 20 of the channel. Colored markers highlight the three consecutive hours analyzed in the bottom panels. The bottom panels compare the velocity profiles obtained from the 2D model (solid lines) with those from the 1D model (dashed lines) at the selected times, illustrating the differences across the channel width.

**Table R1.** Average Velocity in Section 20 of the Idealized Channel for 2D and 1D Simulations Over a 6-Hour Period (3 Hours of Flood Tide and 3 Hours of Ebb Tide)

|  | u average 2D (ms⁻¹) | u average 1D (ms⁻¹) |
|---|---|---|
| *Flood Hour 1* | -0.75 | -0.86 |
| *Flood Hour 2* | -0.52 | -0.61 |
| *Flood Hour 3* | -0.16 | -0.21 |
| *Ebb Hour 1* | 0.82 | 0.89 |
| *Ebb Hour 2* | 0.62 | 0.69 |
| *Ebb Hour 3* | 0.26 | 0.32 |

We appreciate the reviewer's suggestion and will take it into account for future work.

Biemond, B., de Swart, H. E., & Dijkstra,H. A. (2024). Quantification of salttransports due to exchange flow and tidalflow in estuaries. Journal of GeophysicalResearch: Oceans, 129, e2024JC021294. https://doi.org/10.1029/2024JC021294

Bowden, K. F. (1983). *Physical Oceanography of Coastal Waters*. Ellis Horwood Series in Marine Science. E. Horwood. [University of Michigan, Digitized Nov 3, 2007]. ISBN 0853126860, 9780853126867.

Brockway, R., Bowers, D., Hoguane, A., Dove, V., and Vassele, V.: A note on salt intrusion in funnel–shaped estuaries: Application to the Incomati estuary, Mozambique, Estuar. Coastal Shelf S., 66, 1–5. https://doi.org/10.1016/j.ecss.2005.07.014, 2006.

Gay, P. and O'Donnell, J.: A simple advection–dispersion model for the salt distribution in linearly tapered estuaries, J. Geophys. Res., 112, C07021. https://doi.org/10.1029/2006JC003840, 2007.

Gay, P. and O'Donnell, J.: Comparison of the salinity structure of the Chesapeake Bay, the Delaware Bay and Long Island Sound using a linearly tapered advection–dispersion model, Estuaries Coasts, 32, 68–87. https://doi.org/10.1007/s12237-008-9101-4, 2009.

Lewis, R. E. and Uncles, R. J.: Factors affecting longitudinal dispersion in estuaries of different scale, Ocean Dynam., 53, 197–207. https://doi.org/10.1007/s10236-003-0030-2, 2003.

Xu, Y., Hoitink, A. J. F., Zheng, J., Kästner, K., and Zhang, W.: Analytical model captures intratidal variation in salinity in a convergent, well-mixed estuary, Hydrology and Earth System Sciences, 23, 4309–4322. https://doi.org/10.5194/hess-23-4309-2019, 2019.

2. **There is an inconsistency in the terminology. In some instances, the authors refer to salt intrusion and in others to salt wedge or even salt front and it seems they don't distinguish between these terms. I would advise to remain consistent throughout the manuscript and give an explicit definition. Salt intrusion is usually measured as the landward penetration of a bottom isohaline while the salt wedge is defined as a bottom layer of denser than the surface water. Consequently, I reckon that what is seen in the figures is rather the salinity horizontal gradient (or salinity front) instead of salt intrusion or wedge. Furthermore, the model results are compared with observations taken at 2m below the surface, but the depth can be much deeper in certain sections as it can be seen in Figure 1b. Therefore, I think it is possible that the discrepancy observed between model results and observations without the sinks may be due to the depth averaging which may moderate higher bottom salinity.**

We appreciate the reviewer's comment and agree that the terminology used throughout the manuscript was inconsistent. We have corrected this by replacing all related terms with "salt/saline intrusion," as we believe that in a well-mixed estuary, this is the most appropriate term to describe the extent of saltwater moving upstream. We have also replaced it with "horizontal

salinity gradient" when referring to longitudinal variations in salt concentration, as this term describes how salinity changes as one moves horizontally through the estuary. We believe these corrections will clarify the use of these terms and avoid potential confusion.

We apologize for the errors in the original wording.

Salt/Saline intrusion refers to the upstream movement of saltwater into freshwater areas, particularly during low river flow or high tides.

Horizontal salinity gradient: Variation in salt concentration in the water over a horizontal distance in the estuary.

**"Therefore, I think it is possible that the discrepancy observed between model results and observations without the sinks may be due to the depth averaging which may moderate higher bottom salinity. "**

We appreciate the comment, but we believe that in this particular case, the observed discrepancy between the model results without the sinks and the observations is not due to the depth averaging.

The estuary we are modelling is characterized by being well mixed, with minimal vertical salinity gradients. In this type of system, vertical mixing is strong enough to maintain a relatively homogeneous salinity throughout the water column. Therefore, we believe that the depth averaging used in the model is an adequate representation of the actual conditions in the estuary, as it reflects the homogeneous nature of the salinity distribution observed in the field.

It should be noted that this is valid in our study because we are in low freshwater discharge conditions (all the campaigns analyzed in this study). Under high discharge conditions, this approach would not be valid because the water column would not be perfectly mixed. Therefore, in this study, we only analyze the low flow cases which correspond to approximately 80% of the year.

We appreciate your comment and understand the concern about the potential impact of depth averaging in systems with more pronounced vertical gradients. However, in this particular case, we believe that the current approach is valid for representing salinity conditions in the well-mixed estuary.

In the validation of the hydrodynamic model (see Sirviente et al., 2023), observations were validated with both state harbor tide gauges, which measure at the surface, and current meters, which are anchored at various points. The high reliability of the simulations with all observations indicates that the 1D model is able to reproduce $u$ and $\eta$ with a high degree of accuracy, thus demonstrating that the model approach of using depth averages in this way is appropriate. This indicates that there is homogeneity in velocity, meaning that there are no significant gradients causing stratification, which means that the estuary is well mixed.

To understand this in terms of salinity, since there is no evidence of velocity stratification, we can assume that vertical salinity gradients are minimal and that salinity concentrations are homogeneous throughout the water column. Therefore, the comparison between simulations and observations is appropriate, and the differences between simulations without sinks and observations are not due to depth averaging.

The vertical mixing and the reduced salinity gradient are evidenced by the data recorded by the CTD during each campaign. CTD profiles were performed at the different sampling points shown

in Figure 1 of the manuscript, allowing us to analyze the vertical behavior of the salinity. The figures below correspond to the vertical salinity profiles measured during the MG1, MG2 and MG3 campaigns.

For MG3, CTD profiles were taken at 1-hour intervals over two tidal cycles, for a total of 25 hours at each site. For the MG1 and MG2 campaigns, measurements were taken at 1-hour intervals for a maximum of 10 hours. The sampling points for MG3 are the same as those shown in Figure 1 of the manuscript, as is the case for MG2. However, for MG1 there are more CTD sampling points than shown in the figure. The points for MG1 shown in the CTD plots correspond exactly to the positions of MG2 (see Figure 1a of the manuscript). It should be noted that there are no vertical profiles available for MG1-1 and not all sampling stations have 10 profiles.

When analyzing the behavior of the vertical profiles, it can be observed that in MG3 the water column always remains mixed, showing only very slight vertical salinity gradients during certain hours. The MG3 figure shows that vertical mixing prevails during the tidal cycles. Similarly, during the MG2 and MG1 campaigns, a strong vertical mixing is observed throughout the water column at all CTD profiles in all points of the river.

This observation was essential in simplifying our approach, allowing us to adopt a one-dimensional (1D) model, assuming that the salt concentration is homogeneous throughout the water column. We would like to emphasize that this strong mixing behavior is not always characteristic of the estuary. This intense mixing occurs under low discharge regimes, where freshwater flow is minimal and tidal dominance is evident (as in our study case). In scenarios with moderate or high discharge regimes, the vertical salinity gradient behaves differently, leading to stratification of the water column. Under such conditions, this model cannot be applied as it would underestimate the salinity levels.

[Figure]

**Fig. R1**. Top panel corresponds to the tidal current velocity at each sampling station during the MG3 campaign, with different colors indicating the tidal phases during which each CTD profile was taken. The

bottom panel displays the CTD profiles at each sampling point along the Guadalquivir River during the MG3 campaign.

[Figure]

**Fig. R2.** CTD profiles at each sampling point along the Guadalquivir River during the MG2 campaign.

[Figure]

**Fig. R3.** CTD profiles at each sampling point along the Guadalquivir River during the MG1 campaign.

These figures will be added to the manuscript as supplementary material SM1, SM2 and SM3 (only the points coinciding with the thermosalinograph fixed stations and presented in figure 1a of the manuscript). We will adjust the nomenclature of the other supplementary material figures in the text to ensure that they are in the correct order. We have added these lines to the text:

Line 180 :

"CTD profiles obtained at the same sampling stations as the thermosalinograph data for campaigns MG1, MG2 and MG3 shown in Figure 1a are also used. These profiles are used to analyze the vertical behavior of the water column."

Line 275:

"The salinity profiles obtained from the CTDs show a strong vertical mixing of the water column throughout the whole period and at all points (see Figures SM1, SM2 and SM3). Very reduced vertical salinity gradients can be observed, which allows us to conclude that the vertical behavior of the water column in the GRE is homogeneous under these conditions, allowing the use of a 1D model to simulate the salinity concentration along the river."

> 3. **In continuation to the previous comment. The authors assume that the salinity deficit in their uncalibrated model is exclusively due to water withdrawals. I appreciate that this is an important parameter and even more true for this specific study case, but I believe that the assumption neglects all the other complex physical processes and mechanisms taking place in an estuary. The authors already mention in their manuscript tidal amplification and channel deepening. Don't these two also account for an upstream increase in salinity?**

We appreciate your comments and observations. Indeed, the increase in channel depth and tidal amplification in the upper estuary will contribute to the increase in salinity. However, in our simulations we analyze the actual salinity concentration at the time of the campaign, which already accounts for tidal amplification, and we use the most recent actual depth available, which allows us to account for both processes.

These two factors can explain an increase in salinity upstream over time, as the channel geometry changes. But we are convinced that these are not the only factors that play a role in the increase of salinity in the estuary. Rather, freshwater flow and withdrawals from anthropogenic activities play a fundamental role.

Therefore, we believe that the salinity deficit between the model without sinks and the observations is mainly due to the lack of consideration of the parameter $\delta$. The parameter $\delta$ is a key factor in quantifying the effect of extractions and small contributions of water along the river. In other words, this factor allows us to consider all activities related to the reduction or increase of water volume in the channel (extractions for domestic use, agricultural use, industry, small channels that flow into the estuary, etc.).

It represents an average value between these two actions and has been shown to be positive for our study, indicating that, on average, the extractions exceed the contributions of water to the main channel from the small channels that converge in it. Without the inclusion of $\delta$, we observe that the horizontal salinity gradient is considerably lower than that recorded with the observations.

To clarify and enhance the explanation, we have added a subsection within the *Results* section corresponding to Section 3.1, where we demonstrate the necessity of including sinks to achieve the salinity concentrations observed.

If we analyze the behavior of the horizontal salinity gradient along the estuary, taking into account only the horizontal dispersion, we can see that the system would never reproduce the observed salinity concentrations in the different campaigns. Even when the dispersion coefficient is increased to 190 m²/s, the behavior remains the same. Over time, we observe that salinity in the inner part of the estuary tends to increase; however, it never reaches the observed concentration

levels. Therefore, it can be said that horizontal dispersion alone does not achieve the high salinity values observed along the channel, which opens the hypothesis that some additional effect is likely to cause a greater penetration of saline intrusion into the estuary. (Fig R4 (a) & (b))

If the same experiment is carried out but the parameter $\delta$ (representing all the processes that reduce the volume of water in the estuary) is included (Fig. R4 c-f), the results show that the system reaches the observed salinity values over time. This shows that this term must be included in order to reproduce the salinity concentrations observed in the different campaigns.

Figures R4c and R4d present the experiments that include water withdrawals as a constant value in time and space ($\delta = 0.005$ mm). As shown, as the simulation time progresses, the system achieves the salt concentrations range presented in the observations. This contrasts with the previous cases, where only the dispersion term was included in the experiments (Figures R4a and R4b), and the observed range could not be achieved.

On the other hand, Figures R4e and R4f show the experiments employing time-varying $\delta$. In this simulation, a stronger sink is applied during the first three days ($\delta = 0.01$ mm), which is then reduced and held constant for the remainder of the simulation ($\delta = 0.001$ mm). In this case, the obtained values closely match those recorded in the observations.

These experiments highlight the necessity of including this term ($\delta$) in the simulations, as otherwise, the horizontal salinity gradient would never reach the observed values. As illustrated in the figure, a certain duration of sink activity is required for the simulated salinity concentrations to approach the range observed. Therefore, it is essential to define an initial condition that considers this progression, allowing the simulation to adequately capture the evolution of the water withdrawals and their impact on salinity over time. This is particularly justified by Figures 4e and 4f, where using a stronger $\delta$ during the first three days followed by a weaker $\delta$ reproduces the observed behavior.

[Figure]

**Fig. R4.** Comparison of simulated (temporal behavior of the simulation at the corresponding observation points) and observed salinity over the MG3 vessel trips: (a) and (b) show simulations including only the horizontal dispersion for the MG3 vessel trip upstream (a) and downstream (b), with the observations in black. (c) and (d) show simulations incorporating δ term (δ) for the entire simulation period as constant value, and (e) and (f) are the simulation including a time-varying δ, compared to the observations (in black) for the MG3 vessel trip upstream (c and e) and downstream (d and f).

The δ value was determined empirically during the calibration process, through a sensitivity analysis in which different δ values were tested in simulations to identify the one that produced concentrations within the observed range while maintaining the temporal and spatial stability of the model.

Once the δ value was identified, experiments were carried out to analyze its behavior. These included constant use of the parameter over time, and experiments including them at specific time intervals, as well as spatial distribution experiments where δ was applied to specific points (e.g. high areas of the river) and regions. However, due to the limited understanding of the true behavior of these processes, and to avoid introducing assumptions or speculation that could affect the validity of the results, it was considered more appropriate to use a constant value rather than additional assumptions.

As observed, when sinks are included in the model, a certain amount of time is required for the system to reach the salinity concentrations observed. This behavior is represented in the model by an initial condition, designed as a logistic curve, which describes how the effect of the sinks manifests and evolves over a given period of time. This curve makes it possible to simulate the gradual adaptation of the system until it reaches the observed concentrations, providing a useful tool for evaluating and validating the model.

The choice of a logistic curve is justified by its ability to model gradual processes, which makes it suitable to reflect the temporal behavior of the sinks and their impact on the estuarine system. Therefore, the use of this parameter ($\delta$) is an efficient way to quantify the inflows and outflows of water from the main channel, largely due to anthropogenic activities.

To evaluate the effects derived from the tidal wave amplification at the head and the increase in depth, simulations using historical bathymetries, as presented in Sirviente et al. (2023), should be conducted. Similarly, observations from those periods are needed to understand the salinity concentration in the estuary at that time. Future studies aim to assess these effects by conducting experiments that modify the channel geometry, following the experimental approach outlined in Sirviente et al. (2023).

Nevertheless, the fact that these factors may contribute to the upstream salinity increase does not justify the observed discrepancies between the simulation without sinks and the observations. The aim of this article is to highlight, through these numerical simulations, the pressure that anthropogenic activities exert on the salinity concentration in the estuary.

In the following figure (Fig. R5) we present an experiment where we simulated the observations of MG1, using a constant depth and width channel, without including the parameter $\delta$. As you can see, there is a slight variation compared to the simulation where we used the actual depth of the estuary (simulation without $\delta$). This is of particular interest because it allows us to roughly observe the effect of increasing the river depth; however, it is unable to reproduce the salinity concentrations observed during these campaigns

[Figure]

**Fig. R5.** Observed horizontal salinity gradient during the MG1 campaign (September 2021). (b) Modeled horizontal salinity gradient using the 2019 nautical chart bathymetry. (c) Modeled horizontal salinity gradient for the MG1 campaign using a constant bathymetry (depth = 6m, width = 100m).

4. **The salt transport module was run for the periods when observations from the measurement campaigns that took place between 2021-2023 where available but the hydrodynamic model is forced with data from 2019! How is this justified? This could be already a source of errors.**

Thank you for your comment.

It is important to clarify that the hydrodynamic model is not forced with data from 2019. In reality, the model is forced at the mouth with the predicted sea level corresponding to the date of each campaign. This prediction is based on the tidal harmonics from the Bonanza tide gauge for the entire year 2019, selected because they were the most recent available at the time of designing

and implementing the model in the estuary in early 2022. The data used comes from the Bonanza tide gauge of *Puertos del Estado*, and currently, if one accesses the database, the latest complete data available is from 2022.

The decision to use the 2019 data is based on the fact that the discrepancies between the two series are minimal and will not generate significant changes in the simulations (as demonstrated below).

Fig. R6 presents the series of differences between the sea level predictions for MG3 (13 days of July 2022) using the harmonics of 2019 and 2022. As can be seen, the differences between both series are small and statistically non-significant.

Furthermore, in light of the validation of the hydrodynamics presented in Sirviente et al. (2023), it is evident that the predictions calculated for the corresponding dates, based on the 2019 harmonics from the Bonanza tide gauge, allow for the generation of reliable simulations. All of this reinforces our hypothesis that using the 2019 harmonics is valid and does not introduce a significant error into our study.

Furthermore, using the harmonics of 2019 allows us to maintain the same methodology employed in Sirviente et al., 2023. Where the high reliability of our hydrodynamical model is presented.

[Figure]

**Fig. R6**. Observed horizontal salinity gradient during the MG1 campaign (September 2021). (b) Modeled horizontal salinity gradient using the 2019 nautical chart bathymetry. (c) Modeled horizontal salinity gradient for the MG1 campaign using a constant bathymetry (depth = 6m, width = 100m).

**Minor comments**

1. **I understand the notation used throughout the manuscript as km 60, km 40 etc. but it doesn't read very well. It is better if it is written as 60 km from the mouth, 40 km from the mouth etc.**

Thank you for the suggestion, we have modified the notation following your recommendation.

2. **Please use superscript numbers when giving units (e.g., lines 48, 50 ,197 etc.)**

Done, thank you

**3. Where are the river flows implemented?**

These tributaries flows (river flows) are included in the upper part of the estuary (last section) as they discharge close to this area. Therefore, the sum of these three flows is the freshwater input to the estuary, specifically at its head point, which is indicated in Figure 1 by a black triangle marked "dam".

Thanks to your comment, we have realized that this information was not clearly defined in the manuscript. For greater clarity, the exact location of these tributaries is indicated now in lines 161-165.

**4. It is implied that there is no freshwater input from the upstream boundary which is set at the dam. Is this realistic? Is it true for every season?**

As mentioned in the previous response. We have included three sources of freshwater in the main channel. These inputs are summed in the last section of the model, which corresponds to the head.

Of these three inputs, the most significant flow comes from the Alcalá del Río dam. However, we have also decided to include the contributions from the tributaries Rivera de Huelva and Zufre, despite their relatively small discharges. This decision was made to represent the freshwater input as realistically as possible. Other tributary flows that enter the main channel were not considered, as their discharges during the simulated time intervals were negligible.

The discharge data are obtained from the database of the *Confederación Hidrológica del Guadalquivir* (Guadalquivir SAIH, https://www.chguadalquivir.es/saih/, last accessed: March 25, 2024). This information allows us to demonstrate that our experiments were conducted under low flow conditions, as they present discharges of Q < 40 m³/s.

We understand that all relevant flows should be included, except for those that present a negligible volume for the time interval being simulated.

**5. In Line 90, I think the authors of this paper refer to salt intrusion length and not duration.**

We appreciate your comment and apologize for the error. Indeed, we intended to refer to the length of salt intrusion, but the wording of the original sentence could lead to confusion. We have corrected the error and modified the sentences as follows:

"Their results indicate that the mean length of salt intrusion would increase by approximately 8% under the expected scenario of a 15% decrease in freshwater discharge over the next 15 years."

Thank you for the comment.

**6. There is a confusion in the manuscript. In some instances, the authors write that the maximum salt intrusion corresponds to the flood and in others to the ebb tide. For example:**

**Lines 324-325 the authors write ' The maximum and minimum extent of the saline wedge within the channel coincided with moments just before high and low tides respectively'. In the next paragraph they write ' during the flood tide the wedge demonstrates minimal intrusion in the estuary ...... during the ebb tide, the maximum saline intrusion occurred'.**

**Line 375-376 ' the maximum ebb current and the maximum flood current which closely correspond to the maximum and minimum salt wedge intrusion, respectively'.**

**But then a few lines further down:**

**Line 380 ' During maximum ebb current (just after low tides), when minimum salt wedge intrusion occurs......during flood tides (just after high tides), the maximum salt intrusion is present'.**

**In the legend of Figure 5 'The solid lines represent the time of maximum salinity (F,Flood) and the dashed lines represent the time of minimum salinity (E,Ebb).'**

**At least, Figure 4a shows that the maximum salinity corresponds to the flood tide which is reasonable for a well-mixed estuary.**

Thank you for your comment; you are absolutely correct, and we sincerely apologize for the error in the text. As you mentioned, the maximum (minimum) penetration occurs during the high (low) water slack. However, it is important to note that there is a time lag of 1.5 hours between the two series. We appreciate the reviewer pointing out this oversight.

To improve the precision of the analysis and ensure greater clarity in the results, both the figure and the text of section 3.3 have been revised. This section is now presented as section 3.4, utilizing the updated model configuration. Moments of spring and neap tides have been selected instead of the intermediate tides previously used, enabling a more accurate and comprehensible analysis.

The section has been rewritten in alignment with all the reviewers' comments.

**"3.4. Tidal cycle dynamics.**

Once the reliability of the model had been confirmed by the results of the experimental validation presented in the previous sections, it was used to simulate the dynamic of the salinity intrusion during a spring-neap tidal cycle. To do this, we conducted a simulation extended over 15 days (15/07/2022-30/07/2022) using the same model configuration presented in section 3.1 for the MG3 campaign. This period was selected because it comprised records of observations distributed throughout the spring-neap tidal cycle, allowing for the validation of the simulations.

We focused on two 24-hour periods to describe the dynamics of the horizontal salinity gradient during different phases of the semi-diurnal cycle (Fig. 4a). A lag of about 1.5 h was observed between tidal height and salinity profiles (Fig. 4a), which means that the maximum and minimum salinity concentration values coincide with high water slack moments and with low water slacks moments, respectively. Fig. 4b and d, shows a gradual decrease in salinity values upstream.

[revised manuscript text omitted]

We hope now it is more clear and there is not option to be confused.

7. **The term 'salt wedge intrusion' is not right. It is either salt intrusion or salt wedge, not all together.**

Thank you for comment, we have modified each term following comment 2 (Major comments section)

**8. Figure 3, indicate where km 30 , 40, 50 etc. is**

Done. Figure 3 has been corrected including your suggestion.

[Figure]

**9. Line 323-324 what do you mean 'a gradual decrease in salinity values upstream can be seen' . Do you mean gradual decrease during neap tide?**

No, the sentence refers to Figure 4. However, it is true that the way the sentence is worded is confusing. We meant to say that if you look at Figure 4a you can see a 1.5 hour lag between tidal height and salinity, and if you look at Figures 4b and 4c you can see a gradual decrease in salinity concentration from the mouth to the head of the estuary (along the channel) . We have added references to the figures to improve the clarity of the text.

"A lag of about 1.5 h was observed between tidal height and salinity profiles (Fig. 5a), which means that the maximum and minimum salinity concentration values coincide with moments just before high and low tide, respectively. Fig. 5b and d, shows a gradual decrease in salinity values upstream."

---

## Author Comment (AC5)

**#Reviewer 3**

We thank the reviewers for their valuable comments, which have been essential in improving our study. In response to feedback from several reviewers, we have optimized the parameterization of the advection-dispersion module, allowing us to use a more realistic dispersion coefficient. In the new version of the article, as detailed in the responses to the questions, we have provided a more precise explanation of the dispersion coefficient used. Additionally, a new section (Section 3.1) has been added, where we justify in detail the need to include sinks, as well as provide a thorough description of the sink parameter. All the results presented in the article have been updated to reflect the new simulations performed with the improved model version. Although these new simulations do not alter the results or discussion presented, they contribute to greater clarity and precision in the presentation of the findings.

Major comments:

1. **The study is based on assumptions which need to be clearly stated in section 2. Please mention how processes such as vertical mixing at the edge of the salinity front, which can significantly influence salinity distribution across the estuary, are accounted for in the model. Include a discussion of the vertical structure of the salt wedge and related citations in the Introduction. The model is validated using salinity data collected at 2 m depth. Salt intrusions could be happening at deeper depths, which seem to be unaccounted for in this study. Please justify.**

Thank you for the comment.

We have added a paragraph in the Introduction that discusses the vertical salinity structure in the GRE. Lines 93-103 (revised manuscript version)

"The vertical salinity structure in the Guadalquivir estuary is characterized by intense mixing that prevents the formation of significant gradients in salinity and temperature, resulting in a homogeneous distribution of water properties (García-Luque et al., 2003; Diez-Minguito et al., 2012). The low average flow of the river, combined with the high tidal prism resulting from the wide tidal range and shallow channel depth, contributes to the Guadalquivir estuary being a well-mixed estuary with very low vertical gradients in salinity and temperature (García-Lafuente et al., 2012). This type of well-mixed estuary is characterized by a uniform distribution of salinity, facilitated by strong tidal currents that prevent stratification. Similarly, the vertical circulation shows a relatively uniform current pattern during low-flow periods, with no significant changes in velocity or flow direction at different depths (Losada et al., 2017). The hydrodynamic model presented by Sirviente et al. (2023), validated with different observations recorded both at the surface and at depth, shows that there are no significant variations in velocities at different depths; therefore, the presence of significant stratification is unlikely, which favors the homogeneous distribution of salinity in the water column."

In well-mixed estuaries such as the Guadalquivir, during periods of low discharge (which occurs for most of the year and under which all the simulations in this study were performed), one can assume a homogeneity of salinity concentration in the water column; that is, the vertical salinity gradients will be very small. This allows us to use our 1D model and validate it with the observations obtained from the ship's thermosalinograph.

This can be demostrated attending to the CTD porfiles recorded in our campaings.

The vertical mixing and the reduced salinity gradient are evidenced by the data recorded by the CTD during each campaign. CTD profiles were performed at the different sampling points shown in Figure 1 of the manuscript, allowing us to analyze the vertical behavior of the salinity. The figures below correspond to the vertical salinity profiles measured during the MG1, MG2 and MG3 campaigns.

For MG3, CTD profiles were taken at 1-hour intervals over two tidal cycles, for a total of 25 hours at each site. For the MG1 and MG2 campaigns, measurements were taken at 1-hour intervals for a maximum of 10 hours. The sampling points for MG3 are the same as those shown in Figure 1 of the manuscript, as is the case for MG2. However, for MG1 there are more CTD sampling points than shown in the figure. The points for MG1 shown in the CTD plots correspond exactly to the positions of MG2 (see Figure 1a of the manuscript). It should be noted that there are no vertical profiles available for MG1-1 and not all sampling stations have 10 profiles.

When analyzing the behavior of the vertical profiles, it can be observed that in MG3 the water column always remains mixed, showing only very slight vertical salinity gradients during certain hours. The MG3 figure shows that vertical mixing prevails during the tidal cycles. Similarly, during the MG2 and MG1 campaigns, a strong vertical mixing is observed throughout the water column at all CTD profiles in all points of the river.

This observation was essential in simplifying our approach, allowing us to adopt a one-dimensional (1D) model, assuming that the salt concentration is homogeneous throughout the water column. We would like to emphasize that this strong mixing behavior is not always characteristic of the estuary. This intense mixing occurs under low discharge regimes (> 70% of the year), where freshwater flow is minimal and tidal dominance is dominant (as in our study case). In scenarios with moderate or high discharge regimes, the vertical salinity gradient behaves differently, leading to stratification of the water column. Under such conditions, this model cannot be applied as it would underestimate the salinity levels.

The fact that the water column is so well mixed indicates that we can validate our simulations with data obtained at 2 m depth, since, as seen in the profiles, the variation throughout the water column is reduced, allowing us to state that there is vertical homogeneity.

[Figure]

**Fig. R1**. The top panel corresponds to the tidal current velocity at each sampling station during the MG3 campaign (July 2022), with different colors indicating the tidal phases during which each CTD profile was taken. The bottom panel displays the CTD profiles at each sampling point along the Guadalquivir River during the MG3 campaign.

[Figure]

***Fig. R2.*** CTD profiles at each sampling point along the Guadalquivir River during the MG2 campaign (January-February 2022).

[Figure]

***Fig. R3.*** CTD profiles at each sampling point along the Guadalquivir River during the MG1 campaign (September 2021).

2. **Apart from the anthropogenic freshwater withdrawal, the sink term may also include uncertainties related to unaccounted processes such as drainage from marshes and crop lands, evaporation, vertical mixing etc. A strong justification on the attribution of sink term to anthropogenic effects has to be provided.**

Thank you for your comment. We have included a new section (section 3.1) in order to provide a detailed explanation of why this sink parameter need to be included to simulate real behavior of salinity along the GRE and also a justification of what these parameter represent.

The GRE (Guadalquivir River Estuary) is under intense anthropogenic pressure, as evidenced by the reduction of marshland in recent decades, the expansion of agricultural fields - especially those dedicated to rice cultivation - and the development of urban areas adjacent to the river, among other factors. All these pressures are likely to affect the natural behavior of the GRE and may be one of the causes of the high salinity levels observed throughout the estuary.

One way of quantifying these impacts is in terms of water volume, as certain activities, such as agriculture or industry, withdraw water from the main channel and thereby affect it. In addition, it is important to consider contributions to the channel from smaller tributaries, agricultural fields and other sources.

In our study, the parameter $\delta$ plays a crucial role in quantifying the balance between water withdrawals and the smaller contributions that occur along the river. This allows us to attempt to quantify the impact of all potential activities in the GRE that can be assessed in terms of water volume.

The parameter $\delta$ is a key factor in quantifying the effect of extractions and minor contributions to water volume along the river. In other words, it represents all the natural and anthropogenic processes that can affect the volume of water, such as agricultural abstraction, industrial use, small side channels and evaporation. $\delta$ represents an average value between these two actions, and for our study it was positive, indicating that on average extractions exceed the contributions from smaller channels that drain into the main channel.

However, there is an inherent uncertainty in this parameter due to the complexity of accurately quantifying the amount of water extracted from the channel. The Guadalquivir system is heavily influenced by human activities (high levels of agriculture, industry, dense population in nearby areas, port activities, etc.), and it is also documented that numerous illegal extractions take place. This makes it difficult to obtain accurate data on abstraction within the estuary, as both the specific locations and volumes of water taken are unknown.

The main idea of this study is to show that these actions have a significant effect under low flow conditions, because without them the observed salinity levels would not be reached.

Thanks to the comments of the reviewers, we have reviewed the parameterizations and adapted the code to use higher dispersion values while maintaining the numerical stability of the model. This allowed us to perform a sensitivity analysis using a higher dispersion coefficient (150 m²/s) to evaluate whether the system could reproduce these observed salinity conditions with horizontal dispersion alone.

In this article, the horizontal dispersion coefficient was calculated using the equation proposed by Bowden (1983). This calculation was carried out for all the campaigns considered in the analysis to ensure the use of a constant dispersion appropriate to the system. The results indicate that the maximum constant dispersion, based on speed and depth, is 150 m²/s (this has also been added and explained in detail in the new version of the article). In addition, a sensitivity analysis was carried out with different dispersion coefficients to optimize this parameter as much as possible. In our 1D model, which simplifies the equations governing the balance forces, volume

conservation, and advection-dispersion processes, a dispersion coefficient exceeding 200 m²/s leads to numerical instabilities.

If we analyze the behavior of the horizontal salinity gradient along the estuary, taking into account only the horizontal dispersion, we can see that the system would never reproduce the observed salinity concentrations in the different campaigns. Even when the dispersion coefficient is increased to 190 m²/s, the behavior remains the same (Fig R4). It can be observed that, over time, the salinity concentration increases slightly from 10 km to 40 km. However, it is evident that the observed values are not fully reached. Therefore, it can be said that horizontal dispersion alone does not achieve the high salinity values observed along the channel, which opens the hypothesis that some additional effect is likely to cause a greater penetration of saline intrusion into the estuary.

If the same experiment is carried out but the parameter δ (representing all the processes that reduce the volume of water in the estuary) is included (Fig. R4), the results show that the system reaches the observed salinity values over time. This shows that this term must be included in order to reproduce the salinity concentrations observed in the different campaigns.

Figures 4c and 4d present the experiments that include water withdrawals as a constant value in time and space (δ = 0.005 mm). As shown, as the simulation time progresses, the system achieves the salt concentrations range presented in the observations. This contrasts with the previous cases, where only the dispersion term was included in the experiments (Figures 4a and 4b), and the observed range could not be achieved.

On the other hand, Figures 4e and 4f show the experiments employing time-varying δ. In this simulation, a stronger sink is applied during the first three days (δ = 0.01 mm), which is then reduced and held constant for the remainder of the simulation (δ = 0.001 mm). In this case, the obtained values closely match those recorded in the observations.

These experiments highlight the necessity of including this term (δ) in the simulations, as otherwise, the horizontal salinity gradient would never reach the observed values. As illustrated in the figure, a certain duration of sink activity is required for the simulated salinity concentrations to approach the range observed. Therefore, it is essential to define an initial condition that considers this progression, allowing the simulation to adequately capture the evolution of the water withdrawals and their impact on salinity over time. This is particularly justified by Figures 4e and 4f, where using a stronger δ during the first three days followed by a weaker δ reproduces the observed behavior.

The δ value was determined empirically during the calibration process, through a sensitivity analysis in which different δ values were tested in simulations to identify the one that produced concentrations within the observed range while maintaining the temporal and spatial stability of the model.

Once the δ value was identified, experiments were carried out to analyze its behavior. These included constant use of the parameter over time, and experiments including them at specific time intervals, as well as spatial distribution experiments where δ was applied to specific points (e.g. high areas of the river) and regions. However, due to the limited understanding of the true behavior of these processes, and to avoid introducing assumptions or speculation that could affect the validity of the results, it was considered more appropriate to use a constant value rather than additional assumptions.

As observed, when sinks are included in the model, a certain amount of time is required for the system to reach the salinity concentrations observed. This behavior is represented in the model by an initial condition, designed as a logistic curve, which describes how the effect of the sinks manifests and evolves over a given period of time. This curve makes it possible to simulate the gradual adaptation of the system until it reaches the observed concentrations, providing a useful tool for evaluating and validating the model.

The choice of a logistic curve is justified by its ability to model gradual processes, which makes it suitable to reflect the temporal behavior of the sinks and their impact on the estuarine system. Therefore, the use of this parameter ($\delta$) is an efficient way to quantify the inflows and outflows of water from the main channel, largely due to anthropogenic activities. This approach can be extrapolated to other estuaries with excessively high salinity concentrations in the estuary interior that cannot be explained by dispersion alone. Similarly, this method can be applied to systems under high anthropogenic pressure and similar environmental conditions.

In estuaries with behavior similar to that of the Guadalquivir River, especially in low flow conditions where the tidal action dominates the hydrodynamic behavior, the omission of the anthropogenic effect may lead to an underestimation of salinity concentrations. Therefore, including these effects through the $\delta$ parameter allows for more realistic simulations and helps to understand the impact of these activities. This understanding is essential for effective estuary management, both from a socio-economic and environmental perspective.

The results showed that $\delta$ is positive, indicating that, on average, water withdrawals exceed contributions to the main channel. This finding is consistent with documented evidence of significant anthropogenic impacts on the Guadalquivir system, including numerous illegal withdrawals that hinder the collection of accurate data (e.g. https://www.diariodesevilla.es/andalucia/Confederacion-Guadalquivir-ilegales-abastecian-hectareas_0_1844517515.html).

Our results provide an estimate of how these water withdrawals are occurring, but it is not possible to accurately quantify the exact behavior of water withdrawals due to anthropogenic activities or the exact volume diverted through secondary channels. What it does provide is a general view of the volume of water that needs to be abstracted during these campaigns, including all processes as a whole (including natural processes such as evaporation).

We believe that the use of the parameter $\delta$ is an efficient approach to quantifying the human impact on the system, particularly where there is a high degree of uncertainty in the actual abstraction activities. Furthermore, this methodology can be applied to other estuaries with significant anthropogenic pressures, providing a useful tool for dealing with similar situations of uncertainty.

The hydrodynamic model implemented, with its validation and detailed description available in Sirviente et al., 2023, inherently accounts for other anthropogenic effects, such as tidal amplification at the head of the estuary. Similarly, activities such as dredging and geometric modifications of the channel are considered by incorporating the actual bathymetry of the system

[Figure]

**Fig. R4.** Comparison of simulated (temporal behavior of the simulation at the corresponding observation points) and observed salinity over the MG3 vessel trips: (a) and (b) show simulations including only the horizontal dispersion for the MG3 vessel trip upstream (a) and downstream (b), with the observations in black. (c) and (d) show simulations incorporating δ term (δ) for the entire simulation period as constant value, and (e) and (f) are the simulation including a time-varying δ, compared to the observations (in black) for the MG3 vessel trip upstream (c and e) and downstream (d and f).

3. **Fig. 1b shows that the channel is deep in the 15-25 km distance range, where the salt intrusions appear to be more pronounced (Figs. 5,6). It could be that the mixing induced by strong tidal currents at these depths result in increase in salinity, which is not related to freshwater withdrawal.**

Indeed, in Figure 1b, between kilometers 15 and 25, some sections show greater depths. This graph reflects the actual bathymetry obtained from the nautical chart. Smoothing has been applied in the model to avoid discontinuities and to maintain homogeneity of the data.

In terms of vertical mixing induced by strong tidal currents, there is no significant difference between surface and bottom salinity. As mentioned in the answer to question 1, CTD profiles obtained during different campaigns show homogeneous vertical behavior, indicating that the vertical salinity gradient is practically constant, which implies that discrepancies between surface and bottom salinity are minimal.

In the revised version of the article, in which the model has been adapted as mentioned above, Figures 5 and 6 have been updated. This version presents a more realistic behavior and, by using a more accurate dispersion coefficient, the δ values are lower. Consequently, the increase observed in Figures 6b and 6d is smaller because the increase and decrease of the δ values used in the different experiments are smaller. However, the underlying concept and reasoning presented in the manuscript remain unchanged, although they have been adjusted to reflect these new results. In both figures, the time series of velocity and salinity are also included, overlaid for the Bonanza section (km 4), allowing the moments of maximum and minimum salinity concentration to be shown, as represented in Figures 5c-f and 6c-f.

New Figure 6 (Fig 5 in last manuscript version) and Figure 7 (Fig 6 in last manuscript version) of the manuscript are presented below:

[Figure]

**Figure. 6.** Superposition of current velocity (ms⁻¹) time series and Salinity (psu) time serie at Bonanza station (4 km). black dots mean maximum and minimum salinity moments selected for MG2 (a) and MG3 (b) oceanographic campaigns. Series of salinity (psu) along the Guadalquivir estuary (km) between real

flow and various reductions in freshwater flow for MG2 (c) and MG3 (d). In c and d, the red lines represent experiment (i), the blue lines correspond to experiment (ii) and the cyan lines are experiment (iii). (b) and (d) are the series of salinities using the real freshwater flow and greater freshwater flows for MG2 and MG3, respectively (Experiment (iv) is represented by the blue line, experiment (v) is green line and experiment vi is presented by pink lines). The solid lines represent the time of maximum salinity at Bonanza, and the dashed lines represent the time of minimum salinity at Bonanza. Color dots represent the km of maximum differences between each experiment with experiment (i).

[Figure]

**Figure 7**: Superposition of current velocity (ms$^{-1}$) time series and Salinity (psu) time serie at Bonanza station (4 km). black dots mean maximum and minimum salinity moments selected for MG2 (a) and MG3 (b) oceanographic campaigns. Series of salinity (psu) along the Guadalquivir estuary (km) under a reduction of water withdrawals values are presented in (c) and (d) where original value are represented by blue lines (Experiment i), experiment (ii) corresponding to a smaller reduction is presented in black and a higher reduction of water withdrawal value is presented in red (Experiment (iii). (e) and (f) correspond to experiments increasing water withdrawal values. Original value is presented in blue (Experiment i), and different progressive increases are presented by black lines (Experiment iv), red lines (Experiment v) and green lines (experiment vi). The solid lines represent the time of maximum salinity at Bonanza, and the dashed lines represent the time of minimum salinity at Bonanza. Color dots represent the km of maximum differences between each experiment with experiment (i).

In Figures 5 and 6, we consider that the areas where the changes are most significant are from km 10 to km 30, which could be due to several reasons. First, changes in width are likely to be more important than changes in depth. As shown in Figure 1c, there is a slight narrowing of the system from km 10 to about km 30, where the system widens slightly again. This narrowing could lead to an increase in current velocity, resulting in faster salt transport. However, these effects are not

the primary cause of the observed increase in saltwater intrusion. All simulations use the same bathymetry. One of the advantages of our model over previous models in the literature is that we do not use a channel with constant geometry. Instead, we use real bathymetry that includes changes in both depth and width. This means that our simulations inherently account for all physical processes, such as the Venturi effect due to the narrowing of the channel in certain areas.

As shown in the results presented in the article and in our response to comment 2, it is necessary to account for processes that lead to water volume reduction along the channel.

4. **As noted by the other reviewers, there is confusion regarding the different terminology used for terms such as 'salt wedge' and 'salinity front'. Be consistent with the terminology and define a salt front/wedge. I guess it indicates the region where the lateral gradient in salinity is maximum. In Figs. 5,6 – Mark the location of maximum lateral change in salinity on each curve with a dot in respective color. It will be helpful for the readers to see the spatial variation in the salinity front in each model run.**

Thank you for the suggestion. We have added the corresponding dot to each line. This dot represents the kilometer where the discrepancy between the different experiments and the reference experiment (i) is at its maximum.

We have also corrected the terminology to be consistent through the text.

5. **Observation data from the cruises are gathered in different months, ranging between July-February each year. I'm assuming the anthropogenic water withdrawals do not vary much across these months. Please mention that in the data section.**

In fact, the observations belong to different oceanographic campaigns carried out in different months and years. Specifically, we have observations from September 2021, January-February 2022, July 2022, and October 2023. Table 1 provides a summary of all observations, including their corresponding dates.

Regarding the sinks, as noted in response to comment 2, the parameter $\delta$ has very similar values across all simulations. For MG1, $\delta$ is set to 0.0015 mm. For MG2, the sink increases at certain distances: from 0 km to 22 km, $\delta = 0.0005$ mm; from 22 km to 42.250 km, $\delta = 0.0045$ mm; and from 42.250 km to 85 km, $\delta = 0.0005$ mm. For MG3, $\delta$ is 0.00225 mm, and for MG4, $\delta$ is 0.0012 mm.

This information is included in Section 3.3 because it was obtained experimentally through a sensitivity analysis, as mentioned in response to comment 2. We believe that including it in the data section could cause confusion among readers and lead them to interpret these values as direct observations.

Lines 206 (revised manuscript version):

"To account for water volume withdrawals from the estuary, a parameter called "sink" (denoted by $\delta$) was introduced. It represents the thickness (m) of a water slide of horizontal area equal to $b \cdot \Delta x$, the horizontal area contained between each pair of transversal sections. This parameter is subtracted at each integration time step $\Delta t$ from the previously computed $\eta$ value. This is equivalent to withdrawing a water volume $b \cdot \Delta x \cdot \delta$

at each integration time step $\Delta t$. The suitable value of $\delta$ for each pair of transversal sections is determined together with the validation of the advection and dispersion model, as explained in section 3.2."

Lines 360-372 (revised manuscript version)

"Considering that the intensity, spatial location, and temporal variability of these withdrawals are unknown, the numerical models had to undergo an ad hoc experimental validation for each campaign (MG1, MG2, MG3, and MG4). As mentioned before, for each numerical integration, an initial salt concentration field was defined using a logistic function that was determined by the behavior of the observations. The procedure begins by establishing a $\delta$ value in the hydrodynamic model, running the model, and later using the resulting $u$ and $\eta$ outputs in the advection and dispersion model to fit the salinity observations. The value of $\delta$ was determined empirically through sensitivity analysis until the best-fitting simulation was obtained.

The experimental validation determined that the best fitting of the simulated salinity values to the observations are those presented in the following lines. In MG1, a constant sink $\delta = 0.0015$ mm was implemented in all sections and time steps. For MG2, $\delta = 0.0005$ mm was applied uniformly from 0 km to 22 km and from 42 km to 85 km. From 22 km to 42 km it was increased to $\delta = 0.0045$ mm. In MG3 and MG4, a uniform sink of $\delta = 0.00225$ mm and $\delta = 0.0012$ mm was employed throughout all the sections, respectively. The slightly higher sinks between 22 km to 42 km for MG2 can be justified by the location of crop fields (Fig. 1a) and secondary channels."

Minor comments:

1. **Authors mention mooring observations are used. Are MG1, MG2 and MG3 mooring locations or sampling points for ship? Are the mooring observation integrated with the ship-based data? It may be good to mark the moorings in Fig. 1 and mention the location in the caption. The validation of model results using mooring observations is not shown. It may also be good to add a scatter plot between near-surface salinity from moorings and 2 m salinity from ship-based thermosalinograph data to see how they compare.**

Thank you for the comment. You are absolutely right.

The data referred to as mooring observations are the fixed stations of the thermosalinograph, whose data is recorded at a depth of 2 m depth and corresponds to the points indicated in figure 1a of the manuscript. We have changed "mooring" to "fixed station" to avoid confusion. Similarly, following your suggestion, we have added the figures we mentioned in comment 1 to the supplementary materials.

This figure demonstrates how salinity observations recorded at different depths (CTD salinity profiles) at the same locations as the fixed stations (see Fig. 1a of the manuscript) show homogeneity throughout the entire water column, with very small vertical gradients (which is in agreement with the literature). This allows us to demonstrate several aspects: the strong vertical mixing of the water column, which results in very small vertical salinity gradients, thereby helping us understand the insignificant differences between surface concentration and bottom concentration. Therefore, validation of the model can be performed using the thermosalinograph observations at 2 meters. These figures will be added to the manuscript as supplementary material SM1, SM2 and SM3 (only the points coinciding with the thermosalinograph fixed stations and presented in figure 1a of the manuscript). We will adjust the nomenclature of the other

supplementary material figures in the text to ensure that they are in the correct order. We have added these lines to the text:

Line 180 (in the revised version)

"CTD profiles obtained at the same sampling stations as the thermosalinograph data for campaigns MG1, MG2 and MG3 shown in Figure 1a are also used. These profiles are used to analyze the vertical behavior of the water column."

Lines 275-278 (revised manuscript version)

"The salinity profiles obtained from the CTDs show a strong vertical mixing of the water column throughout the whole period and at all points (see Figures SM1, SM2 and SM3). Very reduced vertical salinity gradients can be observed, which allows us to conclude that the vertical behavior of the water column in the GRE is homogeneous under these conditions, allowing the use of a 1D model to simulate the salinity concentration along the river."

2. **Fig.1c , y-axis label needs to be corrected to "width"**

Done, thank you for catching that mistake

3. **Line 37: Not sure what the word "positive" means in this context.**

We use positive because, GRE is a positive estuary [*Elliot and McLusky*, 2002], in which the freshwater discharges from the basin are sufficient to compensate evaporation losses.

We have modified the text (Lines 37-38) as follows:

"The Guadalquivir River Estuary (GRE) (Fig. 1) is a positive estuary, in which the freshwater discharges from the basin are sufficient to compensate evaporation losses (Diez-minguito et al., 2013). It is generally considered a well-mixed estuary, though this characteristic can change during periods of high discharge, when mixing conditions deviate from the typical pattern (Álvarez et al., 2001)."

4. **Lines 48 and 50: m3/s should be $m^3/s$. Superscript missing in the units in several other places. Please correct.**

Thank you, we have corrected all of them.

5. **Fig. 4 – It is not clear if this model simulation includes sink term or not. Also, please mention in the caption what the contours represent. How does the salt intrusion differ during the spring and neap Adal cycles before and aher including the sink term? It may be worth checking that.**

Thanks for the suggestion.

Figure 4 shows the 15-day simulation corresponding to the MG3 campaign, which is the campaign with the highest number of observations along the river. This allows a solid and reliable validation of the simulation. The simulation includes the parameter $\delta$, as it is essential to reproduce the observed salinity concentrations. Without this parameter, the simulations do not reproduce the observed salinity concentrations. As seen in Figure 3, without this parameter, the horizontal salinity gradient is limited to the first 25-30 km from the mouth (using 5 psu isohaline as the limit), which is inconsistent with the observed salinity concentrations along the estuary, where 5 psu isohaline is close to 60km from the mouth.

We have modified lines 456-459 to clarify this point:

"Once the reliability of the model had been confirmed by the results of the experimental validation presented in the previous sections, it was used to simulate the dynamic of the saline intrusion during a spring-neap tidal cycle. To do this, we conducted a simulation extended over 15 days (15/07/2022-30/07/2022) using the same model configuration presented in section 3.1 for the MG3 campaign."

In Figure 4, the contours represent the different isohalines, showing the variations in salinity along the river under different tidal conditions. We have included this information in the caption.

To improve the precision of the analysis and ensure greater clarity in the results, both the figure and the text of section 3.3 have been revised. This section is now presented as section 3.4, utilizing the updated model configuration. Moments of spring and neap tides have been selected instead of the intermediate tides previously used, enabling a more accurate and comprehensible analysis.

The section has been rewritten in alignment with all the reviewers' comments. Figure 4 now is Figure 5.

"3.4. Tidal cycle dynamics.

Once the reliability of the model had been confirmed by the results of the experimental validation presented in the previous sections, it was used to simulate the dynamic of the saline intrusion during a spring-neap tidal cycle. To do this, we conducted a simulation extended over 15 days (15/07/2022-30/07/2022) using the same model configuration presented in section 3.1 for the MG3 campaign. This period was selected because it comprised records of observations distributed throughout the spring-neap tidal cycle, allowing for the validation of the simulations. A spin-up of 3 days is necessary to stabilize the initial conditions and achieve realistic outputs.

[Figure]

**Figure 5:** (a) Superposition of tidal height (m) and salinity (psu) simulated time series at Bonanza section throughout 15 days of July 2022. Dashed lines indicate the selected 24 h periods referred to in Figs. 4b and 4c; Hovmöller diagrams of simulated salinity variation over these two daily cycles (24 h) during neap tides (A) and Spring tides (B). Isohalines are presented as white lines.

We focused on two 24-hour periods to describe the dynamics of the horizontal salinity gradient during different phases of the semi-diurnal cycle (Fig. 5a). To account for water volume In Figure 5a, it can be observed that the maximum salinity levels occur near the high-water slack, while the minimum salinity levels are recorded around the low water slack. Figure 5b shows the progression of the saline intrusion during neap tides (A). Using the 5 psu isohaline as the boundary for the horizontal salinity gradient, it can be seen that the maximum salinity extends up to 63 km from the mouth, while the minimum values of this isohaline do not exceed 56 km. In contrast, during spring tides (B) (Figure 5c), as expected, higher salinity values are observed throughout all sections of the estuary compared to neap tides. The 5 psu isohaline extends up to 72 km from the mouth, while the minimum values do not exceed 65 km. This shows a difference of approximately 5-8 km between the moments of maximum and minimum intrusion, being this displacement higher for spring tides than neap tides.

In the same way, when comparing the behavior during spring tides to neap tides, we can observe a difference of 8 km between the minimum values and up to 10 km between the maximum values. Therefore, there is an oscillation of approximately 10 km between spring and neap tides. During spring tides, the horizontal salinity gradient reaches higher concentrations further upstream compared to neap tides, where both the maximum and minimum salinity values are lower. This finding is consistent with the results suggested by Díez-Minguito et al. (2013), who documented a net displacement of approximately 10 km between spring and neap tides.

These results suggest that the constant anthropogenic pressure on the estuary has caused a change in saline intrusion, resulting in higher salinity levels upstream of the river compared to the records of previous studies, such as that of Fernández-Delgado et al. (2007). In this study, it was found that over a six-year period (1997–2003), the 5 psu isohaline boundary was located near 25 km at low tide and at 35 km at high tide. The 18 psu isohaline limit was also found to be 5 km and 15 km upstream of the river mouth at low and high tides, respectively."

6. **Fig. 5 – Is this the model surface salinity plotted? Please mention the depth of salinity in the caption. Also, change the legend label in panels (b) and (d) to F +50% Q=18 m³/s**

Thank you for your comment. The salinity represented in the model is the depth average. Because this is a well-mixed estuary, salinity concentrations at different depths do not show significant differences (please see CTD profiles in comment 1). Therefore, using a single vertical point is representative of the entire water column. In this case, since it is a one-dimensional model, the only point considered is the depth average.

We have modified this Figure (now is Figure 6) legend, see new figire in comment 3

7. **Fig. 6 – Use the same y axis limits for panels (a) and (b).**

Thank you, done.

8. **Line 249-250: The November 2023 results are not shown in Fig. 2**

Our apologies for the mistake, it was not November. It has been corrected

Line 388 (revised manuscript version)

The October 2023 observations show

9. **Line 300: may have "an impact" on the salinity wedge penetration**

Done, now it reads as follows, thank you very much:

"On the other hand, the existence of these sinks reveals  that the usage of water, such as those demanded by the adjacent crop fields or other domestic needs, may have an impact on the horizontal salinity gradient."

10. **Line 396: What is 2.5 psu difference? Is it the difference between the slopes of the two lines? Also, in what distance regime?**

It is the maximum difference observed when using a flow rate of Q=40 m³/s compared to the original flow rates (MG2, Q=12 m³/s; MG3, Q=8 m³/s), considering all points in space.

But the text corresponding to this figure has been changed accordingly o the new results.

11. **Line 446: through idealized model setup**

Thank you for your suggestion. However, we would like to clarify that our reference is to the conceptual framework of the experiments rather than the model setup itself. To enhance clarity, we have rewritten the sentence as follows:

"The experiments conducted, based on idealized conditions, provide insight into the magnitude of anthropogenic pressures on the salinization of the GRE"

---

## Author Comment (AC6)

**#Reviewer 4**

We thank the reviewers for their valuable comments, which have been essential in improving our study. In response to feedback from several reviewers, we have optimized the parameterization of the advection-dispersion module, allowing us to use a more realistic dispersion coefficient. In the new version of the article, as detailed in the responses to the questions, we have provided a more precise explanation of the dispersion coefficient used. Additionally, a new section (Section 3.1) has been added, where we justify in detail the need to include sinks, as well as provide a thorough description of the sink parameter. All the results presented in the article have been updated to reflect the new simulations performed with the improved model version. Although these new simulations do not alter the results or discussion presented, they contribute to greater clarity and precision in the presentation of the findings.

**This is a valuable manuscript that helps to gain insight on salt transport processes in well-mixed estuaries, such as the Guadalquivir estuary. The effects on salt distribution due to natural and anthropogenic freshwater outtakes from the estuary is addressed. The topic is relevant as higher salt intrusion in estuaries is expected in the actual global warming and freshwater supply reduction context.**

**My overall impression is positive, although the manuscript requieres a major revision before I could recommend publication.**

Thank you for your kind words and the suggestions you provided. Please find each of them answered below.

**My major concerns are regarding the usage of terms, discrepancies between model output and observations, and citations. Please see specific comments below.**

1. **I think that not all the discrepancies should be attributed to withdrawals. Evaporation rates could be particularly important during the dry season.**

Thank you for your valuable comment.

We agree that not all discrepancies in the observed salinity should be attributed solely to withdrawals. Evaporation rates, especially during the dry season, are indeed an important factor to consider. Natural processes such as evaporation can contribute to a reduction in water volume, potentially increasing salinity levels.

Considering only natural effects, such as evaporation or the small natural channels present in the estuary, would not generate a sufficient volume reduction to account for the high salinity range observed along the river during the different campaigns. It is therefore necessary to include anthropogenic processes such as water withdrawal for agricultural, industrial and urban activities, illegal wells, the creation of secondary channels and the reduction of marshes, among others. All these processes (both natural and anthropogenic) together lead to a reduction in water volume, which could be responsible for the high salinity concentrations observed in the inner part of the estuary. The parameter $\delta$ is a key factor in quantifying the effect of extractions and minor contributions to water volume along the river. In other words, it represents all the natural and anthropogenic processes that can affect the volume of water, such as agricultural abstraction, industrial use, small side channels and evaporation. $\delta$ represents an average value between these two actions, and for our study it was positive, indicating that on average extractions exceed the contributions from smaller channels that drain into the main channel.

Similarly, we have revised Sections 4.1 and 4.2 (now Sections 3.5.1 and 3.5.2, following your suggestion), referring to the moments of maximum salinity at high-water slack and minimum salinity at low-water slack.

2. **If not considered, uncorrected phase lags (which are related to those of the tides) in modeled and observed salinity yield also deviations. As the oceanographic vessel travelled upstream (How much time took the vessel to complete one survey?), the tide propagates and at different locations of the estuary the moment of the tide is different. I think the authors didn't mention whether along-channel modeled salinity is plotted at a given simulation time or at the time the salinity measurement was taken at each location. The authors should discuss that.**

Thank you for your comment.

You are absolutely correct that the tide exhibits a phase lag at different points in the estuary, and this must be accounted for in salinity measurements.

In Fig. R1, we present the time series of current velocity, tidal height, and salinity (all simulated from the model for MG1) at three different points (4 km, 40 km and 80 km form the mouth). As can be seen, the phase lags in both the tide and salinity are accounted for in our simulations. As you rightly pointed out, the tidal wave will exhibit a phase lag as it moves upstream. This aspect was included when we force the model in a section near Bonanza, specifying an appropriate temporal variation of the salinity calculated as a function of the current (u) in this section and derived from the available observations.

[Figure]

**Figure R1.** Time series of 7 days of simulation for tide height (a), current velocity (b), and salinity (c) at three sections of the estuary: 4 km from the mouth (red line), 40 km from the mouth (green line), and 80 km from the mouth (blue line). The model stabilization period is indicated by a grey rectangle.

Regarding the simulations presented with the data obtained from the thermosalinograph as the vessel traveled upstream/downstream through the estuary, we are using the same time instances as the observations (Fig R2a). In Figure R2b, represents the comparison between the observation points and the model simulation points, measuring the spatial differences in meters. In other words, it shows the discrepancy in the location between the observed data points and the model

simulation points, based on the spatial distance. These differences arise because the model's spatial resolution is 25 m, meaning we do not have the exact position of the vessel in the simulation. Instead, we use the nearest point to the vessel's location.

Figure R2c shows the coordinates of each observed point (vessel data) and the corresponding modelled points. Although only the information for MG1 is shown, it is important to note that for each observation presented in the article, the same time instant was used. Our model has a very high temporal resolution (dt= 1 second), which allows us to select the same sampling instants. Similarly, at the spatial level, our model has a spatial resolution of 25m, which enables us to almost exactly select the same observed points. The small discrepancies between both datasets may also be due to the georeferencing of the model, which was carried out through the digitization of the main channel from nautical charts.

[Figure]

**Figure R2.** (a) Temporal difference at each point between simulation and observation data. (b) Spatial difference between observation points locations and simulation points. (c) Map of observation data (vessel trip data) locations in blue, with red dots representing model data locations at each point. This information corresponds to MG1 campaign.

Regarding the time it takes for the vessel to complete its journey, it varies in each case and does not cover a significant temporal range.

In the following table, we summarize the information about the vessel trips for each campaign, titled "vessel trips" in the manuscript. Although the temporal coverage may not be extensive, it allows us to validate the model both in time and space simultaneously. Subsequently, validation is performed using fixed stations (presented in supplementary material as SM4, SM5, SM6). For the MG3 -down- campaign, it has been divided into two parts, as on the 20th, the sampled points were closest to the head. Between the measurements taken on July 20, 2022, and July 21, 2022, sampling was conducted at a fixed station (see SM4b).

| MG | Start Date | End Date | Elapsed Time (hours) |
|---|---|---|---|
| MG1 | 2021-09-20 14:35:00 | 2021-09-20 18:45:00 | 4.17 |
| MG2 | 2022-01-31 16:13:00 | 2022-01-31 23:58:00 | 7.75 |
| MG3 Up | 2022-07-19 11:23:00 | 2022-07-19 16:35:00 | 5.2 |
| MG3 Down (1) | 2022-07-20 17:08:00 | 2022-07-20 19:09:00 | 2.02 |
| MG3 Down (2) | 2022-07-21 19:23:00 | 2022-07-21 21:18:00 | 1.92 |
| MG4 Up | 2023-10-17 09:57:00 | 2023-10-17 16:17:00 | 6.33 |
| MG4 Down | 2023-10-18 07:42:00 | 2023-10-18 11:09:00 | 3.45 |

We have revised the text, as you correctly pointed out that it was not clear that we are comparing the same time instances and approximately the same locations. Thank you for your valuable feedback. We have included the necessary clarifications to enhance understanding in Line 231.

Line 288 (Revised version of the manuscript):

"It should be noted that the simulations presented in this figure represent the simulated salinity concentration generated by the model at each time instance of the observations, corresponding to the nearest possible point to the sampling location of the vessel".

Line 373:

"Fig. 3 shows the behavior of the longitudinal observations collected demonstrating the high accuracy of the simulations (including $\delta$) in replicating the observed data and demonstrating a robust fit across all campaigns. It is important to note that the simulations presented in this figure represent the simulated salinity concentration generated by the model at each time instance of the observations, corresponding to the nearest possible point to the sampling location of the vessel (Fig. SM1)."

3. **Regarding terms usage, some of the them are inappropriate (salt wedge/front) for this estuary during low riverflows. Besides, higher/Lower salt intrusions do not occur at maximum flood/ebb.**

Thank you, we have corrected this by using the precise terms suggested by all the reviewers (Horizontal salinity gradient and salt/saline intrusion). Similarly, it is indeed the case that the salinity maxima and minima do not occur at maximum flood and ebb, as there is a phase lag between them. We did not mean to imply that the maxima occur at maximum flood/ebb, but rather that they occur just after (given the phase lag between salinity and velocity). We have corrected this wording in the text to avoid confusion.

Section 3.3, now renumbered as Section 3.4, has been revised. We have updated the tidal moments and selected neap and spring tides to clarify and simplify the analysis. The text has been modified accordingly. Please see below:

**"3.4. Tidal cycle dynamics.**

Once the reliability of the model had been confirmed by the results of the experimental validation presented in the previous sections, it was used to simulate the dynamic of the saline intrusion during a spring-neap tidal cycle. To do this, we conducted a simulation extended over 15 days (15/07/2022-30/07/2022) using the same model configuration presented in section 3.1 for the MG3 campaign. This period was selected because it comprised records of observations distributed throughout the spring-neap tidal cycle, allowing for the validation of the simulations. A spin-up of 3 days is necessary to stabilize the initial conditions and achieve realistic outputs.

[Figure]

**Figure 1:** (a) Superposition of tidal height (m) and salinity (psu) simulated time series at Bonanza section throughout 15 days of July 2022. Dashed lines indicate the selected 24 h periods referred to in (b) and (c); Hovmöller diagrams of simulated salinity variation over these two daily cycles (24 h) during neap tides(b) and spring tides (c) along the Guadalquivir estuary. Isohalines are represented as white lines.

We focused on two 24-hour periods to describe the dynamics of the horizontal salinity gradient during different phases of the semi-diurnal cycle (Fig. 5a). To account for water volume In Figure 5a, it can be observed that the maximum salinity levels occur near the high-water slack, while the minimum salinity levels are recorded around the low water slack. Figure 5b shows the progression of the saline intrusion during neap tides (A). Using the 5 psu isohaline as the boundary for the horizontal salinity gradient, it can be seen that the maximum salinity extends up to 63 km from the mouth, while the minimum values of this isohaline do not exceed 56 km. In contrast, during spring tides (B) (Figure 5c), as expected, higher salinity values are observed throughout all sections of the estuary compared to neap tides. The 5 psu isohaline extends up to 72 km from the mouth, while the minimum values do not exceed 65 km. This shows a difference of approximately 5-8 km between the moments of maximum and minimum intrusion, being this displacement higher for spring tides than neap tides.

In the same way, when comparing the behavior during spring tides to neap tides, we can observe a difference of 8 km between the minimum values and up to 10 km between the maximum values. Therefore, there is an oscillation of approximately 10 km between spring and neap tides. During spring tides, the horizontal salinity gradient reaches higher concentrations further upstream compared to neap tides, where both the maximum and minimum salinity values are lower. This finding is consistent with the results suggested by Díez-Minguito et al. (2013), who documented a net displacement of approximately 10 km between spring and neap tides.

These results suggest that the constant anthropogenic pressure on the estuary has caused a change in saline intrusion, resulting in higher salinity levels upstream of the river compared to the records of previous studies, such as that of Fernández-Delgado et al. (2007). In this study, it was found that over a six-year period (1997–2003), the 5 psu isohaline boundary was located near 25 km at low tide and at 35 km at high tide. The 18 psu isohaline limit was also found to be 5 km and 15 km upstream of the river mouth at low and high tides, respectively."

Similarly, we have revised Sections 4.1 and 4.2 (now Sections 3.5.1 and 3.5.2, following your suggestion), referring to the moments of maximum salinity at high-water slack and minimum salinity at low-water slack.

**4. Please check references out, too. I think a few of them are not appropriate or out of place.**

Done, thank you for the suggestion.

**Specific comments:**

Abstract

**L13. (there exists advective and diffusive transport)**

Done, thank you

**L15. I agree with one of the other reviewers. There is no salt wedge or salt front in the GRE during low riverflows. Please use salt intrusion or saline intrusion. Salt wedge typically occurs in highly stratified estuaries. Change here and elsewhere. Salt front may form after high freshwater discharges near the mouth.**

Done, thank you. We have modified all terminology including salt intrusion and horizontal salinity gradient.

**L17. Water withdrawal or sink term or outflow?**

We have modified for sink, we are referring to δ parameter which has been called "sink" in the manuscript.

Introduction

**L28. Perhaps Miranda et al. 2017 is much more "physical" than "biological". I suggest to change citation here.**

Done, we have rewritten the sentence and eliminated the citation:

"These coastal areas are typically described as highly productive zones, where nutrients from the land are incorporated into a transitional system between fresh and sea waters"

**L29. + freshwater discharge + tides + wind**

Done, thank you

**L32. I think is better here to cite a general reference on ecological systems better than Donázar-Aramendía et al., (2019), which is somehow more "local".**

Done, we have changed for a more general reference

Boehlert, G. W., & Mundy, B. C. (1988). Roles of behavioral and physical factors in larval and juvenile fish recruitment to estuarine nursery areas. In *American Fisheries Society Symposium* (Vol. 3, No. 5, pp. 1-67).

**L37. Remove Pritchard (not on the GRE) and Losada citations. If you want to include some here, I suggest Álvarez et al. 2001 or Díez-Minguito et al., 2013.**

Done, thank you

**L39. Remove Reyes-Merlo et al. citation. The credit on this should go to Álvarez el al. 2001, I think.**

Done, thank you.

**L41. Remove citation. Citation here (Contreras and Polo, 2012) is on the influence of basin hydrology on the GRE. Please check out that statements throughout the manuscript are correctly cited.**

Done, thank you.

**L50. 'and decreasing saline intrusion.'**

Done, thank you.

**Figure1. Xo is the landward limit of the model. Does the boundary condition of salinity at Xo vary with time?**

Yes, the salinity boundary condition at X0 (free transmission) varies over time at X0 (the first point in the model mesh). This behavior occurs because its value depends on the salinities of neighboring spatial points at the same time instant, making it a dynamic component of the model. We understand that the ability of the boundary condition to adjust to fluctuations in the system is crucial for capturing the complexity of estuarine processes and for providing more accurate simulations that better reflect the real environmental conditions.

**L60. Bermudez et al. 2021 is on microplastic distribution in the GRE. I suggest to remove from here.**

Done, thank you

**L62. 'requires large amounts of freshwater' ?**

Thanks, we have modified the sentence:

"However, it is also a source of economic and environmental conflict due to the coexistence of multiple activities (salt production, agriculture -especially rice, which requires large amounts of freshwater - fishing and navigation)."

**L65. Notice that none of the citations in on degradation of the ecosystem. Moreover, Zarzuelo et al. 2017 is a study developed in Cadiz Bay.**

We have modified it. Thank you

"The high anthropogenic pressure has caused changes in both the hydrodynamics and morphology of the system, favoring the constant degradation of the ecosystem (Mendiguchía et al., 2007; Ruiz et al., 2015: Siles-Ajamil et al. (2019); Sirviente et al., 2023)."

**L79. And downstream, too. Seaward from Bonanza, salinity does not change so much either. Typically is modelled as a tanh(x) function. Nevertheless, it is possible that authors' model domain leaves out this part of the salinity distribution.**

Thank you for your comment. The domain of our model extends up to Bonanza, thus this part of the salinity distribution is not included.

**L86. -1 superscript.**

Done, Thank you

**L87. Climatic forcings**

Thank you, done

**L89. Salt/saline intrusion**

Done, Thank you

**L92. Notice also that hydrodynamic model by Siles-Ajamil et al. is linear. Authors' model is non-linear, which is quite an improvement.**

Thank you; we have noted this in the text

Line 119: Using a linear analytical model, Siles-Ajamil et al. (2019)

**L92. How does compare authors' model with that by Bouke et al. (2022)? (https://doi.org/10.1029/2022JC018669)**

 Thank you, we have included this reference as:

"Biemond et al. (2022) investigate the impact of freshwater pulses (brief periods of high river flow) on estuarine saline intrusion in GRE. They constructed an idealized nonlinear model based on the model of MacCready (2007), but not based on Pritchard equilibrium. The model is width-averaged, which means it doesn't resolve variations across the channel width, but fully captures variations in both the along-channel and vertical directions. The vertical structure is represented using multiple modes (between 5-15 depending on stratification) to capture the depth-dependent aspects of flow and salinity. The model features constant width and depth, as well as constant viscosity and diffusivity coefficients in space and time. The new model demonstrates that the intensity and duration of the pulse are the primary factors controlling the reduction of salt intrusion, with tidal force having a relatively minor influence. Additionally, the time required for saline intrusion to return to its initial position is found to be dependent on the river discharge following the pulse, rather than the distance the intrusion has traveled upstream."

**Methodology**

**L121. Is really the focus of the manuscript the calibration and validation of the model? I think calibration and validation is necessary, but shouldn't be the solely objective of the study. I suggest to rephase the sentence.**

Thank you for the suggestion, we have changed it:

"This manuscript presents an analysis of salinity behavior along the GRE. In this context, in situ observations collected from several short-duration oceanographic campaigns conducted during

the dry seasons from 2021 to 2023 will be employed for the calibration and validation of the advection and dispersion module."

**L123. Ok**

**L125. I suggest to mention or present these issues already in the introduction. It will help to motivate the object and give the proper context to the readers.**

Thank you, following this comment and one suggestion of another reviewer, we have included some information about water withdrawals effect in saline intrusion in other estuaries. And mention this issue too.

After line 65:

"The detrimental effects of anthropogenic activities have been demonstrated in other estuaries around the world. Alcérreca-Huerta et al. (2019) show that the construction of a dam system in the estuary of the Grijalva River (Mexico) in 1959 altered the hydrological regime, reducing the seasonality of water discharge and decreasing the amount of available freshwater. This, together with changes in land use (more agricultural land, less mangrove cover and less vegetation), leads to variations in salinity concentration, with saline intrusion observed up to 46 km upstream, with salinity levels reaching 32.8 PSU. Studies such as Huang et al. (2024), based on numerical simulations using a 3D model, show that anthropogenic activities, in particular the regulation of freshwater flows by infrastructure projects, are drastically changing the dynamics of saline intrusion in the Changjiang River estuary (China). This study shows how an increase in freshwater flows (due to releases from the Three Gorges Reservoir) counteracts the advance of saline intrusion. However, water withdrawals in the city of Yangzhou as part of the implementation of the East Route of the South-to-North Water Transfer Project will inevitably lead to a reduction in inflow during the dry season, resulting in an increase in salinity intrusion in this system by approximately 6-7 km. This relationship between salinity and freshwater flow was previously observed by Webber et al. (2015), who assessed the effects of the Three Gorges Dam, the South-to-North Water Transfer Project, and local water withdrawals on the probability of intrusion in the Changjiang River estuary. They conclude that these projects will increase the probability of saline intrusion and suggest that water management should be adapted to mitigate the risk."

And after line 108:

This manuscript provides an analysis of the current state of saline intrusion in the GRE and evaluates the impact of anthropogenic pressures on the behavior of horizontal salinity gradient. This study represents the first attempt to analyze the impact of anthropogenic water volume withdrawals on estuary hydrodynamics and the associated increase of the horizontal salinity gradient.

**L130. '…along the estuary.'**

**Table 1. Salinity measurements along the estuary are not simultaneous. They are out of phase from one point to another point within the estuary (salinity changes locally during floods, ebbs, etc.). Are the different phases somehow corrected to present 'simultaneous' salinity values along the estuary? (Also in L200)**

Thank you.

Our hydrodynamic model accounts for the tidal phase lag between different points along the estuary. Similarly, the advection-dispersion model incorporates this phase lag, allowing us to simulate salinity more accurately. This means that we can calculate salt concentration at any point in the estuary, taking into account the specific tidal state at that location. In this way, local salinity variations are adequately represented based on the tidal phase at each point.

**Line 185 (Revised manuscript version):**

The observations corresponding to each vessel trip were taken at different points and time instances as the vessel ascended or descended along the GRE. Once the simulations corresponding to each campaign for validation are obtained, the same time instances as each observed data point are used. Additionally, model points are used as close as possible to the observed points, with small differences between them (Fig SM1). This ensures that the same points and time instances as the observed data are compared (tidal phase lag are contemplated in our simulations).

Where Fig SM1 corresponds to Fig R2 presented in this document.

**After Equation (2) '.' not ','.**

Thank you, done.

**L148. Forces per unit mass.**

Thank you, done.

**L149. Units. Separate m from s**

Thank you, done.

**L150. Bottom friction or bottom drag coefficient?**

*K* means bottom friction coefficient

**L161. Why not include El Gergal and Brazo de la Torre too?**

More tributaries were analyzed than those presented in the article, but only those considered to have a significant contribution were included. In the case of these two tributaries, their influence was minimal during the period analyzed. It was decided not to include contributions with very low contribution as they did not result in a significant change in the results obtained.

**L170. Salt transport. Check out here and elsewhere: advection and dispersion transport (there exists advective and diffusive transport)**

Thank you, done.

**Equation (3). No sink terms in this equation? Could evaporation rates be comparable or may affect \delta estimates?**

$\delta$ term is included in the hydrodynamic model. We understand that *delta* refers not only to anthropogenic activities but also to all processes that lead to a reduction in volume, including evaporation. However, we believe that processes such as evaporation are not the sole contributors to the high salinity concentrations observed. Anthropogenic effects play a crucial role and are fundamental in explaining these high salinity levels.

**L197. Please, indicate determination coefficient of the fit and/or other performance scores of the calibration. (Or refer to Table 2)**

A detailed description of the dispersion term (D) has been included in the new Section 3.1, along with a demonstration of the need to account for volume reductions in order to replicate the observed concentration. Also we have referred to Table 2.

Thanks to the reviewers' comments, we reviewed and adjusted the transport model to use a higher, more realistic constant diffusion coefficient without introducing instabilities into the model. In the revised version, based on Bowden (1983), we defined the most appropriate horizontal dispersion coefficient, taking into account the mean depth and tidal amplitude. This calculation was performed for all campaigns included in the analysis to ensure the use of a constant dispersion value appropriate for the system. The results indicate that the maximum constant dispersion based on velocity and depth is 150 m²/s (this has also been added and explained in detail in the revised version of the article). In addition, a sensitivity analysis was performed with different dispersion coefficients to optimize this parameter as much as possible. It was found that a dispersion value higher than 200 m²/s leads to numerical instabilities in the system.

Based on Bowden (1983), the effective horizontal dispersion coefficient can be calculated as $K_X = U^2 H^2/30*K_z$, where $U$ is the maximum tidal velocity, $H$ is the mean channel depth, and $Kz$ is the vertical eddy dispersion coefficient, assumed to be constant. In our case, we used $Kz=0.01$, as proposed by Bowden (1983).

| Campaign | $U$ (ms$^{-1}$) | $K_X$ m$^2$s$^{-1}$ |
|---|---|---|
| MG1 | 0,85 | 143 |
| MG2 | 0,88 | 154 |
| MG3 | 0,88 | 153 |
| MG4 | 0,80 | 127 |

This selection of a constant dispersion coefficient is based on the assumption that lateral dispersion is homogeneous and that strong tidal currents will induce vertical mixing, thereby rendering advection the dominant process in the behavior of the saline intrusion. The Peclet number ($Pe$), defined as $uL/D$, measures the relative contribution between the nonlinear advection and horizontal dispersion, where $u$ is an averaged (in time and along the whole estuary) absolute value of the along-channel gradient of velocity, $L$ is the estuary length, and $D$ corresponds to the horizontal dispersion coefficient (Deng et al., 2024). Taking a value u= 0.5 ms$^{-1}$, extracted from realistic simulations performed with the hydrodynamic module and the values L= 107 km and D=150 m$^2$ s$^{-1}$ yields a value Pe=356, clearly indicating a dominance of the advective transport rate over the diffusive one.

**Results and Discussion**

**L110-115. I think this must be better justified here or elsewhere. Authors must be clear how they drew the conclusion that there must be water withdrawals (although certainly possible). 'Flow data by official sources?' Do you mean freshwater discharges? What were the tidal and riverflow conditions?**

Thank you for this suggestion.

Yes, the official sources we are referring to (for instance, Guadalquivir Hydrologic Confederation) have provided freshwater discharges. On the other side, authors have not been able to find any official source that provided water withdrawals. In the current version of the manuscript we have added a new section (section 3.1) where the justification for including water withdrawals it is better addressed. In that section is analysed the effect of horizontal dispersion vs water

withdrawals in the penetration of the saline front. Here reasonably high dispersion coefficient are allowed, following the formulation due to Bowden (1983). Regarding the phrase "flow data from official sources," we are actually referring to the simulation that only considered horizontal dispersion and freshwater discharge inputs. In other words, we were referring to the simulation where water volume reductions were not included. We have revised this to make the explanation much clearer and more detailed.

"During the model validation phase, it was found that the extension of the saline intrusion into the interior of the estuary coming from the data measurements was much greater than those simulated including only the freshwater flow data and horizontal dispersion. Hence, we became aware that significant undocumented water withdrawals were occurring during the different campaigns (as it has been described in section 3.1)."

New section has been included as:

**"3.1. Effect of Horizontal Dispersion and Water Withdrawal on the Horizontal Salinity Gradient in the Estuary**

First, the model is used to analyze the effect of horizontal dispersion on the development of the horizontal salinity gradient. To this end, experiments were conducted considering only the effect of horizontal dispersion, representing the natural behavior of the system without any additional intervention. Subsequently, experiments incorporating the presence of sinks were carried out, simulating a reduction in the water volume of the channel.

Figure 2 shows the observed horizontal salinity gradient during the MG3 campaign (upstream and downstream trips) along with simulations corresponding to a 13-day period. This figure compares the simulated horizontal gradients under different conditions: Figures 2a and 2b include only the effect of horizontal dispersion; Figures 2c and 2d include both horizontal dispersion and a uniform reduction in water volume ($\delta$), constant in time and space; and finally, Figures 2e and 2f consider horizontal dispersion along with sinks ($\delta$) that vary over time. It is important to emphasize that all the times shown in Figure 2 correspond to moments when the tidal behavior is identical to that observed at each point during the reference period. It should be noted that the simulations presented in this figure represent the simulated salinity concentration generated by the model at each time instance of the observations, corresponding to the nearest possible point to the sampling location of the vessel.

When analyzing the behavior of the horizontal salinity gradient along the estuary, considering only horizontal dispersion, it becomes clear that the system would not reproduce the observed salinity concentrations from the different campaigns. Even when the dispersion coefficient is increased to 190 m² s$^{-1}$ (Figures 2a and 2b) – the highest dispersion coefficient that can be used with this model without causing numerical instability – the observed concentrations are not matched.

Considering only the effect of horizontal dispersion, it can be observed that the system tends to slightly increase the salinity concentration over time in almost all sections of the river up to 45 km from the estuary. Beyond this point, the behavior becomes almost linear, with very low salinities close to 0 psu. These results show that if only the effect of horizontal dispersion is considered, the system is unable to reproduce the observed salinity range.

This highlights the need to include in the simulations those processes that could cause a significant increase in the horizontal salinity gradient. These processes may include those capable of reducing

water volume. Considering only natural effects, such as evaporation or the small natural channels present in the estuary, would not generate a sufficient volume reduction to account for the high salinity range observed along the river during the different campaigns. It is therefore necessary to include anthropogenic processes such as water withdrawal for agricultural, industrial and urban activities, illegal wells, the creation of secondary channels and the reduction of marshes, among others. All these processes (both natural and anthropogenic) together lead to a reduction in water volume, which could be responsible for the high salinity concentrations observed in the inner part of the estuary.

The parameter $\delta$ is a key factor in quantifying the effect of extractions and minor contributions to water volume along the river. In other words, it represents all the natural and anthropogenic processes that can affect the volume of water, such as agricultural abstraction, industrial use, small side channels and evaporation. $\delta$ represents an average value between these two actions, and for our study it was positive, indicating that on average extractions exceed the contributions from smaller channels that drain into the main channel.

However, there is an inherent uncertainty in this parameter due to the complexity of accurately quantifying the amount of water extracted from the channel. The Guadalquivir system is heavily influenced by human activities (high levels of agriculture, industry, dense population in nearby areas, port activities, etc.), and it is also documented that numerous illegal extractions take place. This makes it difficult to obtain accurate data on abstraction within the estuary, as both the specific locations and volumes of water taken are unknown.

When a constant water volume reduction term is included in the channel ($\delta$ = 0.005 mm), both in time and space (Figure 2c and d), meaning that the same volume of water is removed at all points in the estuary during the 13 days of simulation (dt = 1s and dx = 25m), it is observed that the system tends to reproduce higher salinity concentrations along the estuary over time, reaching the observed salinity ranges. This shows that it is essential to include this parameter in the numerical model in order to accurately simulate the high salinity concentrations observed along the GRE. The water volume removed in this experiment throughout the 13 days of simulation was 47.36 106 m3 which is not an excessive amount if it is compared with the water volume needed, for instance, to sow a rice field of 32000 hectares which requires 384.0 106 m3 of water consume.

Figures 2c and 2d show that a certain period of time is required for the sinks to effectively influence the system and produce salinity values close to those observed. Figures 2e and 2f show experiments where a stronger sink ($\delta$ = 0.01 mm) is applied to all sections during the first 3 days of the simulation, after which it is relaxed, and a $\delta$ = 0.001 mm is imposed for the remaining 10 days. It can be seen that the behavior reproduced at all time steps closely matches the observed horizontal salinity gradient, which allows us to conclude that in order to simulate realistic salinity concentrations with this model, it is also necessary to take into account the temporal effect of the sinks.

[Figure]

**Figure 2:** Comparison of simulated (temporal behavior of the simulation at the corresponding observation points) and observed salinity over the MG3 vessel trips: (a) and (b) show simulations including only the horizontal dispersion for the MG3 vessel trip upstream (a) and downstream (b), with the observations in black. (c) and (d) show simulations incorporating δ term (δ) for the entire simulation period as constant value, and (e) and (f) are the simulation including a time-varying δ, compared to the observations (in black) for the MG3 vessel trip upstream (c and e) and downstream (d and f).

These experiments highlight the necessity of including parameters in the simulations, as otherwise, the horizontal salinity gradient would never reach the observed values. As illustrated in the figure, a certain duration of sink activity is required for the simulated salinity concentrations to approach the range observed. Therefore, it is essential to define an initial condition that considers this progression, allowing the simulation to adequately capture the evolution of the water withdrawals and their impact on salinity over time. This is particularly justified by Figures 4e and 4f, where using a stronger δ during the first three days followed by a weaker δ reproduces the observed behavior.

To properly define this initial condition, it is necessary to consider that the observations recorded during the different oceanographic campaigns were made over different time periods (months and years). This implies that each simulation, corresponding to a specific time period, will have a

slightly different initial condition due to the variations in the characteristics of the system over time. This approach emphasizes the importance of adjusting the initial conditions according to the temporal differences observed in the data, allowing for a more accurate representation of the system's behavior during each period considered.

It is important to emphasize that this numerical model, although simple, has been designed as a very useful tool for studying the hydrodynamic and physicochemical properties of the estuary. As a high-resolution 1D model, it is optimized to simulate relatively short time periods, ensuring high computational efficiency. The model is particularly effective at representing specific moments in time and extrapolating to a given time interval. Although it is designed to simulate shorter time periods, it can be used for longer simulations provided that similar conditions are maintained, such as low discharge regimes where the water column is well mixed and vertical gradients are homogeneous. However, in situations with significant stratification, alternative approaches would need to be considered as 1D simulations would not be suitable to accurately simulate the estuary."

**F233. Please indicate if Figure 2 include the profiles with the sink correction. If not, it would be helpful to see also the uncorrected profiles.**

Thank you done. Now this Figure corresponds to Figure 3 and is presented as follows:

[Figure]

**Figure 3:** Comparison between the observations (blue line) and the simulation including sinks (red line) and simulations not including sinks (black lines) of the salinity (psu) for the hole channel for different campaigns (Table 1).

**Figure 2. Name the panels a), b) etc. to link better to what it is described in the main text.**

Thank you done, please see figure in previously comment.

**L248-249. Please explain this according to the spatio-temporal variability of water extraction for agricultural activities.**

This question cannot be answered with certainty. The lack of detailed knowledge about how the sinks operate prevents us from confidently addressing the spatio-temporal variability of water extractions. What we can affirm is that the discrepancies are observed in areas near crop fields, suggesting that these differences are likely linked to water extractions for agricultural purposes.

To clarify this point further, we have added the following explanation in Line 378:

"The discrepancies found between 30 km and 60 km may be attributed to the crop fields adjacent to the channel (this is the area where the largest concentration of crop fields is located on both sides of the GRE)."

**L249. November 2023. Which panel?**

This was a mistake, it is October.

**L250. Could the authors provide a water volume to provide evidence for this? (Ok. Seen answer in L276)**

Okey, thank you.

**L276. I understand that these water outtakes for crops could be difficult to obtain. However, could be estimated with authors' model results.**

We can provide a rough estimation; however, this comes with significant uncertainties due to our limited understanding of how the sinks operate—whether they are constant or variable, for example. We recognize that water extractions likely vary depending on the stage of the agricultural cycle (e.g., during periods when fields are intentionally flooded, water usage would be higher). However, we cannot confirm this with certainty, nor can we account for illegal extractions or situations where artificial channels may be opened for irrigation purposes.

For this reason, we have opted to treat water extractions as constant in our model to maintain methodological rigor. While we can approximate the total volume of water extracted during each campaign based on our results, extrapolating beyond this would not be reliable. Moreover, distinguishing between losses due to illegal extractions from fields, unauthorized well usage, secondary canal diversions, evaporation losses, and other factors remains beyond the scope of our current analysis.

To quantify these factors more accurately, in-situ data collection will be necessary. We plan to conduct this in the near future, which will allow us to investigate this issue in more detail. However, thanks to the results presented in this article, we can confidently state that quantifying these activities is crucial for a true understanding of the current system.

**L276-L282. Again. Explain whether or not the timing of the salinity measurements along-estuary and model output yields discrepancies. Also in L291 Figure 3), please indicate whether or not the model output along-estuary is plotted at the same time? I presume that the results obtained from measurements are not, since they are measured as the vessel travelled upstream. Discrepancies up to 10psu could be observed, depending on the location and timing.**

The data represented by the model, as previously mentioned, corresponds to the same time instances and virtually the same points as the observations. Regarding the comparison of the observations (vessel trips), the salinity gradient presented does not correspond to the same temporal moment at all points, since, as noted, the observations were taken as the vessel traveled upstream or downstream. On the other hand, the model data represented corresponds to the same temporal instances as the observations and to the same points, allowing us to validate the model both in time and space simultaneously.

Figure R3, shows the time series of current velocity for three different sections of the river (4 km, 30 km, and 60 km). The time points where the vessel trip data (up and down) were recorded during the MG4 campaign are marked in red. The purpose of this figure is to illustrate the behavior of current velocity at the times the data were collected. The three sections represent different locations in the river: one in the lower section, one in the middle section, and one in the upper

section. This shows that the tidal moments vary depending on the specific section where the sample was taken, as phase shifts corresponding to each zone are included.

[Figure]

**Figure R3**. Evolution of velocity at three different distances (4 km, 30 km, and 60 km) over time. The observational data (red lines) corresponds to two vessel trips: on 17/10/2023 (solid line) and on 18/10/2023 (dashed line).

**Figure 3. Label is wrong in panel (e) (it says (d))**

Corrected thank you

**L303. What would be the increase of freshwater discharges from the dam in that case?**

We have not exactly quantified the required freshwater flow rate to maintain the salinity intrusion at a level that does not exceed 30 km. However, we know (due to our results) that it must be higher than 185 m³/s in each campaign.

**L315. Does the model need any spin up time?**

Yes, the model requires a spin-up period of 3 days to stabilize the initial conditions and achieve realistic outputs. During this time, the model adjusts to a more stable state before the results can be considered reliable

**L325. Theoretically, maximum (minimum) salinity values occur at high (low) water slacks, which occur at different times along the estuary.**

Thank you, we have corrected this aspect along the hole manuscript.

As a example, line 325 (line 470 in new version):

"To account for water volume In Figure 5a, it can be observed that the maximum salinity levels occur near the high-water slack, while the minimum salinity levels are recorded around the low water slack."

**Figure 4. Panels b) and c). Around km 17, along-channel salinity inversions are observed (larger salinity values upstream). Do have the authors any clue on this? Is due to tidal trapping in Brazo del Oeste?**

Thank you for your comment. Looking at the plot, we do not see a clear indication of salinity inversions around km 17. The x-axis represents time, while the y-axis represents space, and if we examine the plot, we can see a general decrease in salinity as we move upstream in the estuary. Anyway the 'Brazo del Oeste' is not considered in the model geometry. In some preliminary tests this channel was included but it does not improve the experimental validation of the model. So, it was not considered in the final configuration of the model.

Furthermore, this figure has been modified (please see Figure 5 showed above)in the new version of the manuscript, where we used neap and spring tide moments to simplify the analysis and focus on more representative conditions. As for the specific salinity values around 17 km, we can confirm that the concentration at this location is higher than at 18 km, but lower than at 16 km.

**L327. "the wedge demonstrates minimal intrusion". Rephrase**

Done. We have modified the text of section 3.4, please see comment 3 on this document.

**L338-340. As the authors mention, saline intrusion is important for water quality and residence times are relatively large in the estuary. Nevertheless, notice that author's model is cross-sectionally-averaged. This is ok, since mixing rates within the estuary are quite high. But should be recognized that the model does not resolve the vertical segregation of the flow, which could be important in the lower part of the estuary and enhance flushing times there.**

Thank you.

The reviewer is right regarding the possible uncertainties in the flushing times arising from the cross-section average. Anyway, these comments about the transport to the continental shelf in estuary mouth are not considered in the current version of the manuscript.

**L341. I don't see how this conclusion is drawn. Please explain why there is a positive mass balance at the mouth? Is this due to evaporation rates or outtakes for crops within the estuary? Does this volume compensate for freshwater losses within the estuary?**

Thank you, we have removed this, with the new simulation the balance in the bonanza section is negative while it is positive in section at 11 km. To avoid confusion and the potential for erroneous hypotheses, it has been decided to remove this phrase.

**Discussion of… (discussion in Section 3 and also in Section 4?)**

**I find this part the most interesting. Please consider to include it in Section 3.**

**L347-351. Perhaps the most recent estimates of salt intrusion trends in European estuaries are those by Lee et al. (2024) https://doi.org/10.1038/s43247-024-01225-w**

Thank you, this reference has been included.

Line 488-490 (revised version of the manuscript)

"The natural flow regime of the Guadalquivir estuary has undergone significant changes due to different human activities in the basin (Bramato et al., 2010 Lee et al., 2024). Future projections indicate a reduction in freshwater flow for this estuary by the end of the 21st century (Lee et al., 2024)."

**L374. Yearly-averaged value?**

Yes, thank you.

**L375-385. Maximum and minimum saline intrusion occur at slacks, not at maximum flood and ebb. I don't think this alters authors' point, but please amend here and elsewhere, and modify figures if needed.**

Corrected, thank you. We have modified all text and figures.

**L410-413 (here, before and after) I suggest to name the cases, including the reference one, and use the same notation in the Figures.**

Corrected, thank you.

We named the experiment in section (4.1 now it is section 3.5.1) as : Experiment (i), Experiment (ii), Experiment (iii), Experiment (iv, Experiment (v), Experiment (vi).

And in section 4.2 (section 3.5.2 in new version) as: Experiment (i), Experiment (ii), Experiment (iii), Experiment (iv, Experiment (v), Experiment (vi).

"3.5.1. Changes in freshwater flows

The freshwater discharge observed in MG2 and MG3 are respectively $Q = 12$ m³ s$^{-1}$ and $Q = 8$ m³ s$^{-1}$; these values were used as reference values. Five experiments were conducted under the following different freshwater flows:

(i) Original freshwater flow ($Q_{MG2}$=12 m$^3$ s$^{-1}$; $Q_{MG3}$=8 m$^3$ s$^{-1}$)

(ii) Observed freshwater flow reduced by 50%. ($Q_{MG2}$=6 m$^3$ s$^{-1}$; $Q_{MG3}$=4 m$^3$ s$^{-1}$)

(iii) Observed freshwater flow set to zero.

(iv) Observed freshwater flow increased by 50%. ($Q_{MG2}$=18 m$^3$ s$^{-1}$; $Q_{MG3}$=12 m$^3$ s$^{-1}$)

(v) Observed freshwater flow increased up to $Q$=40 m$^3$ s$^{-1}$, established as the low-flow condition, following Díez-Minguito et al. (2012).

(vi) Yearly average freshwater flow of 185 m$^3$ s$^{-1}$, following Costa et al. (2009) and Morales et al. (2020). "

**"3.5.2 Changes in the water volume sinks.**

To evaluate the effect of decreasing or increasing water withdrawals from GRE, four experiments were conducted, taking the sinks established in the validation of the numerical model for MG2 and MG3 campaigns as a reference:

(i)  Reference δ (MG2$_{0-22km}$=0.0005 mm, MG2$_{22-42km}$=0.0045 mm, MG2$_{42-85km}$=0.0005 mm, MG3$_{0-85km}$=0.00225 mm)

(ii)  δ Decrease by 15%. (MG2$_{0-22km}$=0.000425 mm, MG2$_{22-42km}$=0.0038 mm, MG2$_{42-85km}$=0.000425 mm, MG3$_{0-85km}$=0.0019 mm)

(iii)  δ Decrease by 50%. (MG2$_{0-22km}$=0.00025 mm, MG2$_{22-42km}$=0.00225 mm, MG2$_{42-85km}$=0.0005 mm, MG3$_{0-85km}$=0.0011 mm)

(iv)  δ Increase by 15%. (MG2$_{0-22km}$=0.000525 mm, MG2$_{22-42km}$=0.0052 mm, MG2$_{42-85km}$=0.000575 mm, MG3$_{0-85km}$=0.0026 mm)

(v)  δ Increase by 50%. (MG2$_{0-22km}$=0.00075 mm, MG2$_{22-42km}$=0.0067 mm, MG2$_{42-85km}$=0.00075 mm, MG3$_{0-85km}$=0.0034 mm)

(vi)  δ Increase by 100%. (MG2$_{0-22km}$=0.001 mm, MG2$_{22-42km}$=0.009 mm, MG2$_{42-85km}$=0.001 mm, MG3$_{0-85km}$=0.0045 mm)"

**L421. (here, before and after). What E and F stand for? I suggest to change the names or not saying that they correspond with the minimum or maximum intrusion.**

Thank you, this has been corrected. Following your suggestion, we adjust the text indicating that maximum (minimum) intrusion correspond with high(low) water slacks moments.

**Figure 6b and 6d. The estuary becomes 'inverse', with higher salinities than at the mouth.**

In the revised version of the article, in which the model has been adapted as mentioned above, Figures 5 and 6 have been updated. This version presents a more realistic behavior and, by using a more accurate dispersion coefficient, the δ values are lower. Consequently, the increase observed in Figures 6b and 6d is smaller because the increase and decrease of the δ values used in the different experiments are smaller. However, the underlying concept and reasoning presented in the manuscript remain unchanged, although they have been adjusted to reflect these new results. In both figures, the time series of velocity and salinity are also included, overlaid for the Bonanza

section (km 4), allowing the moments of maximum and minimum salinity concentration to be shown, as represented in Figures 5c-f and 6c-f.

New Figure 6 (Fig 5 in last manuscript version) and Figure 7 (Fig 6 in last manuscript version) of the manuscript are presented below:

[Figure]

**Figure. 6.** Superposition of current velocity (ms⁻¹) time series and Salinity (psu) time serie at Bonanza station (4 km). black dots mean maximum and minimum salinity moments selected for MG2 (a) and MG3 (b) oceanographic campaigns**.** Series of salinity (psu) along the Guadalquivir estuary (km) between real flow and various reductions in freshwater flow for MG2 (c) and MG3 (d). In c and d, the red lines represent experiment (i), the blue lines correspond to experiment (ii) and the cyan lines are experiment (iii). (b) and (d) are the series of salinities using the real freshwater flow and greater freshwater flows for MG2 and MG3, respectively (Experiment (iv) is represented by the blue line, experiment (v) is green line and experiment vi is presented by pink lines). The solid lines represent the time of maximum salinity at Bonanza, and the dashed lines represent the time of minimum salinity at Bonanza. Color dots represent the km of maximum differences between each experiment with experiment (i).

[Figure]

**Figure 7***:* Superposition of current velocity (ms⁻¹) time series and Salinity (psu) time serie at Bonanza station (4 km). black dots mean maximum and minimum salinity moments selected for MG2 (a) and MG3 (b) oceanographic campaigns**.** Series of salinity (psu) along the Guadalquivir estuary (km) under a reduction of water withdrawals values are presented in (c) and (d) where original value are represented by blue lines (Experiment i), experiment (ii) corresponding to a smaller reduction is presented in black and a higher reduction of water withdrawal value is presented in red (Experiment (iii). (e) and (f) correspond to experiments increasing water withdrawal values. Original value is presented in blue (Experiment i), and different progressive increases are presented by black lines (Experiment iv), red lines (Experiment v) and green lines (experiment vi). The solid lines represent the time of maximum salinity at Bonanza, and the dashed lines represent the time of minimum salinity at Bonanza. Color dots represent the km of maximum differences between each experiment with experiment (i).

**Conclusions**

**L452-455. I don't think these (plausible) impacts are conclusions of the present work.**

Thank you, we have removed these lines from this section and included them in Section 3.5.2.

**L461-464. That depends on the discharge released from the dam during high riverflow conditions. Freshwater discharges of about 100-200m3/s move off the salt intrusion to the estuary mouth but tides are only significantly damped on the upper part of the estuary. Discharges of one order of magnitude higher damp significantly tides all along the estuary.**

Thank you, we have included this.

Line 673 in revised version:

"Furthermore, the impact of anthropogenic activities extends beyond salinity. Upstream, various physicochemical and biological variables, such as nutrients, organic matter, and contaminants, may also accumulate. The removal of the salt intrusion from the estuary depends on the magnitude of freshwater discharge. Moderate discharges (100-200 m³/s) typically shift the salt intrusion to the estuary mouth and reduce tidal currents in the upper estuary. In contrast, significantly higher discharges (approximately one order of magnitude greater) are required to dampen tidal currents throughout the entire estuary. Under such conditions, accumulated substances at the estuary mouth can be exported into the Gulf of Cadiz."